

# The upper-atmosphere extension of the ICON general circulation model

Sebastian Borchert[1], Guidi Zhou[2,⋆], Michael Baldauf[1], Hauke Schmidt[2], Günther Zängl[1], and Daniel Reinert[1]

[1]Deutscher Wetterdienst, Offenbach am Main, Germany
[2]Max-Planck-Institut für Meteorologie, Hamburg, Germany
⋆Current affiliation: College of Oceanography, Hohai University, Nanjing, China

**Correspondence:** Sebastian Borchert, Deutscher Wetterdienst, Frankfurter Str. 135, 63067 Offenbach am Main, Germany (sebastian.borchert@dwd.de)

**Abstract.** How the upper-atmosphere branch of the circulation contributes to and interacts with the circulation of the middle and lower atmosphere is a research area with many open questions. Inertia-gravity waves, for instance, have moved in the focus of research as they are suspected to be key features in driving and shaping the circulation. Numerical atmospheric models are an important pillar for this research. We use the ICOsahedral Non-hydrostatic (ICON) general circulation model, which is
a joint development of the Max Planck Institute for Meteorology (MPI-M) and the German Weather Service (DWD), and provides, e.g., local mass conservation, a flexible grid nesting option and a non-hydrostatic dynamical core formulated on an icosahedral-triangular grid. We extended ICON to the upper atmosphere and present here the two main components of this new configuration named UA-ICON: an extension of the dynamical core from shallow- to deep-atmosphere dynamics, and the implementation of an upper-atmosphere physics package. A series of test cases and climatological simulations show that
UA-ICON performs satisfactorily and is in good agreement with the observed global atmospheric circulation.

## 1  Introduction

In climate simulations and numerical weather prediction (NWP), there are ongoing efforts to raise the upper model lid, acknowledging possible influences of middle- and upper-atmosphere dynamics on tropospheric weather and climate (e.g., Thompson et al., 2002; Scaife et al., 2012; Charlton-Perez et al., 2013). The dynamics of the large-scale flow in the middle and upper at-
mosphere is determined, for instance, by the interaction with small-scale gravity waves. These waves, predominantly forced in the troposphere, can propagate vertically until they become unstable and break, by which, e.g., momentum carried along their path is deposited to the atmospheric background flow. This is an important route of momentum flux from the lower atmosphere to the middle and upper atmosphere, which shapes the meridional circulation in the latter regions (e.g., Fritts and Alexander, 2003; Kim et al., 2003). To have a model at hand that allows to study such processes on a wide range of spatial and temporal
scales, was one of our central motivations for the upper-atmosphere extension of the ICON model which we present in the following.





The ICOsahedral Non-hydrostatic (ICON) general circulation model (Zängl et al., 2015; Dipankar et al., 2015; Heinze et al., 2017; Giorgetta et al., 2018; Crueger et al., 2018), a joint development of the Max Planck Institute for Meteorology (MPI-M) and the German Weather Service (DWD), is designed as a unified modelling system to allow simulations from the turbulent scales in large-eddy simulations (LES) up to climatological time scales. Its triangular horizontal grid provides an almost

uniform horizontal resolution, and the non-hydrostatic dynamical core makes it applicable on a wide range of spatial scales. The discretization of the governing equations makes use of a combination of finite-difference and finite-volume methods, e.g., to ensure mass conservation which is an important property for climate simulations (e.g., Staniforth and Wood, 2008). In addition to a limited-area mode ICON offers a grid-nesting option (one-, or two-way nesting) for local refinement up to potentially very high horizontal resolutions. With three distinct packages of physics parameterizations ICON meets the different needs

of climate simulations, NWP and LES. To prepare ICON for simulations with model tops in the lower thermosphere, some extensions of the dynamical core and the physics parameterizations are necessary as presented in this work.

Among the most important approximations which are applied to the dynamical core of ICON in its standard configuration are the shallow-atmosphere approximation and traditional approximation (e.g., Phillips, 1966; White and Bromley, 1995; White et al., 2005; Staniforth and Wood, 2008). These approximations are applied to the mapping of the budget equations on

spherical coordinates relative to the center of the Earth. The shallow-atmosphere approximation is basically associated with the neglect of terms related to the spherical curvature of the atmosphere as well as variations of the gravitational field by assuming the field strength to be constant (see Thuburn and White (2013) for a detailed examination of the metrical implications of this approximation). The traditional approximation refers to neglecting the contribution of the horizontal component of the Earth's angular velocity to the Coriolis acceleration. Following the usual terminology we will call the system of equations

with and without the two approximations the shallow-atmosphere equations and the deep-atmosphere equations, respectively. Both approximations are generally applied together, in order to satisfy the conservation of the energy, the axial component of angular momentum, and the potential vorticity, as the possible measures of physical consistency (in their shallow-atmosphere reformulations, e.g. Phillips (1966); White and Bromley (1995); Staniforth and Wood (2003)). However, as Tort and Dubos (2014) have shown, it is possible to extend the shallow-atmosphere equations in such a way that the full Coriolis acceleration

can be retained without violating physical consistency.

The accuracy of the shallow-atmosphere and traditional approximation can be estimated by comparing the magnitude of the terms, which are present in the deep-atmosphere equations, but neglected in the shallow-atmosphere equations to the magnitude of the terms that are present in both systems. Such scale analysis has been used, for instance, by White and Bromley (1995), to show that for diabatically driven flows in the tropics and planetary scale flows the neglected terms of the Coriolis acceleration

might reach magnitudes up to about $10\%$ of the magnitude of key terms of the shallow-atmosphere momentum budget. On the other side, normal mode analyses, done by Thuburn et al. (2002a) for the deep-atmosphere equations, and by Kasahara (2003) with focus on a Boussinesq model featuring the full Coriolis acceleration, show that the differences in the spatial structure and the frequencies of the energetically most significant modes between the shallow- and the deep-atmosphere equations are relatively small (with differences in the frequency magnitude being typically less than about $1\%$, Thuburn et al. (2002b);

Kasahara (2003)). Both, the scale analysis and the normal-mode analysis are important tools to figure out the differences



between the deep and shallow-atmosphere equations. However, the applicability of the results on long-term integrations might be limited. The systematic errors introduced by the approximations, albeit small in magnitude, could accumulate over time and lead to significantly different flow patterns of the model atmosphere. This might be especially important for the large-scale circulations of the middle and upper atmosphere.

5    Furthermore, in view of the ever increasing computational power, some approximations that were meaningful under the restrictions of past computer architectures, might nowadays lose their justification. Therefore, we decided to expand the dynamical core of ICON by a deep-atmosphere option.

Examples for other models that use a deep-atmosphere formulation, or offer the option to do so, are the Met Office's Unified Model (UM) (Cullen et al., 1997; White et al., 2005; Staniforth and Wood, 2008), the Non-hydrostatic Icosahedral Atmospheric Model (NICAM) (e.g., Tomita and Satoh, 2004), the Ocean Land Atmosphere Model (OLAM) (e.g., Walko and Avissar, 2008), the MCore model by Ullrich and Jablonowski (2012), and the Finite Volume Model (FVM) of the Integrated Forecasting System (IFS) developed at the the European Centre for Medium-Range Weather Forecasts (ECMWF) (e.g., Smolarkiewicz et al., 2016). An overview of some of these models can be found in Ullrich et al. (2017).

Apart from the dynamics, the physics parameterizations are the second important model pillar that has to be extended for applications including part of the upper atmosphere. The ICON model offers basically three different physics packages: one which has been largely adopted from the ECHAM-model intended for climate simulations (e.g., Stevens et al., 2013; Giorgetta et al., 2018; Crueger et al., 2018), another one which is used for NWP (some aspects of which can be found in Zängl et al., 2015), and a third one for LES (Dipankar et al., 2015). The upper-atmosphere-specific physics parameterizations have been integrated into the ECHAM and NWP packages, but we will focus on the extended ECHAM physics package in the remainder of this work. To avoid confusion in the following, a side note on our terminology may be in order: if we discuss the physics parameterizations, we typically use the attribute "upper-atmosphere", to denote that their effects become significant in and above, say, the upper-mesosphere-lower-thermosphere-region. In contrast the attribute "deep-atmosphere" is used in the context of the modifications of the dynamical core, since they apply to the entire air column, and no more or less well-defined vertical significance threshold can be made out for them. If we address both extensions as a whole, we use again the attribute "upper-atmosphere".

While the model lid is raised and the deep-atmospheric dynamics is applied, the model cannot produce physically reasonable results without the extended physics parameterizations. In fact, our experience shows that when the model lid is raised to above $\sim 85\,\mathrm{km}$, the model without the extended physics would suffer from strong numerical instability, which could only be suppressed by using extremely large and unphysical numerical damping. The reason is related to the characteristics of the upper atmosphere, most importantly the rarefied air and the broader spectrum of incoming solar irradiance, which give rise to some physical phenomena that are negligible in the lower atmosphere and thus not parameterized in the current model, but become crucial in maintaining the upper atmospheric dynamics and thermodynamics.

One of such physical phenomena is molecular diffusion of momentum and heat, which is of negligible magnitude in the lower atmosphere compared to turbulent diffusion. In the upper atmosphere, as turbulence dies away and the air molecules are capable of travelling a long distance, molecular diffusion becomes dominant. Besides molecules, the upper atmosphere is also abundant





with atoms and radicals produced as a result of photolysis. Chemical heating is the release of heat by recombination reactions between atoms or radicals, and is of particular importance in the upper atmosphere where photolysis products can travel large distances before recombining. Moreover, the higher altitude also means that the upper atmosphere would receive and absorb more solar irradiance in higher frequencies than at lower levels and on the surface. This brings the need of parameterizing

the ultraviolet radiation including the Schumann-Runge bands and continuum, and the extreme ultraviolet bands. The solar radiation also acts to ionize the atmosphere, establishing the ionosphere from about $60\,\mathrm{km}$. The electronically charged ions in the ionosphere are then aligned with the magnetic field of the Earth, thus creating a drag and a heating source to the neutral mass flow. Further, the usual assumption of local thermodyamical equilibrium (LTE) does not hold in the upper atmosphere as a consequence of the low collision frequency between particles, therefore some modification must be made to the longwave

radiation parameterization. The heating and the eddy diffusion of momentum and heat generated by the breaking of gravity waves have to be taken into account near the mesopause, too.

    The outline of the paper is as follows: in section 2 we describe the modifications and extensions to the dynamical core and to the physics parameterizations, followed by a presentation of results from test cases and climate simulations for the evaluation of the new implementations in section 3. We close with a conclusion in section 4.

## 15  2   Model extension to the upper atmosphere

### 2.1   Deep-atmosphere dynamics

The dynamical core of the standard configuration of ICON makes use of the shallow-atmosphere approximation, which mainly consists in simplifying the governing equations measured in a spherical coordinate system in the following way: the radial distance of an air parcel to the center of the Earth $r$ is approximated by the radius of the Earth $a$, and metrical terms which result from the unit vectors of the coordinate system to be functions of position are neglected. In addition the traditional

approximation is applied, by which the acceleration due to the horizontal component of the angular velocity of the Earth is neglected (Phillips, 1966; Staniforth and Wood, 2003). For atmospheric models having a model top below, say, the mesopause region at an altitude of about 70 to $100\,\mathrm{km}$, the shallow-atmosphere approximation is likely a very good approximation (e.g., Ullrich et al., 2014), albeit some adverse impacts might exist, for instance in the tropics where the cosine of latitude is of order

one, questioning the neglect of the non-traditional part of the Coriolis accleration to some extent (White and Bromley, 1995; White et al., 2005). If the model top is raised into the lower thermosphere, the systematic errors introduced by the shallow-atmosphere approximation might start to outweigh its benefits, especially on a climatological time scale. Just to give a simple example concerning the mass of air: assuming an isothermal atmosphere in hydrostatic balance, the mass in a grid cell in a deep model atmosphere is $m(z) \approx m_0(z)(r/a)^2 \exp[(z - z_g)/H]$, where $m_0(z)$ denotes the mass contained by a cell under

shallow-atmosphere approximation, $z$ is the altitude of the cell center above a spherical shell with a radius equal to the mean radius of the Earth $a = 6371229\,\mathrm{m}$, $r = a + z$ is the distance from the center of the sphere, $H$ is the temperature scale height and $z_g$ is the geopotential height motivated in Eq. (8) below. The comparison assumes that $H$ and the air density at $z = 0$ are the same in a shallow and deep model atmosphere. The factor after $m_0$ is always $> 1$ above sea level, and already at an altitude





of $z \sim 100\,\mathrm{km}$ there is about $20\%$ more mass in a grid cell according to this estimate (assuming $H \approx 10\,\mathrm{km}$). This is due to the fact that the gravitational acceleration decreases with height in the deep atmosphere, so that the atmosphere can extend further radially, which finds its expression in the factor $\exp[(z - z_g)/H]$. In addition, the cell volume in the deep atmosphere increases with height, so that it contains more mass at a given density (the factor $(r/a)^2$, compare section 2.1.2). The relevance of this

for the dynamics is certainly a separate question, nevertheless, we think the extension of the dynamical core from shallow- to deep-atmosphere dynamics is an important component for UA-ICON.

### 2.1.1  Model equations

We restrict our considerations to the dry atmosphere hereafter, to focus on the particular aspects of the deep-atmosphere dynamics. The budget equations for the momentum, mass and heat are rearranged to the following set of prognostic equations

(White et al., 2005; Zängl et al., 2015)

$$\frac{\partial \boldsymbol{v}}{\partial t} + \boldsymbol{v} \cdot \boldsymbol{\nabla} \boldsymbol{v} + 2\boldsymbol{\Omega} \times \boldsymbol{v} + \boldsymbol{\Omega} \times (\boldsymbol{\Omega} \times \boldsymbol{r}) = -c_p \theta \boldsymbol{\nabla} \pi - \boldsymbol{\nabla} \phi_g + \rho^{-1} \boldsymbol{F}, \tag{1}$$

$$\frac{\partial \rho}{\partial t} + \boldsymbol{\nabla} \cdot (\boldsymbol{v}\rho) = 0, \tag{2}$$

$$\frac{\partial \pi}{\partial t} + \frac{R}{c_v} \frac{\pi}{\rho\theta} \boldsymbol{\nabla} \cdot (\boldsymbol{v}\rho\theta) = \frac{R}{c_v c_p \rho\theta} Q, \tag{3}$$

and the equation of state reads

$$\rho\theta = \frac{p_{00} \pi^{c_v/R}}{R} \tag{4}$$

where $\boldsymbol{v} = \boldsymbol{v_h} + w\boldsymbol{e_r}$ is the wind vector split in horizontal and radial components, $\boldsymbol{\Omega}$ is the angular velocity vector of the Earth, and $\boldsymbol{r}$ denotes the position relative to the center of the Earth. In addition, $\pi = (p/p_{00})^{R/c_p}$ denotes the Exner pressure relative to the reference pressure $p_{00} = 1000\,\mathrm{hPa}$, $\theta = T/\pi$ is the potential temperature, $\phi_g$ denotes the gravitational potential, and $c_p = 1004.64\,\mathrm{J\,K^{-1}\,kg^{-1}}$, $c_v = 717.6\,\mathrm{J\,K^{-1}\,kg^{-1}}$, $R = c_p - c_v$ are the specific heat capacities at constant pressure and

volume, and the gas constant for dry air, respectively. Finally, $\boldsymbol{F}$ denotes velocity tendencies due to dissipative processes or other parameterized momentum sources, and $Q$ denotes Exner pressure tendencies due to diabatic processes.

We have added the centrifugal acceleration $\boldsymbol{\Omega} \times (\boldsymbol{\Omega} \times \boldsymbol{r})$ in Eq. (1), since it is of importance for a test case presented in section 3.1.1. In the standard configuration of ICON, where this term is absent, we have apparent gravity $-\boldsymbol{\nabla}\phi_{g+c}$ (true gravity plus centrifugal acceleration) on the right-hand side of Eq. (1), to which the spherical-geopotential approximation is applied: in

case of Eq. (1) the vertical unit vector of the coordinate system, in which the atmospheric flow is measured, is defined as the normal vector of the spherical equipotential surfaces $\boldsymbol{e_r} = \boldsymbol{\nabla}\phi_g/|\boldsymbol{\nabla}\phi_g|$. In case of apparent gravity we continue to write $\boldsymbol{e_r} = \boldsymbol{\nabla}\phi_{g+c}/|\boldsymbol{\nabla}\phi_{g+c}|$, but the equipotential surfaces are still approximated by spherical surfaces, neglecting the oblateness caused by the centrifugal acceleration (Gill, 1982; White et al., 2005; Vallis, 2006; Staniforth and Wood, 2008). Equation (3) follows from the budget equation of the heat density $\partial(\rho s)/\partial t + \boldsymbol{\nabla} \cdot (\boldsymbol{v}\rho s) = Q/T$, by formulation in terms of the potential

temperature $\partial(\rho\theta)/\partial t + \boldsymbol{\nabla} \cdot (\boldsymbol{v}\rho\theta) = Q/(c_p\pi)$ and multiplication with $R\pi/(c_v\rho\theta)$, where $s = s_0 + c_p \ln(\theta/\theta_0)$ denotes the mass-specific heat (or entropy) and $s_0$, $\theta_0$ are some reference values (e.g., Gassmann and Herzog, 2015).



The advection term in Eq. (1) is expressed in the so-called 2d-vector-invariant formulation (e.g., Phillips, 1966; Sadourny, 1972; Vallis, 2006)

$$\boldsymbol{v} \cdot \boldsymbol{\nabla} \boldsymbol{v} = \boldsymbol{\omega_h} \times \boldsymbol{v_h} + \boldsymbol{\nabla_h} \frac{|\boldsymbol{v_h}|^2}{2} + w \frac{\partial \boldsymbol{v_h}}{\partial r} + \boldsymbol{v} \cdot \boldsymbol{\nabla} (w \boldsymbol{e_r}), \tag{5}$$

where $\boldsymbol{\omega_h} = \boldsymbol{\nabla_h} \times \boldsymbol{v_h} = \zeta \boldsymbol{e_r}$ is the vertical component of the relative vorticity. During the development of ICON, it turned out that it is advantageous to separate the Exner pressure into a hydrostatically balanced part and a deviation from it $\pi = \pi_0(r) + \pi'$, where $-c_p \theta_0 \boldsymbol{\nabla} \pi_0 - \boldsymbol{\nabla} \phi_g = 0$. Thus the right-hand side of Eq. (1) reads

$$-c_p \theta \boldsymbol{\nabla} \pi - \boldsymbol{\nabla} \phi_g = -c_p \theta \boldsymbol{\nabla} \pi' - c_p \theta' \frac{\mathrm{d} \pi_0}{\mathrm{d} r}. \tag{6}$$

The background temperature profile of the hydrostatically balanced part is defined by (Zängl, 2012)

$$T_0(z_g) = T_{str} + (T_{sl} - T_{str}) \exp \left( -\frac{z_g}{H_{scal}} \right), \tag{7}$$

where $z_g = z/(1 + z/a)$ denotes the geopotential height, and $T_{sl} = 288.15 \, \mathrm{K}$ as well as $T_{str} = 213.15 \, \mathrm{K}$ are characteristic temperature values at sea level ($z_g = 0$) and in the stratosphere, respectively. $H_{scal} = 10000 \, \mathrm{m}$ is a characteristic (geopotential) temperature scale height. The geopotential height follows as "natural" height measure from the gravitational acceleration (Gill, 1982; Wood and Staniforth, 2003; Wood et al., 2014; Ullrich et al., 2017)

$$\boldsymbol{\nabla} \phi_g \cdot \mathrm{d} \boldsymbol{r} = g \left( \frac{a}{r} \right)^2 \mathrm{d} r = g \left( \frac{1}{1 + \frac{z}{a}} \right)^2 \mathrm{d} z = g \mathrm{d} \left( \frac{z}{1 + \frac{z}{a}} \right) = g \mathrm{d} z_g, \tag{8}$$

with the geopotential $\phi_g = -g a^2/r + \phi_{g,0}$, where $\phi_{g,0}$ denotes some constant, and $g = 9.80665 \, \mathrm{m \, s^{-2}}$ is the mean gravitational acceleration at sea level (this is actually the magnitude of apparent gravity, but since the contribution of the centrifugal acceleration is relatively small, we use this value also for the true gravity). The values of $z_g$ range from $\lim_{z \to -a} z_g = -\infty$ to $\lim_{z \to \infty} z_g = a$.

The profile (7) allows to integrate the hydrostatic equilibrium analytically. Note, if the centrifugal acceleration is treated in its explicit form $\boldsymbol{\Omega} \times (\boldsymbol{\Omega} \times \boldsymbol{r})$, it is not taken into account in the hydrostatic equilibrium outlined above.

In ICON the governing equations (1) to (3) are measured in a spherical coordinate system with local unit vectors $\boldsymbol{e_t}(\boldsymbol{r})$, $\boldsymbol{e_n}(\boldsymbol{r})$ and $\boldsymbol{e_r}(\boldsymbol{r})$ (forming a right-handed system in this order), where $\boldsymbol{e_t}$ and $\boldsymbol{e_n}$ are the horizontal unit vectors tangential and normal to an edge separating two adjacent triangular cells, and $\boldsymbol{e_r}$ is defined in the center of the bottom and top surfaces of a cell (see Fig. 1)[1]. In order to find expressions for the projection of the momentum equation (1) onto the unit vectors of the local edge coordinate system and $\boldsymbol{e_r}$, we can make use of the projections in the standard geographical coordinate system, with its horizontal unit vectors in meridional (south-north) and zonal (west-east) directions, $\boldsymbol{e_\varphi}$ and $\boldsymbol{e_\lambda}$ (e.g., Zdunkowski and Bott,

---

[1]It would be more correct to refer to the horizontal unit vectors as $\boldsymbol{e_t}(\boldsymbol{r_j})$ and $\boldsymbol{e_n}(\boldsymbol{r_j})$, since they are defined only at the grid triangle edges $j$. In contrast to the zonal and meridional unit vectors of the geographical coordinate system, $\boldsymbol{e_\lambda}(\boldsymbol{r})$ and $\boldsymbol{e_\varphi}(\boldsymbol{r})$, $\boldsymbol{e_t}$ and $\boldsymbol{e_n}$ do not converge to a differentiable continuum for the number of triangles going to infinity, since the jumps in orientation from one edge to another adjacent edge remain, whatever the horizontal grid resolution. Nevertheless, if computations require spatial derivatives of vectorial quantities, they can be performed in geographical coordinates and the result can projected onto the $\boldsymbol{e_t}(\boldsymbol{r_j})$ and $\boldsymbol{e_n}(\boldsymbol{r_j})$. So we regard $\boldsymbol{e_t}$ and $\boldsymbol{e_n}$ as an "indirectly differentiable" continuum and write $\boldsymbol{e_t}(\boldsymbol{r})$ and $\boldsymbol{e_n}(\boldsymbol{r})$, in order to simplify matters.

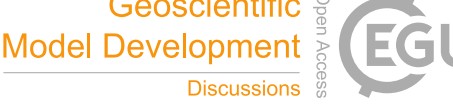



**(a)**                                                       **(b)**

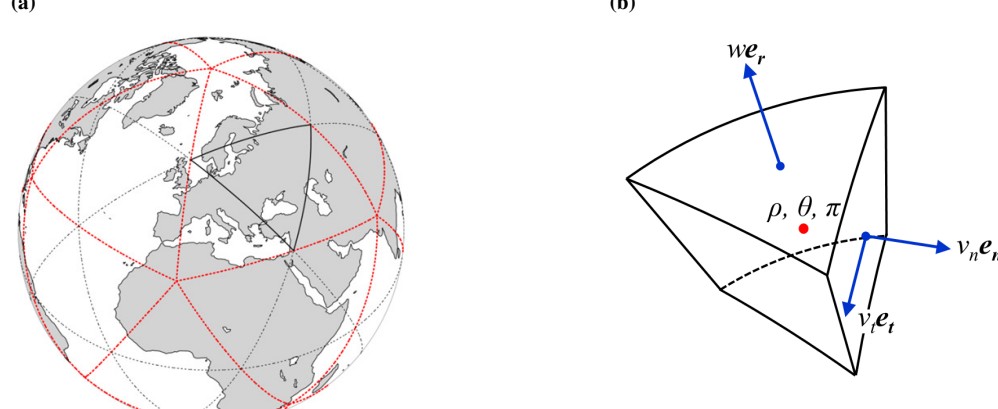

**Figure 1.** Schematic illustration of a global horizontal triangular grid as used by ICON (a), and of a grid cell (b). The grid is of R2B0-type (twofold root division, zero bisections). The points in and on the grid cell are shown, where the (prognostic) variables are defined: density $\rho$, potential temperature $\theta$, and Exner pressure $\pi$ in the cell center, tangential and normal wind components $v_t$, and $v_n$ on the side faces (corresponding to the three primal edges of a triangle on the horizontal grid), and vertical wind component $w$ on the bottom and top surfaces of the cell (assuming $v_t, v_n, w > 0$ for the cell drawing).

2003). The Coriolis and centrifugal accelerations are excluded from the following consideration, since their projections onto the unit vectors of the local coordinate systems of the triangular grid can be determined directly. Each edge is part of a great circle of the Earth, which can be regarded as an imaginary equator. Rotating the geographical coordinate system accordingly and in such a way that $\boldsymbol{e_t}$ is parallel to $\boldsymbol{e_\lambda'}$ at the considered location (where the prime indicates this coordinate transformation),

we would find $\boldsymbol{e_n}$ being parallel to $\boldsymbol{e_\varphi'}$ (the radial unit vectors of both systems are parallel anyway). The governing equations for the velocity components $\boldsymbol{v} = u\boldsymbol{e_\lambda} + v\boldsymbol{e_\varphi} + w\boldsymbol{e_r}$ (or here $\boldsymbol{v} = u'\boldsymbol{e_\lambda'} + v'\boldsymbol{e_\varphi'} + w\boldsymbol{e_r}$ ) can be found in many textbooks (e.g., Gill, 1982; Zdunkowski and Bott, 2003; Holton, 2004; Vallis, 2006). Evaluating them at the equator ($\varphi^{(\prime)} = 0$) yields almost immediately the components of the governing equations for $\boldsymbol{v} = v_t\boldsymbol{e_t} + v_n\boldsymbol{e_n} + w\boldsymbol{e_r}$ in the local edge coordinate systems, if we identify $v_t$ with $u'$ and $v_n$ with $v'$. The equation for the vertical velocity component $w$ in the local coordinate system defined

in the center of the vertical cell interfaces can be found in a similar way. Together they read

$$\frac{\partial v_{\mathrm n}}{\partial \mathrm t} + \left\{\frac{a}{r}\right\}\frac{\partial K_h}{\partial \mathrm n} + (\zeta + f_r)\,v_t + w\left(\frac{\partial v_{\mathrm n}}{\partial r} + \frac{v_{\mathrm n}}{r} - f_t\right) + \underline{\Omega^2 r \sin(\varphi)\cos(\varphi)\boldsymbol{e_\varphi}\cdot\boldsymbol{e_n}} = -c_p\theta\left\{\frac{a}{r}\right\}\frac{\partial \pi}{\partial \mathrm n} + \frac{F_{\mathrm n}}{\rho}, \tag{9}$$

$$\frac{\partial w}{\partial \mathrm t} + v_{\mathrm n}\left(\left\{\frac{a}{r}\right\}\frac{\partial w}{\partial \mathrm n} - \frac{v_{\mathrm n}}{r} + f_t\right) + v_t\left(\left\{\frac{a}{r}\right\}\frac{\partial w}{\partial \mathrm t} - \frac{v_t}{r} - f_{\mathrm n}\right) + w\frac{\partial w}{\partial r} - \underline{\Omega^2 r \cos^2(\varphi)} = -c_p\left[(\theta_0 + \theta')\frac{\partial \pi'}{\partial r} + \theta'\frac{\mathrm d\pi_0}{\mathrm d r}\right] + \frac{F_r}{\rho},$$
$$\tag{10}$$

where $K_h = (v_t^2 + v_n^2)/2$ is the horizontal mass-specific kinetic energy, and $f_r = 2\Omega\sin(\varphi)$, $f_{t,n} = 2\Omega\cos(\varphi)\boldsymbol{e_\varphi}\cdot\boldsymbol{e_{t,n}}$ are the Coriolis parameters. The values of the projections $\boldsymbol{e_\varphi}\cdot\boldsymbol{e_{t,n}}$ are already provided by the standard shallow-atmosphere

configuration of ICON, so they pose no additional problem. Here and in the following we will formulate the deep-atmosphere equations as a modification of the shallow-atmosphere equations. This simplifies the comparison and corresponds actually to





the way we have implemented the deep-atmosphere dynamics in ICON. For instance, we made use of the expansion of the gradient in the local coordinate systems $\boldsymbol{\nabla} = \boldsymbol{e_t}\{a/r\}\partial/\partial\mathrm{t} + \boldsymbol{e_n}\{a/r\}\partial/\partial\mathrm{n} + \boldsymbol{e_r}\partial/\partial r$, if $\boldsymbol{\nabla} = \boldsymbol{e_t}\partial/\partial\mathrm{t} + \boldsymbol{e_n}\partial/\partial\mathrm{n} + \boldsymbol{e_z}\partial/\partial z$ is its expansion under shallow-atmosphere approximation (with $\partial/\partial\mathrm{t}$ and $\partial/\partial\mathrm{n}$ denoting the horizontal derivatives along great circle arcs under shallow-atmosphere approximation, i.e., $r = a + z \approx a$, and $\boldsymbol{e_z} = \boldsymbol{e_r}$, $\partial/\partial z = \partial/\partial r$). So the factors $\{\cdot\}$

become 1 under the shallow-atmosphere approximation. In addition, the underlined terms in Eqs. (9) and (10) are neglected in the shallow-atmosphere equations.

The velocity component $v_\mathrm{t}$ is reconstructed diagnostically from $v_\mathrm{n}$ (e.g., through an interpolation by radial basis functions).

At the end of this section, we have a brief look at the expressions $Q$ and $\boldsymbol{F}$ in Eqs. (1) and (3). Although this might be a matter of taste, they can be separated into two parts: $Q = Q_\mathrm{dyn} + Q_\mathrm{phy}$, $\boldsymbol{F} = \boldsymbol{F_\mathrm{dyn}} + \boldsymbol{F_\mathrm{phy}}$, of which the first part is still counted

as contribution to the dynamics, whereas the second part summarizes the tendencies from the physics parameterizations, which we will not consider further in this section (if a prognostic equation for turbulent kinetic energy is part of the model, the considerations about $Q$ and $\boldsymbol{F}$ need special care, to avoid "double countings", but we explicitly exclude that case in the following considerations).

Expressions for $Q_\mathrm{dyn}$ and $\boldsymbol{F_\mathrm{dyn}}$ typically follow from the budget (and state) equations of the infinitesimal air parcels of the

continuum view-point, to which a mass-weighted Reynolds-average (Hesselberg, 1925; Favre, 1969) is applied (together with the assumption that the air flow behaves quasi-ergodically on the averaging scales, e.g., Herbert (1975); Sievers (1982))

$$\frac{\partial\bar{\rho}\hat{\chi}}{\partial t} + \boldsymbol{\nabla} \cdot \left(\hat{\boldsymbol{v}}\bar{\rho}\hat{\chi} + \overline{\boldsymbol{v}''\rho\chi''} + \bar{\boldsymbol{J}}_{\boldsymbol{\chi}}\right) = \bar{\sigma}_\chi, \tag{11}$$

where $\chi$ is some mass-specific extensive variable with its molecular flux density $\boldsymbol{J_\chi}$ and source density $\sigma_\chi$. In addition, $\overline{(\cdot)}$ denotes the Reynolds-average, $\hat{(\cdot)} = \overline{(\rho\,\cdot)}/\bar{\rho}$ the mass-weighted average, and $(\cdot)'' = (\cdot) - \hat{(\cdot)}$. The turbulent flux density is

typically parameterized by some gradient ansatz

$$\overline{\boldsymbol{v}''\rho\chi''} = -\bar{\rho}\mathbf{K}_{\hat{\chi}} \cdot \boldsymbol{\nabla}\hat{\chi}, \tag{12}$$

where the diffusion coefficient tensor $\mathbf{K}_{\hat{\chi}}$ may be some function of the state variables.

In the lower atmosphere the magnitude of the turbulent flux density is typically orders of magnitude larger than the magnitude of the molecular flux density, for what reason $\bar{\boldsymbol{J}}_{\boldsymbol{\chi}}$ might be neglected there. For $Q_\mathrm{dyn}$ this ansatz usually means some expression

similar to

$$Q_\mathrm{dyn} = \rho^{-1}\boldsymbol{\nabla} \cdot (c_p\rho\pi\mathbf{K}_\theta \cdot \boldsymbol{\nabla}\theta), \tag{13}$$

if the budget equation for the heat density $\bar{\rho}\hat{s}$ is written in terms of the potential temperature $\hat{\theta}$. We dropped again the bars and hats in Eq. (13), as this has been done right away from the beginning for Eqs. (1) to (3), which actually also represent averaged equations of the form (11). Although the ansatz (13) can lead to violations of the second law of thermodynamics (Herbert,

1975; Gassmann and Herzog, 2015; Gassmann, 2018), it is probably the most common ansatz and also applied in ICON.





The expression for $\boldsymbol{F_{\mathbf{dyn}}}$ has to parameterize the momentum transfer between neighbouring air volumes due to relative motions and typically reads (e.g., Pichler, 1984; Zdunkowski and Bott, 2003)

$$\boldsymbol{F_{\mathbf{dyn}}} = -\boldsymbol{\nabla} \cdot \mathbf{F},$$
$$\mathbf{F} = -\kappa_1 \left( \frac{\boldsymbol{\nabla v} + \boldsymbol{\nabla v}^\intercal}{2} - \frac{\boldsymbol{\nabla} \cdot \boldsymbol{v}}{3} \mathbf{1} \right) - \kappa_2 \frac{\boldsymbol{\nabla} \cdot \boldsymbol{v}}{3} \mathbf{1} - \kappa_3 \left( \frac{\boldsymbol{\nabla v} - \boldsymbol{\nabla v}^\intercal}{2} \right), \tag{14}$$

where $\mathbf{F}$ denotes the turbulent momentum flux density tensor, $\mathbf{1}$ is the identity tensor and $(\cdot)^\intercal$ denotes the tensor transpose. The three terms on the right-hand side of Eq. (14) correspond to the three forms of (linear) relative motion, into which $\boldsymbol{\nabla v}$ can be expanded. These are the volume-conserving shear motion (or deformation), the shape-conserving compression/stretching, and the volume- and shape-conserving rotation (Pichler, 1984; Zdunkowski and Bott, 2003), respectively. The coefficients $\kappa_1$, $\kappa_2$, and $\kappa_3$ may be functions of the state variables. If $\mathbf{F}$ parameterizes the symmetric Reynolds stress tensor $\overline{\boldsymbol{v}'' \rho \boldsymbol{v}''}$ the

antisymmetric rotational part of (14) with coefficient $\kappa_3$ has to be dropped. So by means of the Reynolds average ansatz any flux between the external angular momentum density of an air volume $\boldsymbol{l_{\mathbf{ext}}} = \boldsymbol{r} \times \rho \boldsymbol{v}$ and its internal angular momentum density $\boldsymbol{l_{\mathbf{int}}} = \boldsymbol{\Theta} \cdot \boldsymbol{\omega}$ can be excluded (where $\boldsymbol{\Theta}$ and $\boldsymbol{\omega}$ are the tensor of the moment of inertia density and the angular velocity, respectively)[2]. This is a standard approximation in atmospheric physics, regardless of the averaging scale (Herbert, 1978). Budgeting $\boldsymbol{l_{\mathbf{int}}}$ as a further "vectorial" extensive variable in addition to the momentum density $\boldsymbol{p} = \rho \boldsymbol{v}$ would of course be

impossible for today's models, due to the computational costs.

In addition, a physically consistent implementation would actually require to consider a contribution from the corresponding frictional heating to $Q_{\mathrm{dyn}}$ which is proportional to $-\mathbf{F}^\intercal \cdot \cdot \boldsymbol{\nabla v}$ [3]. Its magnitude is generally assumed to be small, nevertheless, it is a cumulative process, since the sign of this term is always positive. Due to its computational costs, this term is neglected in ICON.

We refrained from deep-atmosphere modifications of the dissipative terms $Q_{\mathrm{dyn}}$ and $\boldsymbol{F_{\mathbf{dyn}}}$ for reasons exemplified on $\boldsymbol{F_{\mathbf{dyn}}}$: the expansion of (14) in Cartesian coordinates, as currently implemented in ICON, is still managable with regard to its computational costs. However, if (14) is expanded in spherical coordinates, we are faced with numerous additional metric terms, due to the non-vanishing spatial derivatives of the unit vectors $\boldsymbol{e_t}$, $\boldsymbol{e_n}$, and $\boldsymbol{e_r}$ (compare e.g. Baldauf and Brdar, 2016). Their rigorous implementation into the dynamical core would cause considerable additional computational costs, which are not af-

fordable for us for the time being. Furthermore, terms structurally similar to Cartesian expansions of (14) are implemented in the dynamical core of ICON as explicit numerical damping to ensure stability. Since they represent no physical process, at least not explicitly, it is not clear, if one would take any advantage from a deep-atmosphere modification. Not to mention that the numerical damping typically neglects terms from the full Cartesian expansions of (14), so that coordinate-independence might be lost and a transformation into spherical coordinates becomes a more or less subjective plausibility consideration.

---

[2] Here, we regard (or define) turbulence as that part of the internal motions in the considered air volume that has neither a net momentum nor a net angular momentum. So the internal angular momentum is not part of the turbulence.

[3] Here $\cdot\cdot$ denotes the double scalar product. Given the two dyadic products $\boldsymbol{ab}$ and $\boldsymbol{cd}$ of the arbitrary vectors $\boldsymbol{a}$, $\boldsymbol{b}$, $\boldsymbol{c}$ and $\boldsymbol{d}$, their double scalar product is defined as: $\boldsymbol{ab} \cdot \cdot \boldsymbol{cd} = (\boldsymbol{b} \cdot \boldsymbol{c})(\boldsymbol{a} \cdot \boldsymbol{d}) = \boldsymbol{a} \cdot (\boldsymbol{b} \cdot \boldsymbol{cd}) = (\boldsymbol{ab} \cdot \boldsymbol{c}) \cdot \boldsymbol{d} = (\boldsymbol{d} \cdot \boldsymbol{a})(\boldsymbol{c} \cdot \boldsymbol{b}) = \boldsymbol{cd} \cdot \cdot \boldsymbol{ab}$ (compare Wilson, 1929; Zdunkowski and Bott, 2003). So the double scalar product of two arbitrary tensors $\mathbf{A}$ and $\mathbf{B}$, measured in some normal basis $\boldsymbol{e_i}$, $i = 1, 2, 3$: $\mathbf{A} = A_{ij} \boldsymbol{e_i} \boldsymbol{e_j}$, $\mathbf{B} = B_{kl} \boldsymbol{e_k} \boldsymbol{e_l}$ reads: $\mathbf{A} \cdot \cdot \mathbf{B} = A_{ij} B_{kl} \boldsymbol{e_i} \boldsymbol{e_j} \cdot \cdot \boldsymbol{e_k} \boldsymbol{e_l} = A_{ij} B_{kl} (\boldsymbol{e_j} \cdot \boldsymbol{e_k})(\boldsymbol{e_i} \cdot \boldsymbol{e_l}) = A_{ij} B_{ji}$.





### 2.1.2 Numerical implementation

A thorough description of the spatial and temporal discretization of the governing equations used in ICON can be found in Zängl et al. (2015), and for development stages of ICON in Bonaventura and Ringler (2005); Wan (2009) and Wan et al. (2013). Here, we will focus only on those elements of the spatial discretization that are affected by the deep-atmosphere modifications.

First we would like to point to an important simplification, which we have done in our implementation of the deep-atmosphere dynamics into ICON. Since ICON is used for operational NWP at DWD, a key criterion for new developments to be integrated into the dynamical core is computational efficiency (of course, this criterion applies to climate models as well). For this reason priority is given to efficiency over accuracy, where we assume this to be acceptable. Where possible the deep-atmosphere dynamics are realized by modification factors to the existing terms of the shallow-atmosphere dynamics. In

addition, topography is not taken into account by the deep-atmosphere modification, which, for instance, means that the dynamics in a cell of the first grid layer above the Himalaya experience the same deep-atmosphere modification as the dynamics in a cell above the ocean surface. This simplification, albeit the most severe one, has many advantages. First we avoid the complicated calculations of the geometric measures of a cell (e.g., its volume and surface areas) in spherical geometry, if they are distorted by topography and lose the center of Earth as their center of curvature. Second the above mentioned deep-atmosphere

modification factors depend only on height, which saves a considerable amount of memory and computational cost. Since topography imprints on the grid layers only up to a certain height in ICON (typically up to $16\,\text{km}$), the errors introduced by this measure are assumed to be relatively small. For instance, the difference of a typical grid cell volume at mean sea level and at $16\,\text{km}$ is of the order of a few permilles in the spherical geometry. Furthermore the mass conserving property of the dynamical core of ICON is not affected by our simplification.

We start with the modification of the gravitational acceleration (Wood and Staniforth, 2003)

$$g \rightarrow g \left\{ \frac{a}{r} \right\}^2, \tag{15}$$

with the modification factor parenthesized by braces. The formula (15) enters (10) only implicitly via the hydrostatic background state. For its computation (15) is used only once during model initialization. Next, we consider the gradient in edge normal and tangential directions

$$\text{grad}_{\text{t,n}}(\chi) = \frac{\Delta\chi}{\Delta\text{t,n}} \left\{ \frac{a}{r} \right\}, \tag{16}$$

where $\chi$ stands for the mass-specific horizontal kinetic energy $K_h$ and the Exner pressure $\pi$ in Eq. (9), and the vertical velocity $w$ in Eq. (10). $\Delta\chi/\Delta\text{t,n}$ denotes the gradient as it is computed under shallow-atmosphere approximation. The factor $a/r$ derives from the modification of the length of horizontal distances $l \rightarrow l\{r/a\}$. Like all other modification factors, it is precomputed during model initialization for the vertical positions, where the gradients have to be evaluated during run time.

Another quantity the computation of which has to be modified for the deep atmosphere is the vertical vorticity component $\zeta$ on the left-hand side of Eq. (9). $\zeta$ is computed at the vertices of the triangular cells and then interpolated to the edge centers,





where $v_\mathrm{n}$ is defined. This allows to compute the vorticity from the $v_\mathrm{n}$ using the Stokes theorem

$$\zeta = \left( \frac{1}{A_v} \sum_{i=1}^{n_e} v_{\mathrm{n},i} l_{d,i} f_{o,i} \right) \left\{ \frac{a}{r} \right\}, \tag{17}$$

where $A_v$ is the area of the hexagonal or pentagonal cells centered at the vertices, which form the dual cells to the primal triangular cells by connecting the mass points of the triangular cells around a vertex, see Fig. 2 in Wan et al. (2013). In
addition, $n_e$ is the number of the dual edges of length $l_d$ around the vertex, crossing the primal edges of the triangular cells perpendicularly ($n_e = 6$, or 5 for the 12 pentagon points of the grid, respectively). Finally, $f_o$ is an orientation factor, which is 1 or $-1$ according to whether $\boldsymbol{e}_\mathbf{n}$ is parallel or antiparallel to the cyclonic direction of the integration path. Again, the first factor on the right-hand side of Eq. (17) is the vorticity under shallow-atmosphere approximation, thus the geometric measures $A_v$ and $l_d$ are those at $r = a$. The modification factor $a/r$ results from the quotient $l_d/A_v$ in spherical geometry, and is identical
to the modification factor for the horizontal gradient (16). The Eqs. (2) and (3) are written in flux form, which guarantees mass conservation up to machine precision, and the conservation of heat as far as advective fluxes are concerned. The deep-atmosphere modifications of the flux divergences result from the quotient of area and volume $A/V$ for the 5 surfaces of a triangular cell. The modification of a cell volume itself reads

$$V = A_c(z_\mathrm{top} - z_\mathrm{bot}) \left\{ \frac{r_\mathrm{bot}^2 + r_\mathrm{bot} r_\mathrm{top} + r_\mathrm{top}^2}{3a^2} \right\}, \tag{18}$$

where $V = A_c(z_\mathrm{top} - z_\mathrm{bot})$ is the cell volume under shallow-atmosphere approximation, with the cell surface area $A_c$, and $r_\mathrm{bot} = a + z_\mathrm{bot}$, $r_\mathrm{top} = a + z_\mathrm{top}$ are the radii of the cell's bottom surface and top surface, respectively. This modification is required, for instance, if global integrals of mass and other quantities are computed for diagnostic purposes. Together with the area modification for the three side faces and the bottom and top surfaces

$$A_s = l_p (z_\mathrm{top} - z_\mathrm{bot}) \left\{ \frac{r_\mathrm{bot} + r_\mathrm{top}}{2a} \right\}, \tag{19}$$

$$A_\mathrm{bot,top} = A_c \left\{ \frac{r_\mathrm{bot,top}}{a} \right\}^2, \tag{20}$$

where $l_p$ is the length of the primal edges, we find for the flux divergences

$$\mathrm{div}(\xi) = \left( \frac{1}{A_c} \sum_{i=1}^{3} \xi_i l_{p,i} f_{o,i} \right) \left\{ \frac{3}{4} \frac{2a}{r_\mathrm{bot} + r_\mathrm{top}} \left[ 1 - \frac{r_\mathrm{bot} r_\mathrm{top}}{(r_\mathrm{bot} + r_\mathrm{top})^2} \right]^{-1} \right\}$$

$$+ \frac{\xi_\mathrm{top} \left\{ 3 \left[ 1 + \frac{r_\mathrm{bot}}{r_\mathrm{top}} + \left( \frac{r_\mathrm{bot}}{r_\mathrm{top}} \right)^2 \right]^{-1} \right\} - \xi_\mathrm{bot} \left\{ 3 \left[ 1 + \frac{r_\mathrm{top}}{r_\mathrm{bot}} + \left( \frac{r_\mathrm{top}}{r_\mathrm{bot}} \right)^2 \right]^{-1} \right\}}{z_\mathrm{top} - z_\mathrm{bot}}. \tag{21}$$

Here, $\xi_{i,\mathrm{top,bot}}$ denotes the scalar product of the flux density and $\boldsymbol{e}_\mathbf{n}$, or $\boldsymbol{e}_\mathbf{r}$ at the respective cell face. The first term on the
right-hand side of Eq. (21) sums the fluxes across the side faces, where the orientation factor $f_o$ is either 1 or $-1$ according to whether $\boldsymbol{e}_\mathbf{n}$ is parallel or antiparallel to the normal vector of the side faces from the point of view of the cell. The second term sums the fluxes across the top and bottom surfaces.

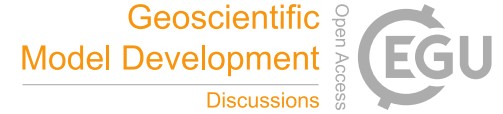

The underlined terms in Eqs. (9) and (10) are new implementations in the dynamical core. Their discrete formulation closely follows the formulation of structurally similar terms of the discrete shallow-atmosphere equations in Zängl et al. (2015).

The discretization employed in ICON necessitates the spatial interpolation of variables between cell center, edge midpoint, vertex and the center of the vertical cell interfaces (bottom and top surface, Zängl et al. (2015)), which could be affected potentially when changing from a shallow- to a deep-atmosphere formulation. In the horizontal, linear, bilinear and area-weighted interpolations are used primarily. These interpolations are assumed to take place on coordinate surfaces $z = $ const., i.e. planes in case of the shallow atmosphere (at least practically, but see Thuburn and White (2013) for the geometric implications of the shallow-atmosphere approximation), and spherical shells in case of the deep atmosphere. This means that no explicit deep-atmosphere modification is necessary, because it would be the same for the respective geometric measures (distances, areas) on a coordinate surface, which cancel each other in the interpolation. For the vertical interpolation between cell centers and the cell interfaces a linear interpolation is used in case of the shallow atmosphere. We interpret this as an interpolation along the coordinate lines $\lambda, \varphi = $ const. in case of the deep atmosphere, so that we can avoid deep-atmosphere modifications in this case, too. If we interpret the vertical interpolation, e.g., as a volume-weighted average, a modification would be necessary. Apart from the afore-mentioned standard interpolations, ICON makes use of an upwind-biased reconstruction of density and potential temperature from cell centers to the centers of the (horizontal and vertical) interfaces. These reconstructions enter for instance the divergence of mass and heat flux (Zängl et al., 2015), and make use of a Taylor-expansion up to the first order along backward trajectories, which requires to modify the corresponding gradient according to Eq. (16).

For the time integration a two-time-level predictor-corrector scheme is used, which is not directly affected by deep-atmosphere modifications. However, certain terms on the right-hand side of the governing equations, which are associated with vertical sound wave propagation, are treated implicitly in the temporal discretization, in order to allow reasonable time steps. This directly affects the discretized forms of Eqs. (3) and (10), which can be combined to yield a diagnostic, elliptic equation for the vertical wind $w$ for each vertical grid cell column of the model atmosphere (Zängl et al., 2015). Thus, some of the coefficients of this equation have to be complemented by the corresponding deep-atmosphere modification factors.

### 2.1.3 Model initialization

One motivation to implement the deep-atmosphere dynamics in ICON is to increase the accuracy of simulations with a model top $\gtrsim 100\,\mathrm{km}$. However, initial data are usually only available up to altitudes of about 70 to $80\,\mathrm{km}$. For instance, the model top is at $75\,\mathrm{km}$ in the operational NWP with ICON at DWD, and the IFS model of the ECMWF, whose operational analysis data can be used to initialize ICON, has its model top at $0.01\,\mathrm{hPa}$ [4]. To our knowledge no regular and reliable measurements with global coverage are available for the (lower) thermosphere, which could be used for some kind of data assimilation. Therefore, climatological tables appear as a possible second choice. However, the momentary state at a point in the atmosphere generally deviates more or less strongly from the state given by any climatology. The model thermosphere initialized with climatological data will consequently undergo an adjustment process during some spin-up phase. With this uncertainty in mind, we decided to begin with a very simple approach, which is based on a mean vertical temperature profile (neglecting

---

[4]See https://www.ecmwf.int/en/forecasts/documentation-and-support (accessed October 2018).





horizontal and temporal variations) obtained from Bates (1959); Hedin (1983); Fleming et al. (1988). Later on, this could be improved by using a climatology generated by UA-ICON itself. In the following, we will shortly describe the method to blend the initial data below the mesopause region and the prescribed atmospheric state above the mesopause region which is derived from the climatological values.

We assume the data used to define the initial state of the model atmosphere of UA-ICON to originate from hydrostatic models (this is true for the IFS-data, as well as for the climatology employed here). The altitude assigned to such a data point is actually the geopotential height. This is no issue, if the shallow-atmosphere approximation is made and the geopotential height coincides with the geometric height employed in the dynamical core of ICON. However, in a deep-atmosphere model the geometric and geopotential heights differ. Therefore the data initialization of UA-ICON takes place on geopotential heights,

to and from which the geometric heights of the grid levels are transformed using

$$z_g = \frac{z}{1 + \frac{z}{a}}, \quad z = \frac{z_g}{1 - \frac{z_g}{a}}. \tag{22}$$

The geopotential height below which initial data, e.g., from the IFS model, are available, will be denoted $\mathfrak{z}_g$ in the following. Climatological temperatures are taken from Fleming et al. (1988), who offer tables with zonally averaged, monthly temperature values, denoted $T_F$, from mean sea level to a geopotential height of $120\,\mathrm{km}$ in $5\,\mathrm{km}$ intervals. For our current, simple approach

these data sets are averaged temporally and meridionally, to obtain a single vertical temperature profile. Temperature values within the $5\,\mathrm{km}$ intervals are computed by a cubic spline interpolation (e.g., Bronstein et al., 2001). Above $120\,\mathrm{km}$ an analytical temperature profile from Bates (1959) (see also Hedin, 1983) is used, which is formally identical to Eq. (7)

$$T_B(\lambda, \varphi, z_g) = T_\infty + [T_{120\mathrm{km}}(\lambda, \varphi) - T_\infty] \exp\left[-\frac{z_g - 120\,\mathrm{km}}{H_B(\lambda, \varphi)}\right], \tag{23}$$

where $T_{120\mathrm{km}} = T_F(z_g = 120\,\mathrm{km})$, and $T_\infty$ is approximately the temperature for the limit $z \to \infty$. This limit corresponds to

a geopotential height of $z_g = a$, which follows from Eq. (22) by multiplying the right-hand side of the first equation with $1 = (1/z)/(1/z)$ and applying the limit. The value of $T_\infty$ could be set, for instance, to the mean exospheric temperature of about $1035\,\mathrm{K}$. For a steady transition between $T_F$ and $T_B$, the scale height is set to $H_B = (T_\infty - T_{120\mathrm{km}})/(\mathrm{d}T_F/\mathrm{d}z_g)_{z_g=120\,\mathrm{km}}$. In addition, the temperature blending requires extrapolating the temperature data from below $\mathfrak{z}_g$ to above, which is done by a simple linear extrapolation

$$T_{\mathrm{IFS}}(\lambda, \varphi, z_g) = \mathfrak{T}_{\mathrm{IFS}}(\lambda, \varphi) + \gamma(\lambda, \varphi)(z_g - \mathfrak{z}_g), \tag{24}$$

with $\mathfrak{T}_{\mathrm{IFS}}(\lambda, \varphi) = T_{\mathrm{IFS}}(\lambda, \varphi, z_g = \mathfrak{z}_g)$, and $\gamma = \mathrm{d}T_{\mathrm{IFS}}/\mathrm{d}z_g|_{z_g=\mathfrak{z}_g}$. To obtain a statically stable stratification, $\gamma$ is limited by the dry adiabatic lapse rate: $\gamma \geq -\Gamma_d = -g/c_p$ (e.g., Holton, 2004). The blending reads

$$T(\lambda, \varphi, z_g) = T_{\mathrm{IFS}}(\lambda, \varphi, z_g)\alpha(z_g) + T_{\mathrm{clim}}(z_g)[1 - \alpha(z_g)], \tag{25}$$

$$\alpha(z_g) = \begin{cases} 1, & \text{for} \quad z_g < \mathfrak{z}_g \\ \frac{1}{2}\left[1 + \cos\left(\frac{z_g - \mathfrak{z}_g}{H_{\mathrm{blend}}}\pi\right)\right], & \text{for} \quad \mathfrak{z}_g \leq z_g \leq \mathfrak{z}_g + H_{\mathrm{blend}} \\ 0, & \text{for} \quad \mathfrak{z}_g + H_{\mathrm{blend}} < z_g \end{cases} \cdot \tag{26}$$





where $T_{\text{clim}}$ is $T_{\text{F}}$ or $T_{\text{B}}$, respectively, and $H_{\text{blend}}$ is a tunable blending scale height, which allows to control over what distance the transition between $T_{\text{IFS}}$ and $T_{\text{clim}}$ takes place. Currently we use a value of $H_{\text{blend}} = 10\,\text{km}$ to avoid negative absolute temperatures which could result from Eq. (24), although somewhat larger values for $H_{\text{blend}}$ do likely satisfy this as well. The blending factor $\alpha$ satisfies $\mathrm{d}\alpha/\mathrm{d}z_g|_{z_g = \mathfrak{z}_g,\,\mathfrak{z}_g + H_{\text{blend}}} = 0$, in order to guarantee a steady transition at $z_g = \mathfrak{z}_g$ and $z_g = \mathfrak{z}_g + H_{\text{blend}}$.

Given the temperature field (25), the other variables above $\mathfrak{z}_g$ are determined by the hydrostatic and geostrophic equilibrium. On the one hand this provides a relatively simple way for their computation, and on the other hand it helps to reduce the magnitude of the dynamic tendencies and therefore the strength of the adjustment process during the first time steps of the numerical integration. The pressure is computed from a numerical integration of the discretized form of $\partial p/\partial z_g + gp/(RT) = 0$, starting at $\mathfrak{z}_g$ with $p_{\text{IFS}}(\lambda, \varphi, z_g = \mathfrak{z}_g) = \mathfrak{p}_{\text{IFS}}$, where the deep-atmosphere-specific terms, underlined in Eq. (10), are neglected,

to simplify matters. Once temperature and pressure are known, $\rho$, $\pi$ and $\theta$ can be diagnosed. The horizontal wind is determined from a blending formally identical to the temperature blending (25) (using the same $\alpha$). The IFS part for the blending above $\mathfrak{z}_g$ is a simple linear extrapolation of the horizontal velocity, formally identical to Eq. (24). The "climatological" part shall satisfy $f_r \boldsymbol{e_r} \times \boldsymbol{v_{h,\text{clim}}} = -\beta(\varphi)(\boldsymbol{\nabla_h} p)/\rho$, where $p$ and $\rho$ are the hydrostatic pressure and density, respectively. Associated with the thermal wind balance (Zdunkowski and Bott, 2003; Holton, 2004), relatively strong horizontal temperature gradients

between $\mathfrak{z}_g$ and $\mathfrak{z}_g + H_{\text{blend}}$ can cause the magnitude of $\boldsymbol{v_{h,\text{clim}}}$ to increase with height and reach values which violate the CFL stability criterion (Zdunkowski and Bott, 2003; Holton, 2004). To avoid this $\boldsymbol{v_{h,\text{clim}}}$ is multiplied by a factor $[1 + (z_g - \mathfrak{z}_g)/H_{v_h}]\exp[-(z_g - \mathfrak{z}_g)/H_{v_h}]$, with a tuneable decay scale height $H_{v_h}$ (we use a value of $H_{v_h} = 10\,\text{km}$). The value of this factor and its vertical derivative is 1 and 0 at $z_g = \mathfrak{z}_g$, respectively, so as not to affect the continuity of the extrapolated wind at that altitude. Of course this factor causes $\boldsymbol{v_{h,\text{clim}}}$ to violate the geostrophic balance to some extent, but it turned out that

this is less severe for the spin-up phase than an initial wind field violating the CFL-criterion locally. In addition, the horizontal pressure gradient is multiplied by the factor

$$\beta(\varphi) = \begin{cases} 1, & \text{for} \quad \varphi_{\text{trop}} < |\varphi| \\ \frac{1}{2}\left[1 - \cos\left(\frac{|\varphi|}{\varphi_{\text{trop}}}\pi\right)\right], & \text{for} \quad |\varphi| \leq \varphi_{\text{trop}} \end{cases}, \tag{27}$$

in order to reduce its magnitude smoothly to zero towards the equator, where the geostrophic balance does not apply. $\varphi_{\text{trop}} > 0°$ is a tuneable tropical latitude (we use $\varphi_{\text{trop}} = 10°$). Finally, the vertical wind above $\mathfrak{z}_g$ is computed from a blending, again

formally identical to Eq. (25), of a linearly extrapolated $w_{\text{IFS}}$ and a $w_{\text{clim}} = 0$, in accordance with the hydrostatic balance and the boundary condition of a vanishing vertical wind at the model top.

## 2.2 Upper-atmosphere physics

A new physics package, which parameterizes processes specific to the upper atmosphere has been developed for UA-ICON. This package, referred to as the UA package, can be called in combination with either the NWP package or the ECHAM

package. The processes taken into consideration in the UA package are summarized in Table 1.

The processes are categorized into 3 groups, kinetics, radiation and chemical heating, as described below. Most of the parameterizations are adopted from the HAMMONIA model, a spectral model based on ECHAM5 (Roeckner et al., 2006), covering





**Table 1.** Physical parameterizations implemented in the UA package of UA-ICON shown with references.

| Process | Reference |
| --- | --- |
| molecular diffusion | Huang et al. (1998); Banks and Kockarts (1973) |
| frictional heating | Gill (1982) |
| ion drag and Joule heating | Hong and Lindzen (1976) |
| gravity wave turbulent mixing | Hines (1997a, b) |
| ultraviolet: Schumann-Runge bands and continuum ($O_2$) | Strobel (1978) |
| extreme ultraviolet ($N_2$, O, $O_2$) | Richards et al. (1994) |
| non-LTE infrared cooling ($CO_2$, NO, $O_3$) | Fomichev and Blanchet (1995, 1998); Ogibalov and Fomichev (2003) |
| infrared cooling at $5.3\,\mu m$ (NO) | Kockarts (1980) |
| chemical heating | climatology from HAMMONIA |

the atmosphere up to the thermosphere ($1.7 \times 10^{-7}\,\mathrm{hPa}$, $\sim 250\,\mathrm{km}$). A detailed description of the physics parameterizations used in HAMMONIA can be found in Schmidt et al. (2006), thus here we keep the description brief, only noting important and differing treatments. An overall difference is that in UA-ICON all the parameterizations are implemented such that the computation only starts at a certain altitude above which the forcings are expected to become relevant. This increases computational
efficiency significantly.

### 2.2.1 Kinetics

Above the mesopause, molecular diffusion, which is negligible at lower altitude, becomes significant. In fact, there the downward transport of heat by molecular diffusion appears as a strong cooling in balance with the strong solar heating. Hence, it is of primary importance to parameterize molecular processes in this region of the upper atmosphere. Molecular transport of heat,
momentum and tracers are parameterized in the UA package following Huang et al. (1998) and Banks and Kockarts (1973), as in HAMMONIA. Besides direct transport of heat by molecular diffusion, the momentum transport also leads to energy deposition in the form of heat, known as frictional heating. In the UA package this is parameterized following Gill (1982). The computation starts at $75\,\mathrm{km}$.

In the mesosphere and lower thermosphere, unlike in lower layers, a larger number of air particles are ionized and thus
aligned with the electromagnetic field of the Earth. This produces a force on the neutral mass flow, its tangential and normal components known as the ion drag and the Lorenz force, respectively. Joule heating is produced by the ion drag as well. In UA-ICON, as in HAMMONIA, this effect is parameterized following the simple Hong and Lindzen (1976) approach. The computation starts at $80\,\mathrm{km}$.

A large portion of gravity wave momentum energy is deposited and transferred to turbulent energy near the mesopause,
where turbulence is otherwise very weak. The turbulent mixing effects induced by gravity waves can be estimated using the Hines (1997a, b) parameterization included in the ECHAM package, but it is switched off in standard ICON. In UA-ICON we





enable the calculation and pass the computed turbulent diffusion coefficient to the turbulent mixing subroutine to account for gravity wave-induced turbulent mixing.

### 2.2.2 Radiation

In standard ICON, the PSrad radiation package (Pincus and Stevens, 2013) is employed, which is itself an extension to the RRTMG model (Mlawer et al., 1997; Iacono et al., 2008). The shortwave component of the radiation package covers wavelengths of the solar spectrum longer than 200 nm. This is a sufficient bandwidth only up to about the mesopause, beyond which radiative heating in the ultraviolet and extreme ultraviolet frequencies become dominant. In the UA package, starting from 60 km, ultraviolet solar forcing for the $O_2$ Schumann-Runge bands (SRB; 175 nm to 205 nm) and continuum (SRC; 125 nm to 175 nm) is calculated based on the model of Strobel (1978). Efficiency factors multiplied to the SRBC heating rates account for the loss of internal energy due to airglow processes and are taken from Mlynczak and Solomon (1993). For the extreme ultraviolet (EUV; 5 nm to 105 nm) solar forcing, starting above 90 km, a model based on Richards et al. (1994) taken from HAMMONIA is currently used. Efficiency factors multiplied to the EUV heating rates are based on Roble (1995). Their values also account for the energy loss due to radiative cooling in the 5.3 µm NO band. Since this process is explicitly calculated in our model (see below), a factor of 1.33 is multiplied to these efficiency factors (see Richards et al., 1982). Some additional adjustments to the PSrad/RRTMG shortwave radiation are necessary in UA-ICON due to the introduction of chemical heating. More details on this are given in section 2.2.3.

The longwave component of PSrad/RRTMG covers terrestrial wavelengths shorter than 1 mm. This bandwidth is still valid at thermospheric heights, yet a few important additions have to be made:

1. The usual assumption of local-thermodynamical-equilibrium (LTE) does not hold above the mesopause, thus non-LTE effects must be taken into account. As in HAMMONIA, non-LTE infrared cooling by $O_3$ and $CO_2$ is calculated from the parameterization of Fomichev and Blanchet (1995) with the modifications of Fomichev and Blanchet (1998). The calculation starts at 65 km, and the calculated values are multiplied by a scaling factor $\alpha$ equaling 0 at 65 km and linearly growing to 1 at 75 km. Correspondingly, the longwave radiation computed by PSrad/RRTMG is scaled with the factor $1 - \alpha$, effectively discarding it above 75 km.

2. As in HAMMONIA, a parameterization of $CO_2$ non-LTE absorption in the near infrared following Ogibalov and Fomichev (2003) is employed. The computed values are ignored below 19.25 km and fully considered above 24.5 km.

3. NO cooling at 5.3 µm is calculated utilizing the parameterization from Kockarts (1980). The computation starts at 60 km.

A noteworthy difference between ICON and HAMMONIA is that ICON is a non-hydrostatic model on height levels, whereas HAMMONIA is hydrostatic on hybrid pressure levels. In HAMMONIA, for the use in the radiation computation, number densities of radiatively active tracers are calculated based on the mass of air in a given layer which is derived from pressure differences between the upper and lower surfaces of a layer. This approach is only valid under the assumption of hydrostatic balance, since otherwise pressure is not guaranteed to decrease strictly monotonically with increasing altitude. Therefore, in





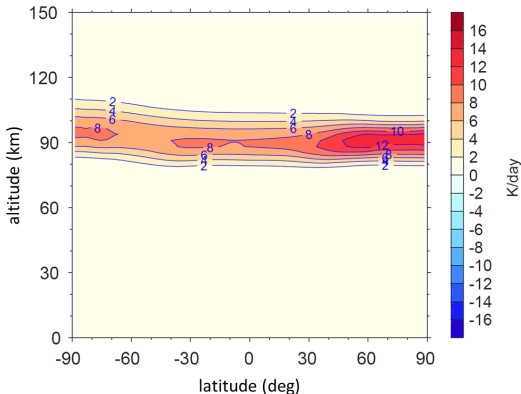

**Figure 2.** Zonal mean chemical heating rates (K/day) averaged for the month of January from HAMMONIA simulations. Such monthly and zonal means are prescribed in UA-ICON.

the UA package the computation of number density for the radiation parameterization is utilizing the mass of air, which is a globally conserved quantity in ICON.

Moreover, HAMMONIA has the upper boundary of its top pressure level at $0\,\mathrm{hPa}$, effectively covering the whole atmosphere. The height levels of ICON, on the other hand, cover a finite range and unavoidably ignore the atmospheric air mass above the model lid. The effect of this missing amount of air on radiative fluxes is ignored in UA-ICON.

### 2.2.3 Chemical heating

Chemical heating, i.e., the release of heat due to recombination reactions between atoms or radicals produced as a result of photolysis, becomes important in maintaining the upper atmospheric thermodynamic balance, where the photolysis products can travel a long distance before recombining. HAMMONIA employs a condensed version of the MOZART3 chemistry model (Kinnison et al., 2007) to explicitly compute chemical heating online. In our case, however, given the finer target resolution (from $160\,\mathrm{km}$ down to a few tens of kilometers) and the central goal of studying gravity waves, using a coupled chemistry model is overly expensive. Therefore, we deploy a simpler strategy of prescribing monthly zonal-mean climatological chemical heating rates from a 35-year HAMMONIA simulation with constant present-day boundary conditions. As an example, the chemical heating rates for January are shown in Fig. 2. Technically, below $70\,\mathrm{km}$ all heating is calculated in the PSrad/RRTMG shortwave radiation code and no chemical heating rates are prescribed, whereas above $80\,\mathrm{km}$ full chemical heating rates are used, but the efficiency factor of the shortwave radiation is reduced to 0.23. Between $70\,\mathrm{km}$ and $80\,\mathrm{km}$ the two heating sources are linearly merged.

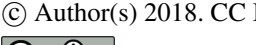



## 3 Model evaluation

### 3.1 Idealized test cases

To test the deep-atmosphere implementation in the dynamical core, we used two test cases. In the first test case the propagation of a sound wave is considered, for which an analytical solution (of the linearized equations) is available. It is aimed at testing especially the accuracy of the spherical geometry in its imprint on the grid cells and the corresponding modification factors described in section 2.1.2, and at testing the metric terms and the complete Coriolis acceleration in the components of the momentum equation (9) and (10). The second test case is the Jablonowski-Williamson baroclinic instability test case (Jablonowski and Williamson, 2006) in its extension for deep-atmosphere dynamical cores by Ullrich et al. (2014). It reveals if the height dependence of gravity is properly implemented and maintains the hydrostatic background state of the test case atmosphere (especially, if gravity enters the momentum equation only implicitly, as in (10)). In addition, the performance of the entire dynamical core is tested, when it comes to reproducing the development of the baroclinic wave. Both test cases make use of the small-Earth approach of Wedi and Smolarkiewicz (2009) to pronounce deep-atmosphere effects.

### 3.1.1 Sound wave test case

The particular motivation for this first test case is that an analytical solution is available to which the numerical solution can be compared. We have developed this test case with a method originally proposed by Läuter et al. (2005) for the shallow-water equations on the sphere. The method was developed further for the shallow- and deep-atmosphere equations e.g., by Staniforth and White (2008); Baldauf et al. (2014). An atmosphere at rest in the absolute frame is considered. If a non-trivial, analytical solution is known for this case, it can be transformed into a rotating frame (e.g., regarded as a rotating Earth slipping through the air without exchange of tangential momentum). Depending on the solution in the absolute frame being either stationary or time-dependent, potentially all aspects of a dynamical core can be tested. However, a disadvantage of this method is that the centrifugal acceleration has to be taken into account explicitly in the dynamical core (see Eq. (1)). Some aspects of this transformation method are shown in appendix A1, and a thorough mathematical description can be found in the literature cited above.

Given a solution in the absolute frame, it appears to be advected with $\boldsymbol{v} = -\boldsymbol{v_F} = -\boldsymbol{\Omega} \times (\boldsymbol{X} - \boldsymbol{A})$ from the perspective of the rotating frame (with the center of Earth $\boldsymbol{A}$ and an arbitrary point $\boldsymbol{X}$ not coincident with $\boldsymbol{A}$). In practice this means that the solution has merely to be rotated by an angle $-\Omega t$ about an axis being parallel to $\boldsymbol{\Omega}$ and crossing $\boldsymbol{A}$. Therefore, we will direct our attention to the solution to be rotated, in the following.

Baldauf et al. (2014) derived analytical gravity and sound wave solutions for the linearized deep-atmosphere equations. However, certain terms of the equations had to be omitted in order to allow the solution to be expanded in a system of orthonormal basis functions. Although these terms were shown to be only of second order, their effects are not controllable in a dynamical core, where the omission of most of the corresponding discretized terms is unfeasible without greater effort. Therefore, it would be desirable to find solutions for the linearized equations with the omitted terms restored, or alternatively with only those omissions retained, which can be realized in a dynamical core without greater effort. The fact that the gravity $-g(a/r)^2(\boldsymbol{X} - \boldsymbol{A})/r$





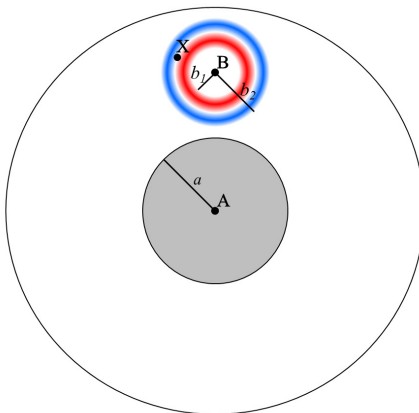

**Figure 3.** Schematic illustration of the initial state of the spherical sound wave (where the blue and red color shading indicates the positive and negative values of the pressure perturbation associated with the sound wave). The small-Earth (without topography) is depicted in gray, and the black circles represent the model bottom and top.

imprints a spherical symmetry on the equations with its distinct point $\boldsymbol{A}$ turns out to be a severe obstacle to that aim. So we decided to switch off gravity (which in model practice means to set the constant parameter $g$ to a very small value $> 0$, since divisions by $g$ are used throughout the model, especially in the physics parameterizations). This greatly simplifies the problem, but has the consequence that the test case makes no statement about the implementation of gravity. Under these circumstances

the atmosphere, enclosed between the spherical boundaries of the model bottom and top, is isotropic, with the constant pressure $p_0$ and temperature $T_0$. As long as the sound waves, propagating with the speed of sound $c_s = \sqrt{(c_p/c_v)RT_0}$, do not interact with the boundaries, they are not "aware" of the spherical shape of the atmosphere as a whole. Therefore the challenge for the model is to properly simulate the sound wave propagation on the anisotropic spherically curved grid. For this test case we consider a spherically symmetric acoustic wave, shown schematically in Fig. 3, which consists of an outward propagating part,

only. The derivation is shown in appendix A2, and the solution for the pressure perturbation $p' = (c_p/R)(p_0/\pi_0)\pi'$ associated with the sound wave reads

$$
\begin{aligned}
p'(x,t) = \delta p \Bigg\{ & \frac{x - c_s t}{x} \sin\left(\pi \frac{x - b_1 - c_s t}{b_2 - b_1}\right) \sin\left(2\pi n \frac{x - b_1 - c_s t}{b_2 - b_1}\right) \\
& + \frac{b_2 - b_1}{x} \left[ \frac{\sin\left(\pi(2n-1)\frac{x-b_1-c_s t}{b_2-b_1}\right)}{2\pi(2n-1)} - \frac{\sin\left(\pi(2n+1)\frac{x-b_1-c_s t}{b_2-b_1}\right)}{2\pi(2n+1)} \right] \Bigg\} \left[\Theta(x - b_1 - c_s t) - \Theta(x - b_2 - c_s t)\right], \quad (28)
\end{aligned}
$$

where $\delta p = (c_p/R)(\delta T/T_0)p_0$ denotes the pressure amplitude of the wave, determined by a temperature amplitude $\delta T$ in our

implementation, $x = |\boldsymbol{X} - \boldsymbol{B}|$ is the distance from the center of the spherical wave $\boldsymbol{B}$, $x = b_1 > 0$, and $x = b_2 > b_1$ are the radial boundaries within which the wave has a non-vanishing amplitude at $t = 0$ (see Fig. 3), and $n$ is the number of wave crests (in this work we consider only $n = 1$). In addition, $\Theta$ denotes the Heaviside step function defined as (e.g., Bronstein





$$\Theta(\xi) = \begin{cases} 0, & \text{for} \quad \xi < 0 \\ 1, & \text{for} \quad \xi \geq 0 \end{cases}.$$ (29)

The solution (28) is only valid until the first reflection occurs. Of course, sound waves have been thoroughly investigated in the literature and solutions to the sound wave equation are all but new (e.g., Kirchhoff, 1876), but since their propagation in

combination with the method of Läuter et al. (2005) involves potentially all parts of a dynamical core (except for gravity), we found this test useful, also in view of the relative scarcity of test cases dedicated to deep-atmosphere dynamical cores in the literature. In order to highlight the effect of the spherical curvature (on the model grid) the radius of Earth can be rescaled $a \to \eta_1 a, (\eta_1 < 1)$. This is the small-Earth approach proposed by Wedi and Smolarkiewicz (2009) (see also Baldauf et al., 2014; Ullrich et al., 2014). The model time step is rescaled accordingly in order to account for the correspondingly smaller mesh size

of the horizontal grid. Furthermore it might be advantageous to rescale the angular velocity of the Earth $\Omega \to \eta_2 \Omega$, in order to control the velocity $\boldsymbol{v} = -\boldsymbol{v_F}$ with which the sound wave is advected. Further details on the implementation can be found in appendix A2. Apart from that, we followed closely the guidelines given by Baldauf et al. (2014) for the implementation into UA-ICON.

We envisaged two concrete test configurations: The first without rotation ($\eta_2 = 0$), to simulate the sound wave propagation

in the absolute frame, and the second with rotation ($\eta_2 > 0$), to test if the dynamical core is able to maintain the balance of the background velocity in the rotating frame $\mathrm{d}\boldsymbol{v_F}/\mathrm{d}t + 2\boldsymbol{\Omega} \times \boldsymbol{v_F} + \boldsymbol{\Omega} \times [\boldsymbol{\Omega} \times (\boldsymbol{X} - \boldsymbol{A})] = 0$, so as to advect the sound wave in a shape-conserving way. Further parameter settings are listed in Table 2. The temperature amplitude of the sound wave $\delta T$ was chosen small enough that its non-linear dynamics as computed by the dynamical core is negligibly small (an initial amplitude of $\delta T = 0.1\,\mathrm{K}$ appears large when compared, e.g., to typical values used in Baldauf et al. (2014), but even for larger

amplitudes, $\sim 1\,\mathrm{K}$, the numerical solution of UA-ICON was in relatively good agreement with the analytical solution of the linearized equations).

A height-longitude cross section at the equator of the numerical solution from UA-ICON, the analytical solution, as well as the difference between the two for both configurations are shown in Fig. 4, shortly before the periphery of the sound wave would impinge on the bottom and top boundaries. The angular velocity of the second configuration was chosen such that

the center of the sound wave, $\boldsymbol{B}$, would be advected by the zonal background wind $u(\boldsymbol{B}) = -u_F(\boldsymbol{B}) = -\Omega r \cos(\varphi)|_{\boldsymbol{B}} = -100\,\mathrm{m\,s^{-1}}$ in the rotating frame. This is close to the value used by Baldauf et al. (2014) in their test scenario (C). A time step of $\Delta t = \eta_1 \cdot 13.2\,\mathrm{s}$ fits both, the maximum magnitude of the propagation velocity $|v|_{\max} = c_s = 317\,\mathrm{m\,s^{-1}}$ in the first configuration, and $|v|_{\max} = |u_F|_{\max} + c_s = 134\,\mathrm{m\,s^{-1}} + 317\,\mathrm{m\,s^{-1}} = 451\,\mathrm{m\,s^{-1}}$ in the second configuration.

Shape and amplitude are relatively well captured by the numerical solution in both configurations, with the difference in

amplitude to the analytical solution being about one order of magnitude smaller than the magnitude of the wave's pressure perturbation itself. However, in the second configuration, where the sound wave is advected westward while radially propagating, the magnitude of the error has increased slightly, and the symmetry of the pressure difference with respect to a vertical axis crossing the center of the sound wave is lost due to the horizontal advection. The amount by which symmetry is lost is





**Table 2.** Parameters used for the sound wave test case with UA-ICON. (Where necessary, a comma separates different values for the two configurations differing in their angular velocity $\Omega$.)

| | |
|---|---|
| Temperature of atmospheric background state $T_0$ | $250\,\mathrm{K}$ |
| Pressure of atmospheric background state $p_0$ | $1000\,\mathrm{hPa}$ |
| Temperature amplitude of sound wave $\delta T$ | $0.1\,\mathrm{K}$ |
| Initial radial boundaries of sound wave $(b_1; b_2)$ | $(2000\,\mathrm{m}; 30000\,\mathrm{m})$ |
| Location of sound wave center $\boldsymbol{B} \to (\lambda; \varphi; z)$ | $(180°; 0°; 50000\,\mathrm{m})$ |
| Number of sound wave crests $n$ | 1 |
| Rescale factor for radius of Earth $\eta_1$ | 1/66 |
| Rescale factor for angular velocity of Earth $\eta_2$ | 0, 9.38 |
| Height of model top $H_{\mathrm{top}}$ | $100000\,\mathrm{m}$ |
| Horizontal ICON grid R$n$B$k$ | R2B6 (mean horizontal mesh size: $\Delta\varphi = 0.355°$) |
| Constant vertical grid layer thickness $\Delta z$ | $555.6\,\mathrm{m}$ |
| Time step $\Delta t$ | $\eta_1 \cdot 13.2\,\mathrm{s}$ |
| Gravitational acceleration $g$ | $10^{-30}\,\mathrm{m\,s^{-2}}$ |

a measure for the phase error of the horizontal advection implementation (e.g., Skamarock and Klemp, 2008). The pressure difference in the first configuration not being radially symmetric with respect to the center of the sound wave is probably due to at least three anisotropies between the vertical and the horizontal: first, the horizontal and vertical mesh sizes $\Delta x = 597.8\,\mathrm{m}$ and $\Delta z = 555.6\,\mathrm{m}$ slightly differ. Second, the extension of a grid cell increases with height in the spherical geometry, and third,

the horizontally explicit-vertically implicit scheme employed in the dynamical core of ICON introduces an anisotropy as well.

We repeated the simulation for different grid resolutions and computed the $L^2$-norm and $L^\infty$-norm of the pressure difference between the numerical and analytical solutions on the entire circum equatorial height-longitude cross section, of which a part is plotted in Fig. 4, and at time $t = 60\,\mathrm{s}$, according or in analogy to the formula employed by Baldauf and Brdar (2014, p. 1983). All pressure values entering the computation of the two norms are weighted equally, i.e., no weighting with the cell

volume is applied (in which case the two norms would not be with respect to the pressure difference $\Delta p$, but with respect to a work-like quantity $\propto \Delta pV$). The results and some further information on the employed grids are shown in Fig. 5. In the first configuration, without rotation, the convergence rate is dominated by a second-order behaviour, although a relatively small first-order component seems to be present, especially in case of the $L^\infty$-norm. In the second configuration, with rotation, the convergence rate seems to start with a second-order behaviour for the lower grid resolutions, and changes into a first-order

behaviour for the higher resolutions. This is in agreement with the findings of Baldauf et al. (2014) for their test scenario (C) (compare their Fig. 7). The reason for the first-order convergence in the presence of a background wind is still unknown. Nevertheless, we regard the agreement between the analytical and numerical solutions in both configurations as satisfactory,

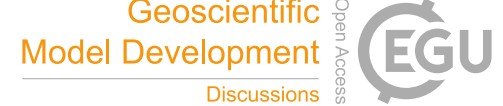



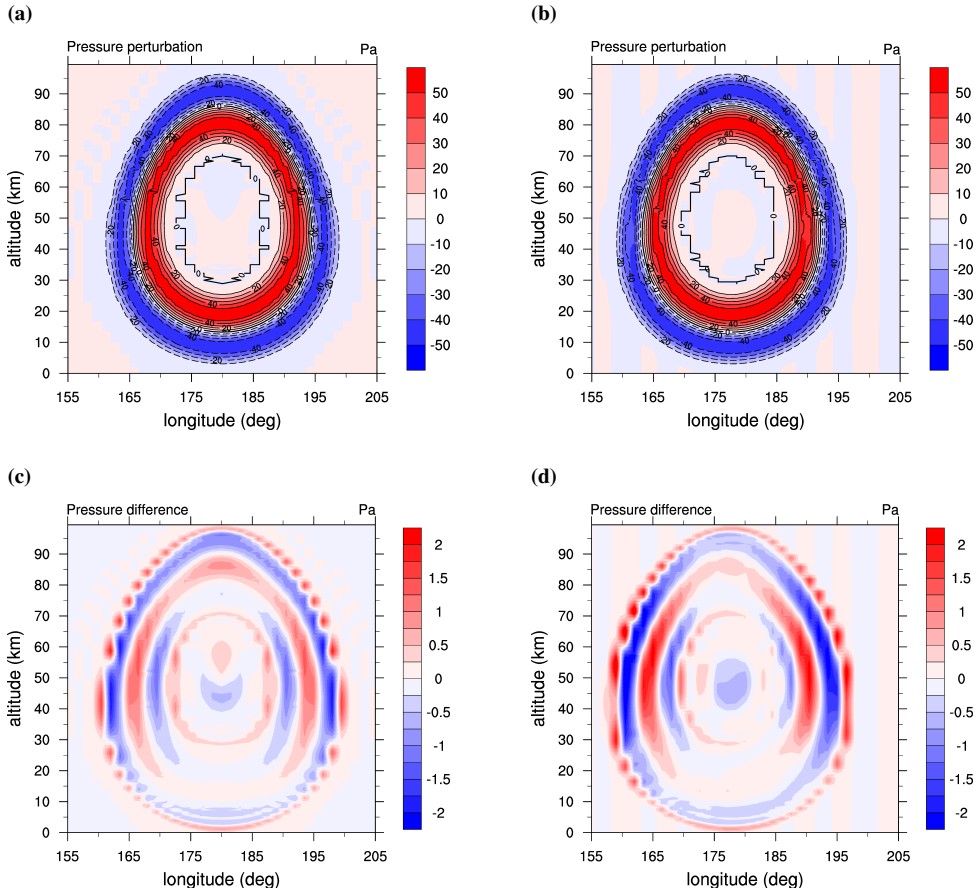

**Figure 4.** Pressure perturbation associated with the spherical sound wave for a height-longitude cross section at the equator, and at time $t = 60\,\mathrm{s}$. Upper row: the numerical solution from a simulation with UA-ICON on a R2B6L180-grid is in color (where L denotes the number of vertical grid layers), isolines depict the analytical solution. The parameters of the sound wave are listed in Table 2. Left: configuration without rotation ($\Omega = 0$). Right: configuration with rotation ($\Omega = \eta_2 \cdot 7.29 \times 10^{-5}\,\mathrm{rad\,s^{-1}} = 6.84 \times 10^{-4}\,\mathrm{rad\,s^{-1}}$). Lower row: pressure difference from subtracting the numerical from the analytical solution for the two respective configurations in the upper row. (The longitude counts positive in eastward direction, in addition the sound wave appearing to have an oval shape is due to the compression of the circular sector into the rectangular plot.

as we assume a critical deficiency in the deep-atmosphere modification of the dynamical core to leave a much more distinct fingerprint in the numerical solution.

### 3.1.2 Jablonowski-Williamson baroclinic instability test case

The previous test case focused on one particular emergent structure of the atmosphere. However, if we turn our focus to the
5  atmospheric features on the synoptic scale, other structures, such as baroclinic waves, are much more important than sound





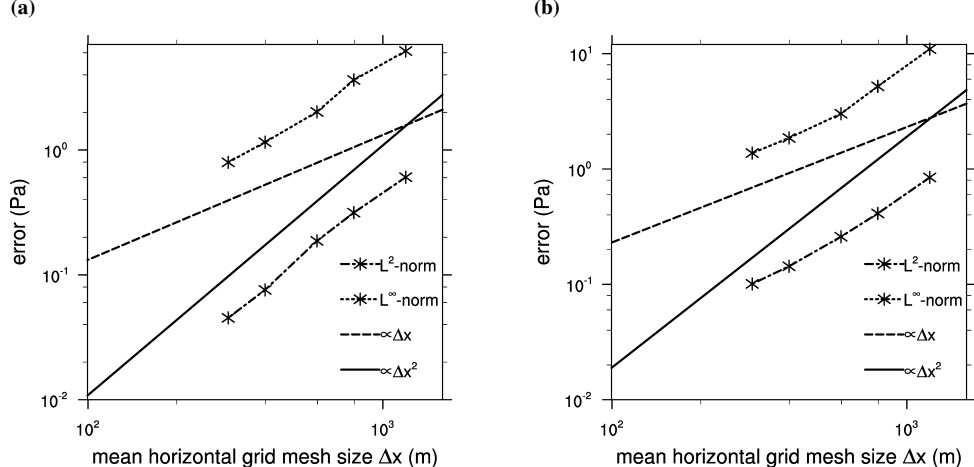

**Figure 5.** The $L^2$-norm and $L^\infty$-norm of the difference between the numerical solution and the analytical solution of the pressure field at the height-longitude cross section at the equator, and at time $t = 60\,\text{s}$. Left: configuration without rotation ($\Omega = 0$). Right: configuration with rotation ($\Omega = \eta_2 \cdot 7.29 \times 10^{-5}\,\text{rad s}^{-1} = 6.84 \times 10^{-4}\,\text{rad s}^{-1}$). The norms are plotted for the grids: R2B5L90 ($\Delta x = \eta_1 \cdot 78.9\,\text{km}$, $\Delta z = 1111.1\,\text{m}$, $\Delta t = \eta_1 \cdot 26.4\,\text{s}$), R3B5L136 ($\Delta x = \eta_1 \cdot 52.6\,\text{km}$, $\Delta z = 735.3\,\text{m}$, $\Delta t = \eta_1 \cdot 17.6\,\text{s}$), R2B6L180 ($\Delta x = \eta_1 \cdot 39.5\,\text{km}$, $\Delta z = 555.6\,\text{m}$, $\Delta t = \eta_1 \cdot 13.2\,\text{s}$), R3B6L277 ($\Delta x = \eta_1 \cdot 26.3\,\text{km}$, $\Delta z = 361.0\,\text{m}$, $\Delta t = \eta_1 \cdot 8.8\,\text{s}$), and R2B7L360 ($\Delta x = \eta_1 \cdot 19.7\,\text{km}$, $\Delta z = 277.8\,\text{m}$, $\Delta t = \eta_1 \cdot 6.6\,\text{s}$), where L denotes the number of vertical grid layers, $\Delta x$, $\Delta z$ and $\Delta t$ are the mean horizontal mesh size, the grid layer thickness, and the time step, respectively. The dashed and solid lines indicate $\mathcal{O}(\Delta x)$ and $\mathcal{O}(\Delta x^2)$ behaviour, respectively.

waves. The Jablonowski-Williamson baroclinic instability test case (Jablonowski and Williamson, 2006) is a standard test to investigate the performance of atmospheric models in representing a key feature of midlatitude dynamics. It consists of a baroclinically unstable atmosphere in hydrostatic and geostrophic balance to which a perturbation is added which triggers the instability. This test case reveals on the one hand, if the model is able to maintain the hydrostatically and geostrophically

balanced background state during the first days of the wave evolution, when its amplitude is still relatively small, and on the other hand, how the model performs in reproducing the amplitude growth of the wave and its shape. However, a disadvantage of this test is that no analytical solution for the problem is known, so that the evaluation has to be based on a model intercomparison. Ullrich et al. (2014) have extended this test case for deep-atmosphere dynamical cores and introduced some further improvements to the original formulation. The approach of the small-Earth is employed to highlight the differences

between the shallow- and the deep-atmosphere dynamics. The rescale factors are $\eta_1 = 1/20$ and $\eta_2 = 20$, for the earth radius and angular velocity, respectively. For the test with UA-ICON we used a R3B4-grid which provides a horizontal mesh size of $\Delta \varphi = 0.95°$. This is close to the value used for the production of the numerical benchmark solution in Ullrich et al. (2014). The vertical grid is streched, with layer thicknesses increasing from the model bottom to the model top at $30\,\text{km}$. Following Ullrich et al. (2014), we use 30 levels, however, the vertical stretching of the ICON grid differs from their formula (28). The

results for day 8 and 10 of the simulation with UA-ICON are shown in Fig. 6 (upper row). In order to study the convergence



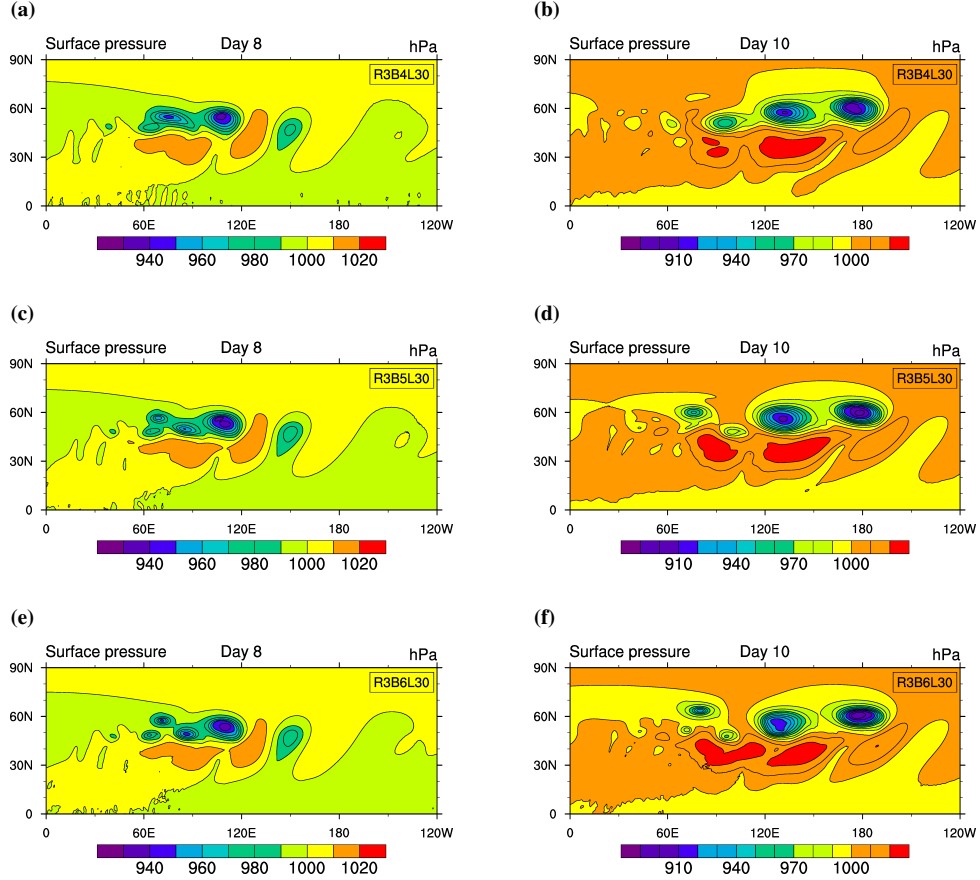

**Figure 6.** Surface pressure (in hPa) at days 8 (left) and 10 (right) from UA-ICON simulations of the Jablonowski-Williamson baroclinic instability test for deep-atmosphere dynamical cores. Results from three different horizontal grid resolutions are shown: R3B4 ($\Delta\varphi = 0.95°$, $\Delta t = \eta_1 \cdot 96\,\mathrm{s}$), R3B5 ($\Delta\varphi = 0.48°$, $\Delta t = \eta_1 \cdot 48\,\mathrm{s}$) and R3B6 ($\Delta\varphi = 0.24°$, $\Delta t = \eta_1 \cdot 24\,\mathrm{s}$), where $\Delta\varphi$ denotes the mean horizontal mesh size and $\Delta t$ is the dynamical time step. The vertical resolution is the same for all simulations: 30 levels (L30) up to a model top at $30\,\mathrm{km}$. The grid is vertically stretched, from a thickness of $\Delta z_{\mathrm{min}} = 100\,\mathrm{m}$ for the lowermost level up to $\Delta z_{\mathrm{max}} = 1969\,\mathrm{m}$ for the uppermost level.

behavior, we doubled the horizontal grid resolution twice (see Fig. 6, the middle and lower rows), and did likewise for the vertical resolution in case of the horizontal grid R3B5 (see Fig. 7).

First of all we can state that UA-ICON is able to maintain the hydrostatic and geostrophic balances of the background state in the first days of the simulation relatively well. This indicates, for instance, that the vertical variation of the gravitational acceleration is adequately implemented. Second the amplitude and shape of the baroclinic wave, as they show up in the surface pressure in Fig. 6, compare relatively well to the benchmark solution of Ullrich et al. (2014, their Fig. 9), and also to the solution of Wood et al. (2014, their Figs. 4 and 5). However, some differences can be recognized, especially in the tail of the baroclinic wave. The convergence tests revealed that the numerical solution is largely converged with regard to the vertical





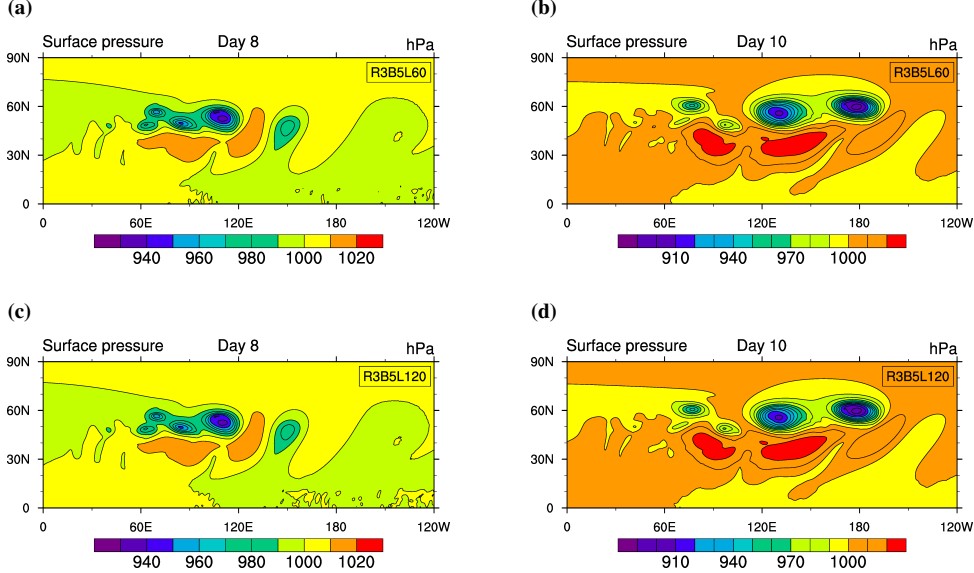

**Figure 7.** As in Fig. 6, but here for different vertical resolutions: 60 levels (L60, $\Delta z_{\min} = 50\,\mathrm{m}$, $\Delta z_{\max} = 937\,\mathrm{m}$) and 120 levels (L120, $\Delta z_{\min} = 25\,\mathrm{m}$, $\Delta z_{\max} = 450\,\mathrm{m}$). The horizontal resolution is the same for all simulations: R3B5 ($\Delta \varphi = 0.48°$, $\Delta t = \eta_1 \cdot 48\,\mathrm{s}$).

resolution (Fig. 7). This is true for the horizontal resolution as well in the zonal range from 120E to 240E (120W), say. As mentioned before, the tail of the baroclinic wave in the zonal range from about 60E to 120E shows a greater variation between the different horizontal resolutions (Fig. 6). To see if the resolution R3B6 is converged in that regard, we tested the R3B7-grid as well. However, it developed a numerical instability in the tail region of the baroclinic wave around day 8. The reason for the
instability is not clear yet, but two elements may contribute to it.

    First, the non-traditional part of the Coriolis acceleration and the metric terms in the expansion of the velocity advection in a spherical coordinate system (see Eqs. (9) and (10)) may play a role in the development of the instability, as tests, in which they were switched off, indicate. In the following, we limit the consideration to the Coriolis acceleration as a representative. We consider Gibbs equation for an infinitesimal volume of dry air, fixed in shape and position in the observer frame (compare
Falk, 1968, 1990; Straub, 1996)

$$\frac{\partial e}{\partial t} = \boldsymbol{v} \cdot \frac{\partial \boldsymbol{p}}{\partial t} + (\mu + \phi_g)\frac{\partial \rho}{\partial t} + T\frac{\partial s}{\partial t}, \tag{30}$$

where $e$, $\boldsymbol{p}$ and $s$ are the energy, momentum and heat densities, respectively, and $\mu$ denotes the "chemical" potential. The Coriolis acceleration $\partial \boldsymbol{p}/\partial t = -2\boldsymbol{\Omega} \times \boldsymbol{p} + \cdots$ conserves the absolute value of the momentum $|\boldsymbol{p}|$ and is energetically neutral, since it does not contribute to the first term on the right-hand side of Eq. (30), if the equation of state of mechanics $\boldsymbol{v} = \boldsymbol{p}/\rho$ is
applied. In ICON, $\rho$ and $s$ are contained by the grid cells. The components of the momentum, or rather its intensive counterpart, the velocity $\boldsymbol{v}$ are defined on the cell interfaces. We may translate this C-grid staggering into the following Gibbs equation





(energy budget) for the overall system "model atmosphere"

$$\sum_j \delta E_j + \sum_i \delta E_i = \sum_j \boldsymbol{v_j} \cdot \delta \boldsymbol{P_j} + \sum_i \left[ (\mu_i + \phi_{g,i}) \delta m_i + T_i \delta S_i \right], \tag{31}$$

where $E$, $\boldsymbol{P}$, $m$ and $S$ denote the amounts of energy, momentum, mass and heat contained by the respective subsystem "cell" (index $i$), or "interface region" (index $j$). In addition, $\delta$ denotes the difference between two successive states, and a quantity $\delta X$,

with $X = E, \boldsymbol{P}, \ldots$ will be called a "flux" in the following. That is, we ragard the $\boldsymbol{P_j}$ as complete vectorial degrees of freedom, despite typically referring to them as "components" more or less intentially.[5] That the volumes of the cells and the interface regions overlap, is irrelevant here, since the flux of energy $\delta E_j$ is solely associated with the flux of momentum $\delta \boldsymbol{P_j}$, but not associated with the flux of extensive variables contained by the cells. Likewise the energy flux $\delta E_i$ is only associated with the fluxes of the extensive variables contained by the cells ($\delta m_i$ and $\delta S_i$), but not with the flux of momentum, which is contained by

the interface regions, so there is no double counting. If the momentum vector $\boldsymbol{P_j}$, contained by the interface region, could point into an arbitrary direction, the formulation of a Coriolis acceleration that changes the direction of $\boldsymbol{P_j}$, but leaves its absolute value unchanged, is possible. However, this does not hold in case of the C-grid staggering of the velocity. In terms of Eq. (31), with cells and interface regions as two distinct subsystems, it translates into the definition that the interface regions are capable of containing only momentum that is perpendicular to the interface (maybe in an averaged sense, if the interface is not a plane).

This fixed direction of $\boldsymbol{P_j}$ (and $\boldsymbol{v_j}$) makes it impossible, to satisfy $\boldsymbol{v_j} \cdot (2\boldsymbol{\Omega} \times \boldsymbol{P_j}) = 0$. Changing from local to global energy conservation, i.e. $\sum_j \boldsymbol{v_j} \cdot (2\boldsymbol{\Omega} \times \boldsymbol{P_j}) = 0$, may be a possible way out. However, we could not follow this alternative route in our implementation of the non-traditional part of the Coriolis acceleration, since it turned out to be very difficult on the triangular grid, if possible at all. So our current implementation conserves energy neither locally nor globally, at least formally. This may affect numerical stability negatively. Especially, since the magnitude of the terms of the non-traditional Coriolis acceleration,

such as $-w f_{\mathrm{t}}$ on the left-hand side of Eq. (9) is increased significantly through the small-Earth approach. Not only due to the factor $\eta_2 = 20$ in $f_{\mathrm{t}} = 2\eta_2 \Omega \cos(\varphi) \boldsymbol{e_\varphi} \cdot \boldsymbol{e_t}$, but also since the magnitude of the vertical wind in the baroclinic wave about day 8 is at least one order of magnitude larger than in case of the standard Earth ($\eta_2 = 1$, not shown). In addition, the magnitude of the vertical wind increases with increasing horizontal resolution, e.g., roughly by a factor of 2 from R3B4 to R3B6 (not shown).

A second element that may contribute to the observed numerical instability, is the apparent presence of a natural instability in the deep-atmosphere test case that seems to be absent in the shallow-atmosphere version. This instability develops in the vicinity of the equator on a shorter time scale than the baroclinic instability. Typical methods to investigate such instabilities are the wave-dynamical stability analysis of the atmospheric state and the parcel-dynamical stability analysis (e.g., Ertel et al., 1941; Gill, 1982; Holton, 2004; Vallis, 2006). The former may be regarded as a global analysis, where a wave-like perturbation

is added to the atmospheric state in the region of interest and the evolution of its amplitude is investigated within the framework

---

[5] An alternative, maybe more common model interpretation is, to regard the $\boldsymbol{v_j}$ defined on the interfaces, as the components of an "imaginary" cell-based velocity vector $\boldsymbol{v_i}$, to which $\boldsymbol{P_i} = \boldsymbol{v_i} m_i$ would be the corresponding momentum contained by the cell. So Eq. (31) would be replaced by $\sum_i \delta E_i = \sum_i [\boldsymbol{v_i} \cdot \delta \boldsymbol{P_i} + (\mu_i + \phi_{g,i}) \delta m_i + T_i \delta S_i]$, where the $\boldsymbol{v_i}$ and $\delta \boldsymbol{P_i}$ depend on the $\boldsymbol{v_j}$ and $\delta \boldsymbol{P_j}$ in some way. However, this alternative interpretation entails other problems on a triangular grid, as explained by Gassmann (2011).





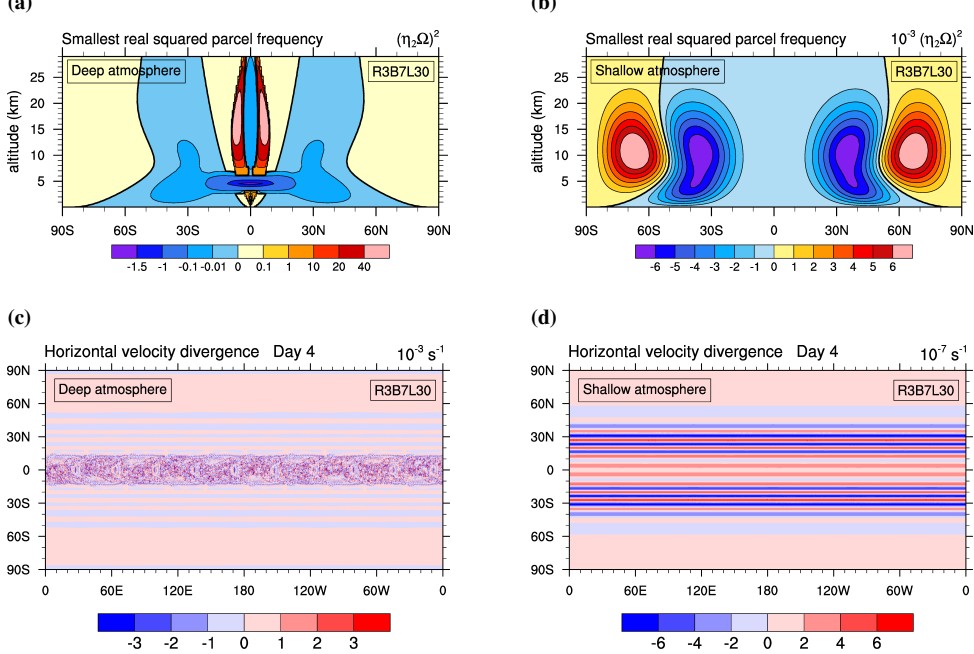

**Figure 8.** Upper row: the smallest real root $\tilde{\omega}_{\min}^2$ of Eq. (33) for the zonally symmetric atmospheric background state of the Jablonowski-Williamson baroclinic instability test for the deep atmosphere (a) and the shallow atmosphere (b). The computation was performed on the grid R3B7L30 ($\Delta\varphi = 0.12°$). Note the different color scales and the non-linear color scale of the left figure. The one black contour highlighted by thickness is the zero contour. Lower row: the horizontal velocity divergence at height $z = 5\,\mathrm{km}$ and day 4 for the deep atmosphere case (c) and the shallow atmosphere case (d). A dynamical time step of $\Delta t = \eta_1 \cdot 12\,\mathrm{s}$ was used for the simulations on the R3B7L30-grid. Note the different color scales.

of the linearized governing equations. In contrast, the parcel-dynamical stability analysis is a local analysis. It investigates, how a displaced fluid parcel behaves due to the state of its closest vicinity. In unstable regions, the displacement would increase exponentially. Due to the differences between the two ansatzes, it is likely possible that an atmospheric region that is unstable for some part of the wave spectrum according to the wave-dynamical analysis, is stable according to the parcel-dynamical

5 analysis and vice versa. Here, we make use of the parcel-dynamical analysis according to Ertel et al. (1941) (compare also Shutts and Cullen, 1987) for basically three reasons. We are only interested in the information, if the atmospheric state is unstable in some regions and how strong this instability may be (which, admittedly, may depend implicitly on the chosen analysis/measure). Our focus is not on the spatial structure of the developing instabilities. Second, the instabilities, as we observe them in the simulations are relatively small-scale and start to develop in the close vicinity of the equator, so that we

10 assume the local nature of the parcel-dynamical analysis to be suitable here. Finally, it is a relatively cheap analysis with regard to computational costs, compared to a wave-dynamical analysis.





According to Ertel et al. (1941), the evolution of a parcel displacement $\delta \boldsymbol{r}$ is governed by the equation

$$\left\{ \frac{\mathrm{d}^6}{\mathrm{d}t^6} + a\frac{\mathrm{d}^4}{\mathrm{d}t^4} + b\frac{\mathrm{d}^2}{\mathrm{d}t^2} + c \right\} \delta \boldsymbol{r} = 0, \tag{32}$$

where the coefficients $a$, $b$ and $c$ depend on the background state of the atmosphere, the stability of which shall be investigated. Detailed expressions for them are given in appendix B. Now, we limit the parcel displacement to the form $\delta \boldsymbol{r}(t) =$
$\delta \hat{\boldsymbol{r}} \exp(-i\omega t)$, so that Eq. (32) can be written in the form (Ertel et al., 1941)

$$(\omega^2)^3 - a(\omega^2)^2 + b(\omega^2) - c = 0 \quad \big| \cdot (\eta_2 \Omega)^{-6} \quad \Rightarrow \quad (\tilde{\omega}^2)^3 - \tilde{a}(\tilde{\omega}^2)^2 + \tilde{b}(\tilde{\omega}^2) - \tilde{c} = 0, \tag{33}$$

where we multiplied Eq. (32) by $-1$. In addition, $\tilde{\omega}^2 = \omega^2/(\eta_2 \Omega)^2$, $\tilde{a} = a(\eta_2 \Omega)^{-2}$, $\tilde{b} = b(\eta_2 \Omega)^{-4}$ and $\tilde{c} = c(\eta_2 \Omega)^{-6}$. By means of the scaling in the last step, we measure $\omega$ in the units of the frequency of a model day $\eta_2 \Omega$, which may be easier to interpret. A detailed analysis of the roots of this cubic equation in $\tilde{\omega}^2$ is beyond the scope of this work. We limited our analysis to find
the smallest real root $\tilde{\omega}_{\min}^2$ (a cubic equation should have at least one real root (Bronstein et al., 2001)). Regions, in which this root is negative, should be unstable with respect to parcel displacements. We applied the analysis to the zonally symmetric background state of the atmosphere of this test case. An approximation of the root $\tilde{\omega}_{\min}^2$ is determined by an iterative algorithm. Some caution may be in order with regard to the accuracy of this approximation, since cubic equations can be quite challenging for root finding algorithms, such as the Newton's method. However, sensitivity tests, in which we changed the number of
iterations, for instance, indicate that the solution is robust. The result is shown in Fig. 8 for the deep-atmosphere version of this test case (Fig. 8a) and for the shallow-atmosphere version (Fig. 8b). The dipole structures in about the northern and southern mid-latitudes that are visible in Fig. 8b, are present in the deep-atmosphere version as well, but we changed the color scale of Fig. 8a, to highlight the most prominent difference between the two test case versions, which is located in the atmospheric column above the equator region. The dark blue and purple region in Fig. 8a, in a height of about $5\,\mathrm{km}$ seems to be the most
unstable region in terms of $\tilde{\omega}_{\min}^2$. There, parcel displacements accelerate on a typical time scale of a model day, which is about three orders of magnitude larger than in the unstable regions over the equator in case of the shallow-atmosphere version. And it is exactly this dark blue region, where we observe the evolution of small-scale structures, for instance, in the field of the horizontal velocity divergence (Fig. 8c). This fine-grained "belt" spanning the equator is absent in the shallow-atmosphere test case (Fig. 8d). We see only larger-scale structures, varying in meridional direction, whose amplitude is about four orders of
magnitude smaller than the amplitude of the afore-mentionde instabilities in the deep-atmosphere version. They are likely the expression of an adjustment process that results from the atmospheric background state, specified in the test case, being an analytical solution to the differential equations of the hydrostatic and geostrophic balances, but only an approximate solution with regard to the respective difference equations of the model. In case of the shallow-atmosphere, we determined the second and third root of Eq. (33) as well. Both are real, greater than zero and greater than $\tilde{\omega}_{\min}^2$ everywhere, so we can be relatively
confident that no instability is missed in that test case version by limiting the analysis to $\tilde{\omega}_{\min}^2$. The small-scale instabilities in the deep-atmosphere simulation gradually spread in meridional direction, and may be recognized in the southern half of the surface pressure plots of Fig. 6 as well. It might be the presence of these instabilities on top of the baroclinic instability that triggers or contributes to the development of the numerical instability in our simulation. One may think that the magnitude of the observed



natural instabilities could be reduced by scaling the explicit numerical damping of the model, or part of it, with some factor $1 + \gamma[(|\tilde{\omega}_{\min}^2| - \tilde{\omega}_{\min}^2)/2]^{1/n}$, say, where $n$ may be 1 or 2, and $\gamma$ is a tunable coefficient. This is, however, unfeasible, since the computation of the coefficients $a$, $b$ and $c$ in Eq. (33), as well as the iterative determination of $\tilde{\omega}_{\min}^2$ are way too expensive in a three-dimensional, time-dependent simulation. Although a more in-depth analysis of the instability is beyond the scope of

this work, we note that it is probably not buoyancy-driven. To show this, we consider the vertical component of the prognostic equation for the parcel displacement (B3) $\mathrm{d}^2\delta r/\mathrm{d}t^2 + \{[\partial^2 p/\partial r^2 - (\partial p/\partial r)^2/(\kappa p)]/\rho\}\delta r + \cdots = 0$, where $\kappa = c_p/c_v$ and the $\cdots$ stand for further terms that are unimportant for this consideration. If we apply that the prescribed atmospheric background state satisfies the hydrostatic balance $\partial p/\partial r = -g\rho + \cdots$, where the $\cdots$ denote further terms in case of the deep atmosphere (see Eq. (12) in Ullrich et al., 2014), we find $[\partial^2 p/\partial r^2 - (\partial p/\partial r)^2/(\kappa p)]/\rho = (g/T)(\partial T/\partial r + g/c_p) + \cdots = N^2 + \cdots$, with the

squared Brunt-Väisälä frequency $N^2$. According to Ullrich et al. (2014), $N$ is real everywhere for the atmospheric background state (see their Fig. 1d). So the instability does not originate from this side, although $N$ has its smallest values in a region that overlies the dark blue region in Fig. 8a.

Since the numerical instability is limited to relatively high horizontal grid resolutions under the extreme conditions of the small-Earth approach, but absent, if we change to the standard Earth, we regard it as acceptable. Apart from that, the comparison

of Figs. (6) and (7) with the benchmark from Ullrich et al. (2014) make us confident that our deep-atmosphere implementation is satisfactory for our purposes.

## 3.2 Climatological test cases

### 3.2.1 Simulation setup

For the evaluation of the model climatology, a UA-ICON simulation with the upper atmosphere physics coupled to the ECHAM

physics package has been performed. The deep-atmosphere dynamics is also switched on, except for the height-dependent gravitational acceleration, due to an unidentified bug in the current version of the interface between the extended dynamical core and the ECHAM physics. We assume that this limitation will not affect the results severely. The model was integrated for 20 years with climatological boundary conditions: sea surface temperature and sea ice concentration are averaged for each calendar month from the PCMDI AMIP dataset (Taylor et al., 1998) version 1.1.2 over 1979–2014; concentrations of radiatively

active gases, namely $O_3$, $CO_2$, $O_2$, $O$, $NO$, $CH_4$, $N_2O$, are averaged in the same manner from a 35-year HAMMONIA simulation with fixed present-day boundary conditions; concentrations of CFC-11 and CFC-12 are fixed at 214.5 pptv and 371.1 pptv, respectively; the 1865 condition of the tropospheric background aerosol from the MAC-v1 dataset (Kinne et al., 2013) is used; no volcanic or anthropogenic aerosols are used; land-surface parameters for the parameterization of the effects of sub-grid scale orography and for the embedded version of the JSBACH land-surface model (v4; Giorgetta et al., 2018) are

fixed as described by Giorgetta et al. (2018). The total solar irradiance is held constant at $1361.371\,\mathrm{W\,m}^{-2}$, and the F10.7 index for the calculation of EUV heating rates is fixed at $150\,\mathrm{sfu}$ ($1\,\mathrm{sfu} = 1 \times 10^{-22}\,\mathrm{W\,m}^{-2}\,\mathrm{Hz}^{-1}$).

The simulation uses the R2B4 grid, which has a horizontal mesh size of about $160\,\mathrm{km}$. In the vertical, the model uses 120 layers for the altitude range from the surface up to $150\,\mathrm{km}$. Rayleigh damping (Klemp et al., 2008; Zängl et al., 2015) is





applied above 120 km with a maximum damping coefficient of $10\,\mathrm{s}^{-1}$ at the top. Such strong damping is necessary to allow for a reasonable computational time step despite the occasionally very large vertical velocities in the thermosphere. The model was integrated with a (physical) time step of 4 min and 5 dynamical substeps each physical time step. Radiation parameterizations – i.e. the PSrad radiation scheme of ICON, the shortwave radiation in the SRBC and EUV, the non-LTE longwave radiation, the

NO heating – are evaluated once every hour, whereas all other parameterizations are evaluated every time step. For all physics parameterization we apply the "all-fast" treatment described by Giorgetta et al. (2018). For non-orographic gravity wave drag, a cutoff maximum vertical wavelength of 12 km is applied, thus prohibiting long gravity waves. Disabling these long gravity waves is physically sensible, as they are believed to strongly propagate horizontally and be subject to internal reflection before reaching the mesopause (Hines, 1997b).

Companion simulations have been performed with two ICON configurations using a standard model lid at 80 km, Rayleigh damping (maximum damping coefficient $1\,\mathrm{s}^{-1}$) applied above 50 km, and 100 vertical levels exactly following the lower part of the vertical grid applied in the UA-ICON simulations. In the first configuration (referred to as ICON in the following) the deep-atmosphere dynamics and the upper-atmosphere physics are disabled. All other numerical and physical settings are identical to the UA-ICON run. The second configuration (referred to as ICON(UA)) additionally has the deep atmospheric

dynamics and upper atmosphere physics enabled. With the help of these two configurations we can estimate which of the differences between ICON and UA-ICON are due to the vertical extension and which are related to the application of extended physics and dynamics also below 80 km.

For evaluation, a 15-year (2002 to 2016) temperature climatology from observations of the Sounding of the Atmosphere using Broadband Emission Radiometry (SABER) instrument on NASA's Thermosphere Ionosphere Mesosphere Energetics

Dynamics (TIMED) satellite is used (v 2.0; Dawkins et al., 2018). A monthly zonal mean zonal wind climatology is taken from the Upper Atmosphere Research Satellite Reference Atmosphere Project (URAP; Swinbank and Ortland, 2003).

### 3.2.2 Comparison of simulated and observed climatologies

Figure 9 shows multi-year zonal mean temperatures for January and July from the UA-ICON and ICON simulations and from SABER. The observed temperature patterns are reasonably reproduced in the simulations for large parts of the stratosphere and

mesosphere. UA-ICON simulates the low summer mesopause temperatures and its altitude well. However, UA-ICON's winter mesopause is characterized by a vertically extended region of low temperatures and misses the observed distinct temperature minimum near 100 km. Further sensitivity tests will be needed to identify, if this may be improved by further tuning of the gravity wave parameterizations or may rather be caused by oversimplifications in other physical parameterizations, e.g. the treatment of chemical heating.

UA-ICON and ICON both simulate the stratopause and tropopause fairly accurately, although the winter stratosphere is slightly warmer than suggested by the observations and the tropical tropopause is too cold, which may at least partly be related to the absence of stratospheric aerosol or to the prescribed climatological ozone. Differences between UA-ICON and ICON on the one hand, and between ICON(UA) and ICON on the other hand are presented in Fig. 10, again for the months of January and July. It is clear that the differences below about 60 km are very similar, indicating that in this region they are mostly related





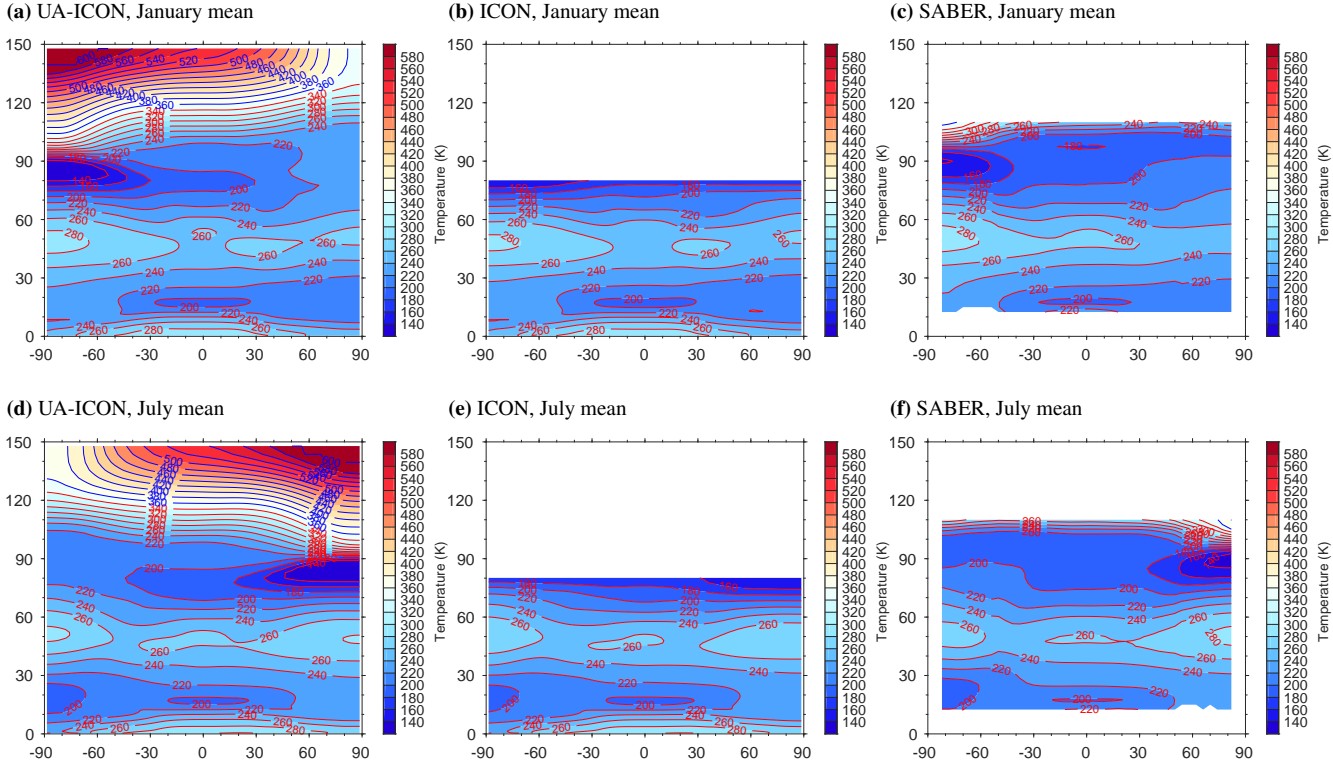

**Figure 9.** Climatological zonal-mean temperature for (top) January and (bottom) July averages from the 20-year simulations with (left) UA-ICON, (middle) ICON and (right) SABER satellite retrievals averaged over the years 2002 to 2016. (The vertical and horizontal axes show the height in km and the latitude in degree, respectively.)

to the extension of the dynamical and physical processes and not to the vertical extension. In most areas the process extension leads to higher temperatures with the strongest signals of up to about 4 K in the summer middle stratosphere and even stronger in the winter lower mesosphere. Above about 60 km the patterns of the differences are again similar but the magnitude is stronger for the difference between UA-ICON and ICON, meaning that here also the vertical extension adds a warming in comparison to the standard configuration of ICON. At the uppermost level of comparison the temperature differences reach several tens of Kelvin (max and min values are denoted at the top of each panel). The vertical extension may affect the temperature difference to standard ICON by both the actual vertical extension and by omitting the Rayleigh damping at these altitudes. It is clear from these comparisons that a vertical model extension beyond 80 km as implemented in UA-ICON even influences simulated climatological means down to at least 60 km.

Zonal mean zonal wind climatologies are presented in Fig. 11. Patterns simulated by UA-ICON and ICON agree qualitatively with the observation-based URAP climatology. The sign reversals of zonal wind in both hemispheres near the mesopause are simulated in UA-ICON, but are in general too strong, i.e. the lower-thermospheric jets are too strong, and peak at too low altitudes. Experience from test-experiments show that this can very likely be adjusted by tuning the non-orographic gravity





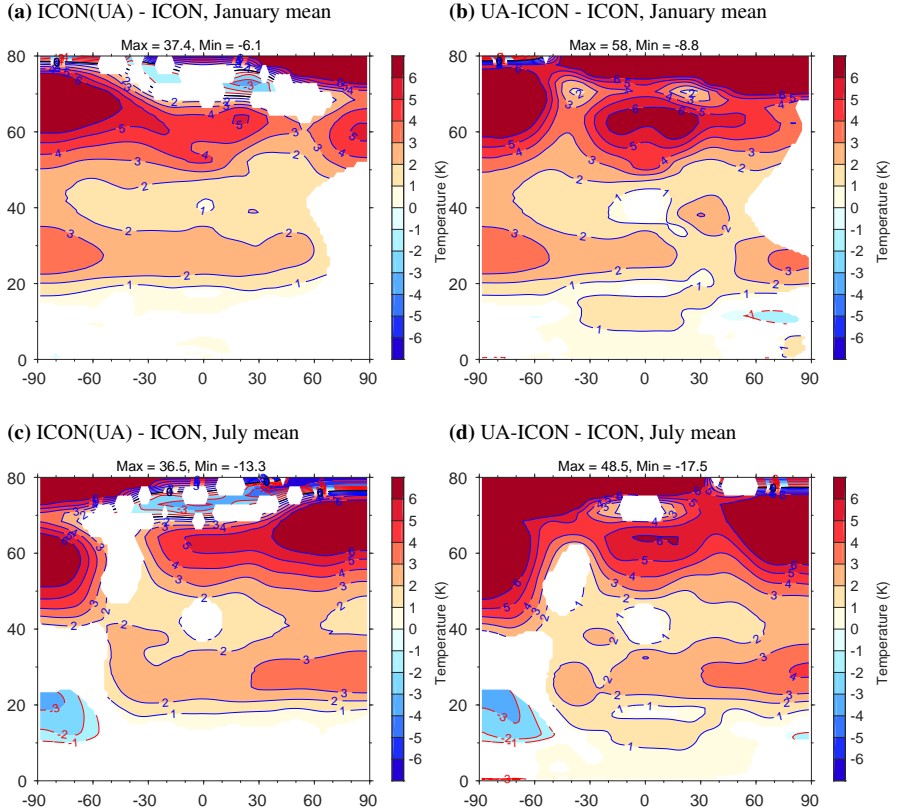

**Figure 10.** Climatological zonal-mean temperature differences for (top) January and (bottom) July averages from the 20-year simulations between (left) ICON(UA) (i.e. with extended dynamics and physics, but lid at 80 km) and ICON, and (right) UA-ICON (i.e. with model lid at 150 km) and ICON. Only differences statistically significant at the 95% level using a t-test are shown. The max and min differences are indicated at the top of each panel. (The vertical and horizontal axes show the height in km and the latitude in degree, respectively.)

wave parameterization. Concerning stratospheric winds, it is obvious that the winter westerlies are too weak in ICON and UA-ICON, an issue already mentioned in the ICON evaluation by Crueger et al. (2018). While UA-ICON and ICON show very similar biases for the boreal winter jet, the problem is reduced in austral winter. It's no surprise that our UA-ICON simulation, performed with the same settings for the sub-grid scale orography (SSO) parameterization as used by Crueger et al. (2018),
5 shows similar issues. A reduction of the orographic gravity wave sources would reduce this issue in particular in the Northern hemisphere, but has not been implemented by Crueger et al. (2018) as it would deteriorate near-surface winds. Retuning of orographic and non-orographic gravity wave parameters is planned for future model versions.





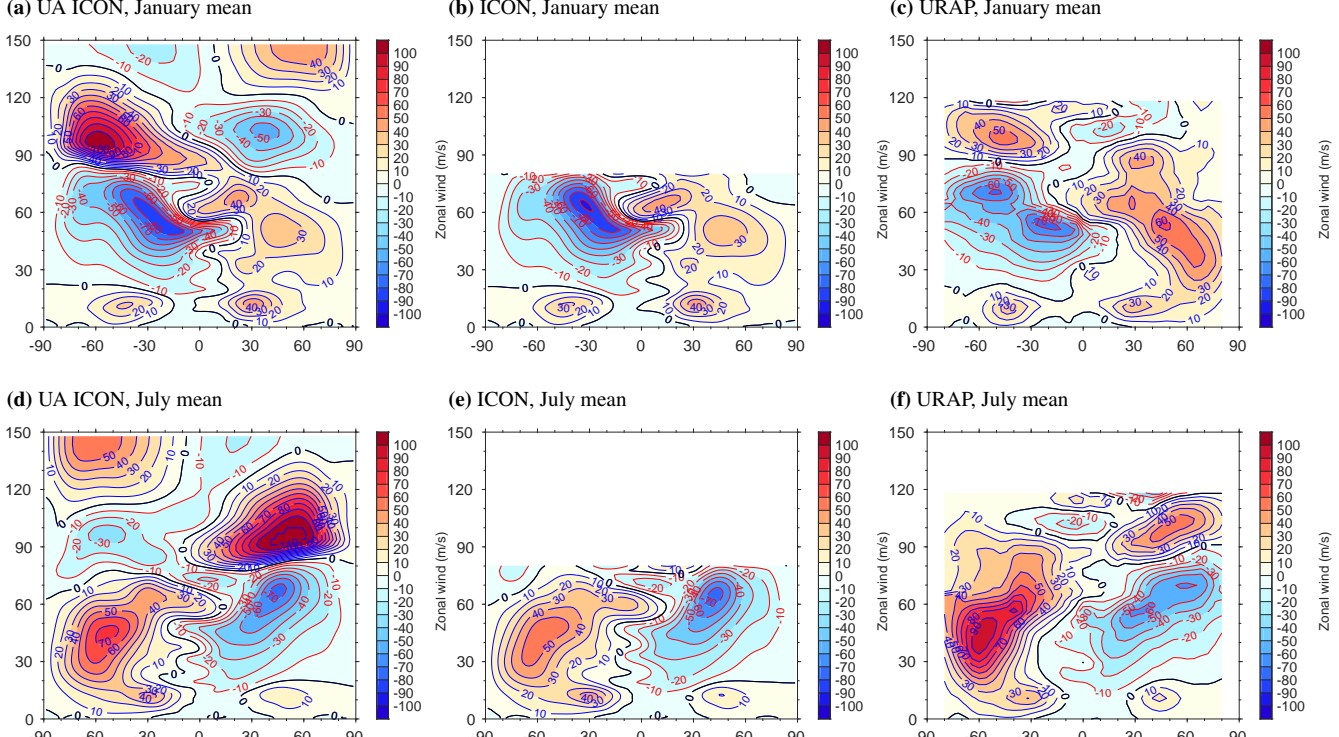

**Figure 11.** Climatological zonal-mean zonal wind for (top) January and (bottom) July averages from the 20-year simulations with (left) UA-ICON, (middle) ICON and (right) the URAP climatology. (The vertical and horizontal axes show the height in km and the latitude in degree, respectively.)

## 4 Conclusions

An upper-atmosphere extension of the ICOsahedral Non-hydrostatic (ICON) general circulation model has been presented. This includes the extension of the dynamical core from a shallow-atmosphere to a deep-atmosphere formulation, in order to account for the spherical shape of the atmosphere and the gravitational field as well as to account for the non-traditional part of the Coriolis acceleration. In addition, the physics parameterizations have been complemented by processes which become relevant in the rarified air of the upper mesosphere and lower thermosphere. For instance molecular diffusion takes over the lead role for mixing from turbulence, and various processes are linked to the relatively strong high-frequency solar irradiance in the upper atmosphere: its absorption is a source of heat; air compounds being ionized by the radiation align with the electromagnetic field of the Earth, forming currents that in turn interact with the neutral air flow via ion drag and Joule heating; atoms and radicals, as the products of photolysis, undergo a recombination reaction that is accompanied by chemical heating.

The new implementations, subsumed under the configuration name UA-ICON, have been validated by means of idealized test cases in terms of the dynamical core, and by climate simulations the results of which were compared to results from





satellite observations. Two test cases were performed. The first one follows a method proposed by Läuter et al. (2005), and considers the propagation of a spherically symmetric sound wave in an atmosphere at rest in an absolute frame and so in a state of solid-body-rotation relative to the rotating Earth. This requires accounting for the centrifugal acceleration explicitly in the dynamical core. In addition gravity is switched off in order to simplify the derivation of an analytical solution, the comparison

to which allows to quantify, e.g., how well the isotropic sound wave propagation is reproduced by the dynamical core on the anisotropic spherically curved grid, and how well it maintains the solid-body-rotation of the air. The second test case is the Jablonowski-Williamson baroclinic instability test case in its formulation for the deep atmosphere by Ullrich et al. (2014). Its focus is on testing the representation of synoptic-scale flows through the simulation of the life cycle of a baroclinic wave on a hydrostatically and geostrophically balanced, zonally symmetric background state of the atmosphere. As with the first test

case, the small-Earth approach of Wedi and Smolarkiewicz (2009) is applied, to amplify the effects of the spherical curvature. No analytical solution is available for the second test case, so it relies on a model intercomparison. In either test case UA-ICON showed satisfying performance, and no indication of a severe deficiency of the deep-atmosphere modification of the dynamical core was found.

For the evaluation of the upper-atmosphere physics three AMIP-type simulations, each spanning 20 years, were performed

and compared to temperature and zonal wind climatologies obtained from measurements of the SABER and TIMED satellite instruments and the URAProject, respectively. The first simulation uses a setup which we regard as typical for our envisaged first applications of UA-ICON. The model top is located at a height of $150\,\mathrm{km}$, and the newly implemented upper-atmosphere physics are enabled in addition to the standard physics of ICON. The deep-atmosphere modifications of the dynamical core are applied as well (except for the vertical variation of the gravitational acceleration). In the second simulation the model

top is lowered to $80\,\mathrm{km}$, and the sponge layer is adjusted accordingly. Apart from that, the settings are identical to those of the first simulation. The third simulation, finally, equals the second simulation, but is a standard ICON simulation with the upper-atmosphere physics and the deep-atmosphere modifications of the dynamics switched off. By comparing the three simulations, we try to quantify the effect of the upper-atmosphere extension on the middle and lower atmosphere. The temperature climatologies for January and July from the first type of simulation with UA-ICON are generally in good agreement with the

observations. For instance, the atmospheric temperature minimum at the summer mesopause is relatively well captured. The less pronounced minimum in the winter hemisphere at about $100\,\mathrm{km}$, however, is less well reproduced in its magnitude and vertical extent, which, as we speculate, might be alleviated by a retuning of the gravity wave parameterization. In addition, a slightly too warm winter stratosphere and too cold tropical tropopause might be due to the absence of stratospheric aerosol in the simulation, or result from the employed ozone climatology. When it comes to the climatologies of the zonal wind for

January and July, we find again that UA-ICON reproduces the qualitative structure of the wind in that part of the atmosphere observed by URAP satisfactorily in general. One remaining issue is that the magnitude of the jets in the lower thermosphere is too large in UA-ICON, roughly by a factor of two. We are currently investigating, to what extent a retuning of the non-orographic gravity wave parameterization could alleviate the problem without losing accuracy in the middle and lower part of the atmosphere. The comparison of UA-ICON simulations and two different model configurations with a lower model top at 80



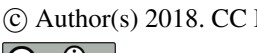

km has shown that the addition of upper atmosphere physics and dynamics also affects stratospheric temperatures (increasing them by up to $4\,\mathrm{K}$), and that the vertical extension has noticeable effects at least down to about $60\,\mathrm{km}$.

As an important application for UA-ICON we have in view the investigation of the impact of gravity waves on the global atmospheric circulation with a special focus on the feedback of the middle and upper atmosphere on the tropospheric weather

and climate. The high scalability of ICON on massively parallel computers will allow us to employ model configurations with much higher grid resolutions and eventually resolve large parts of the atmospheric gravity wave spectrum instead of parameterizing it.

*Code and data availability.*    The ICON-Software is freely available to the scientific community for non-commercial research purposes under a license of DWD and MPI-M. UA-ICON exists in two configurations at the time of writing this article. One for the upper-atmosphere devel-

opments within the development framework of the ICON-Software for climate simulations, another one for the upper-atmosphere develop-ments within the NWP development framework of the ICON-Software. If you would like to obtain UA-ICON, please contact icon@dwd.de. You will be provided an institutional license, which needs to be signed by the representative of your Research Institute and send back to the DWD. The ICON-Software is controlled by a GIT version control system and upon license agreement a tar ball of the version of UA-ICON that was used, to produce the results presented in this article is provided to you. The license in its current form can be viewed

on https://code.mpimet.mpg.de/projects/iconpublic/wiki/How%20to%20obtain%20the%20model%20code (last access: 14 November 2018). The data shown in section 3 as well as the scripts used for their production, postprocessing and plotting are provided as tar ball in the sup-plementary material of this article.

## Appendix A: Sound wave test case

### A1    Formulation in the rotating frame

Here we consider the transformation of a solution to the deep-atmosphere equations, linearized about an atmosphere at rest in the absolute frame, to a frame rotating with angular velocity $\boldsymbol{\Omega}$. For the atmosphere at rest in the absolute frame $\boldsymbol{v_a} = \boldsymbol{v_F} + \boldsymbol{v} = 0$ holds, where $\boldsymbol{v_F} = \boldsymbol{\Omega} \times \boldsymbol{r}$, $\boldsymbol{v}$ being the velocity observed in the rotating frame, and $\boldsymbol{r} = \boldsymbol{X} - \boldsymbol{A}$ (with the center of the Earth $\boldsymbol{A}$ and an arbitrary point $\boldsymbol{X}$ not coincident with $\boldsymbol{A}$). Therefore, in the rotating frame the motion of the air follows

$$\rho \left[ \frac{\mathrm{d}\boldsymbol{v}}{\mathrm{d}t} + 2\boldsymbol{\Omega} \times \boldsymbol{v} + \boldsymbol{\Omega} \times (\boldsymbol{\Omega} \times \boldsymbol{r}) \right] = 0$$

$$\Leftrightarrow \quad \rho \left[ \frac{\mathrm{d}}{\mathrm{d}t}(-\boldsymbol{\Omega} \times \boldsymbol{r}) - 2\boldsymbol{\Omega} \times (\boldsymbol{\Omega} \times \boldsymbol{r}) + \boldsymbol{\Omega} \times (\boldsymbol{\Omega} \times \boldsymbol{r}) \right] = 0$$

$$\Leftrightarrow \quad -\rho \boldsymbol{\Omega} \times \left( \frac{\mathrm{d}\boldsymbol{r}}{\mathrm{d}t} + \boldsymbol{\Omega} \times \boldsymbol{r} \right) = 0 \tag{A1}$$

$$\Rightarrow \quad \frac{\mathrm{d}\boldsymbol{r}}{\mathrm{d}t} = -\boldsymbol{\Omega} \times \boldsymbol{r} = -\boldsymbol{\Omega} \times (\mathbf{1} \cdot \boldsymbol{r}) = -(\boldsymbol{\Omega} \times \mathbf{1}) \cdot \boldsymbol{r} = -\mathbf{W} \cdot \boldsymbol{r}. \tag{A2}$$

where Eqs. (A1) and (A2) are equivalent. In the last step we have introduced the identity tensor $\mathbf{1}$ and the antisymmetric Coriolis tensor $\mathbf{W} = \boldsymbol{\Omega} \times \mathbf{1}$ to simplify the following considerations (compare Wilson, 1929; Zdunkowski and Bott, 2003).





The solution to Eq. (A2) is

$$\boldsymbol{r}(t) = \exp(-\mathbf{W}t) \cdot \boldsymbol{r_0}, \tag{A3}$$

where $\boldsymbol{r_0} = \boldsymbol{r}(t=0)$. Introducing $\widehat{\boldsymbol{\Omega}} = \boldsymbol{\Omega}/\Omega$, and $\widehat{\mathbf{W}} = \mathbf{W}/\Omega$, with $\Omega = |\boldsymbol{\Omega}|$ and using

$$\exp(-\Omega\widehat{\mathbf{W}}t) = \mathbf{1} - \frac{1}{1!}\widehat{\mathbf{W}}\Omega t + \frac{1}{2!}\widehat{\mathbf{W}}^2\Omega^2 t^2 - \frac{1}{3!}\widehat{\mathbf{W}}^3\Omega^3 t^3 \pm \cdots, \tag{A4}$$

where $\widehat{\mathbf{W}}^2 = \widehat{\mathbf{W}} \cdot \widehat{\mathbf{W}}$, for instance, in combination with

$$\widehat{\mathbf{W}}^n = \begin{cases} (-1)^{(n-1)/2}\widehat{\mathbf{W}}, & \text{for odd } n \\ (-1)^{n/2}\widehat{\mathbf{1}}, & \text{for even } n \end{cases}, \tag{A5}$$

where $\widehat{\mathbf{1}} = \mathbf{1} - \widehat{\boldsymbol{\Omega}}\widehat{\boldsymbol{\Omega}}$ and $\widehat{\boldsymbol{\Omega}}\widehat{\boldsymbol{\Omega}}$ denotes the dyadic product of $\widehat{\boldsymbol{\Omega}}$ with itself, the exponential factor in Eq. (A3) can be expanded as

$$\exp(-\Omega\widehat{\mathbf{W}}t) = \widehat{\boldsymbol{\Omega}}\widehat{\boldsymbol{\Omega}} + \cos(\Omega t)\widehat{\mathbf{1}} - \sin(\Omega t)\widehat{\mathbf{W}}. \tag{A6}$$

So Eq. (A3) can be rewritten in a form which is easier to evaluate (compare Wilson, 1929; Bronstein et al., 2001; Zdunkowski and Bott, 2003)

$$\boldsymbol{r}(t) = \widehat{\boldsymbol{\Omega}}\widehat{\boldsymbol{\Omega}} \cdot \boldsymbol{r_0} + \cos(\Omega t)\widehat{\mathbf{1}} \cdot \boldsymbol{r_0} - \sin(\Omega t)\widehat{\mathbf{W}} \cdot \boldsymbol{r_0} = \cos(\Omega t)\boldsymbol{r_0} - \sin(\Omega t)\widehat{\boldsymbol{\Omega}} \times \boldsymbol{r_0} + \left\{[1 - \cos(\Omega t)]\widehat{\boldsymbol{\Omega}} \cdot \boldsymbol{r_0}\right\}\widehat{\boldsymbol{\Omega}}. \tag{A7}$$

Next, given, for instance, an analytical solution in the absolute frame, which is spherically symmetric with respect to the point $\boldsymbol{B}$, therefore depending only on $|\boldsymbol{x} - \boldsymbol{B}| = |(\boldsymbol{x} - \boldsymbol{A}) - (\boldsymbol{B} - \boldsymbol{A})| = \{[(\boldsymbol{x} - \boldsymbol{A}) - (\boldsymbol{B} - \boldsymbol{A})] \cdot [(\boldsymbol{x} - \boldsymbol{A}) - (\boldsymbol{B} - \boldsymbol{A})]\}^{1/2}$, we would find the solution unaltered in the rotating frame relative to $\exp(-\mathbf{W}t) \cdot (\boldsymbol{B} - \boldsymbol{A})$, since e.g.

$$\begin{aligned}
[\exp(-\mathbf{W}t) \cdot (\boldsymbol{X} - \boldsymbol{A})] \cdot [\exp(-\mathbf{W}t) \cdot (\boldsymbol{B} - \boldsymbol{A})] &= [(\boldsymbol{X} - \boldsymbol{A}) \cdot \exp(-\mathbf{W}t)^{\intercal}] \cdot [\exp(-\mathbf{W}t) \cdot (\boldsymbol{B} - \boldsymbol{A})] \\
&= (\boldsymbol{X} - \boldsymbol{A}) \cdot [\exp(\mathbf{W}t) \cdot \exp(-\mathbf{W}t)] \cdot (\boldsymbol{B} - \boldsymbol{A}) \\
&= (\boldsymbol{X} - \boldsymbol{A}) \cdot (\boldsymbol{B} - \boldsymbol{A}). \tag{A8}
\end{aligned}$$

Here, $(\cdot)^{\intercal}$ denotes the transpose, and we have used the identity $\exp(-\mathbf{W}t)^{\intercal} = \exp(\mathbf{W}t)$ which can be derived from Eq. (A6) and $\widehat{\boldsymbol{\Omega}}\widehat{\boldsymbol{\Omega}}^{\intercal} = \widehat{\boldsymbol{\Omega}}\widehat{\boldsymbol{\Omega}}$, $\mathbf{1}^{\intercal} = \mathbf{1}$ and $\widehat{\mathbf{W}}^{\intercal} = -\widehat{\mathbf{W}}$. In addition, $\exp(\mathbf{W}t) \cdot \exp(-\mathbf{W}t) = \mathbf{1}$ can be shown to hold, using Eqs. (A6), (A5), and the definition of $\widehat{\mathbf{W}}$. The above is not a mathematically rigorous proof for the general case, but we hope it can help to illustrate the basic idea behind the method. For detailed mathematical examinations we refer the reader to Läuter et al. (2005); Staniforth and White (2008). In the following section such a spherically symmetric solution to be rotated is derived.

## A2 Derivation of the sound wave solution

Here we derive the analytical solution for the expansion of sound waves for the deep-atmosphere equations (1)-(4) reduced to the reversible processes (i.e., $\boldsymbol{F} = 0$ and $Q = 0$), without gravity ($g = 0$), linearized about an isothermal atmosphere of temperature $T_0 = \text{const.}$, at rest in the absolute frame (i.e., $\boldsymbol{v_a} = \boldsymbol{v_F} + \boldsymbol{v_0} = \boldsymbol{\Omega} \times \boldsymbol{r} + \boldsymbol{v_0} = 0$). In particular, the neglection





of gravity leads to a constant base state for pressure $\pi_0 = \text{const.}$ (from Eq. (1)) and density $\rho_0 = \text{const.}$ (from Eq. (4)), too. Otherwise, the derivation of an analytic solution even for the linearized equations would be very difficult.

Linearisation of Eqs. (1) to (4) about this base state leads to:

$$\frac{\partial \boldsymbol{v}'}{\partial t} = -c_p \theta_0 \boldsymbol{\nabla} \pi', \tag{A9}$$

$$\frac{\partial \rho'}{\partial t} + \rho_0 \boldsymbol{\nabla} \cdot \boldsymbol{v}' = 0, \tag{A10}$$

$$\frac{\partial \pi'}{\partial t} + \frac{R}{c_v} \pi_0 \boldsymbol{\nabla} \cdot \boldsymbol{v}' = 0, \tag{A11}$$

resulting in the sound wave equation

$$\frac{\partial^2 \pi'}{\partial t^2} - c_s^2 \nabla^2 \pi' = 0. \tag{A12}$$

with the squared speed of sound $c_s^2 = (c_p/c_v)RT_0$.

Without gravity and disregarding the spherical boundaries of the atmosphere there is no longer a distinct direction (or rather a distinct point). Therefore, we seek a solution which is spherically symmetric with respect to an arbitrary point $\boldsymbol{B}$ within the boundaries. Using the expansion of the Laplace-operator in spherical coordinates (e.g., Bronstein et al., 2001), we can write

$$\frac{\partial^2 \pi'}{\partial t^2} - c_s^2 \left( \frac{2}{x} \frac{\partial \pi'}{\partial x} + \frac{\partial^2 \pi'}{\partial x^2} \right) = 0, \tag{A13}$$

with the radial coordinate $x = |\boldsymbol{X} - \boldsymbol{B}|$ (to avoid any confusion with the distance $r$ from the center of the Earth). This equation is valid for $x > 0$. Using the transform

$$\tilde{\pi} = x\pi', \tag{A14}$$

one finds for $x \cdot$ (A13) (Nolting, 2004)

$$\frac{\partial^2 \tilde{\pi}}{\partial t^2} - c_s^2 \frac{\partial^2 \tilde{\pi}}{\partial x^2} = 0. \tag{A15}$$

In addition, we can derive from Eq. (A11)

$$\frac{\partial \pi'}{\partial t} = -\frac{R}{c_v} \pi_0 \left( \frac{2}{x} v' + \frac{\partial v'}{\partial x} \right) \quad \bigg| \, x \cdot$$

$$\Rightarrow \quad \frac{\partial \tilde{\pi}}{\partial t} = -\frac{R}{c_v} \pi_0 \frac{1}{x} \frac{\partial x \tilde{v}}{\partial x} \tag{A16}$$

where $\boldsymbol{v}' = v' \boldsymbol{e_x}$, $\boldsymbol{e_x} = (\boldsymbol{X} - \boldsymbol{B})/|\boldsymbol{X} - \boldsymbol{B}|$, and $\tilde{v} = x v'$. Equation (A16) is required for the specification of the initial conditions. Given the 1d wave equation (A15) a general solution is of the form (Nolting, 2004)

$$\tilde{\pi}(x,t) = f_1(x + c_s t) + f_2(x - c_s t), \tag{A17}$$

where $f_1$ and $f_2$ denote incoming and outgoing waves of arbitrary shape, respectively. Note, that the transform (A14) induces a boundary condition at $x = 0$, because finite values of $\pi'$ require $\tilde{\pi}(x = 0, t) = 0$. Therefore, the outgoing wave for later times is determined by the incoming wave at the origin:

$$f_2(-c_s t) = -f_1(+c_s t) \quad \text{for } t \geq 0.$$

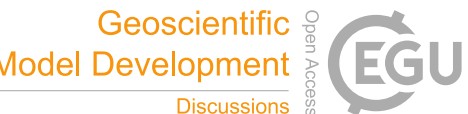

Since the wave equation (A15) is of second order in time, initial conditions for $\tilde{\pi}(x, t=0)$ and for its derivative

$$\frac{\partial \tilde{\pi}}{\partial t}\bigg|_{t=0} = c_s \left( \frac{\mathrm{d}f_1}{\mathrm{d}x} - \frac{\mathrm{d}f_2}{\mathrm{d}x} \right) \tag{A18}$$

are required. Instead, we require that the incoming wave vanishes ($f_1 = 0$) and prescribe the initial wind field

$$v'(x) = \delta v \sin\left( \pi \frac{x - b_1}{b_2 - b_1} \right) \sin\left( 2\pi n \frac{x - b_1}{b_2 - b_1} \right) [\Theta(x - b_1) - \Theta(x - b_2)], \tag{A19}$$

where $\delta v$ is a constant wind amplitude, $n$ denotes the number of wave crests, and $x = b_1 > 0$ and $x = b_2 > b_1$ are the radii, between which the sound wave has a non-vanishing amplitude at $t = 0$. The Heaviside step function is given by (Bronstein et al., 2001)

$$\Theta(\xi) = \begin{cases} 0, & \text{for} \quad \xi < 0 \\ 1, & \text{for} \quad \xi \geq 0 \end{cases}. \tag{A20}$$

The function (A19) was chosen not least because it satisfies $v'(x = b_1, b_2) = 0$ and $\mathrm{d}v'/\mathrm{d}x|_{x=b_1, b_2} = 0$, and guarantees that the derived fields $\pi'$, $\theta'$, and $\rho'$ do likewise, so that discontinuities pose no extra challenge to the numerical simulation. For the implementation of the test case, we preferred to specify the wave amplitude by means of a temperature amplitude $\delta T$, instead of the wind amplitude $\delta v$, and use the relation $\delta v = (c_v/R)(\delta T/T_0)c_s$ for this purpose. From the initial wind field (A19) we can calculate the derivative $df_2(x)/dx$ by Eqs. (A16) and (A18). Integration of $df_2/dx$ yields the Exner pressure perturbation (compare Bronstein et al., 2001)

$$\pi'(x, t) = \frac{R\pi_0 \delta v}{c_v c_s} \left\{ \frac{x - c_s t}{x} \sin\left( \pi \frac{x - b_1 - c_s t}{b_2 - b_1} \right) \sin\left( 2\pi n \frac{x - b_1 - c_s t}{b_2 - b_1} \right) \right.$$
$$\left. + \frac{b_2 - b_1}{x} \left[ \frac{\sin\left( \pi(2n-1)\frac{x-b_1-c_s t}{b_2-b_1} \right)}{2\pi(2n-1)} - \frac{\sin\left( \pi(2n+1)\frac{x-b_1-c_s t}{b_2-b_1} \right)}{2\pi(2n+1)} \right] \right\} [\Theta(x - b_1 - c_s t) - \Theta(x - b_2 - c_s t)]. \tag{A21}$$

or, alternatively, expressed as pressure perturbation

$$p' = \frac{c_p}{R} \frac{p_0}{\pi_0} \pi'. \tag{A22}$$

The initialization of the remaining thermodynamic fields (here, e.g., $\rho$ and $\theta$) can be chosen arbitrarily as long as the linearized ideal gas law is fulfilled. From Eq. (4) follows

$$\frac{\rho'}{\rho_0} + \frac{\theta'}{\theta_0} = \frac{c_v}{R} \frac{\pi'}{\pi_0}. \tag{A23}$$

In our implementation we used $\rho'(x, t=0) = c_v \rho_0 (A21)/(R\pi_0)$ and determined $\theta'(x, t=0)$ from Eq. (A23).

Of course, the solution outlined above holds only until the wave front, initially at $x = b_2$, impinges on the solid boundaries, either the model bottom or top, and is reflected. Therefore, the integration time for this test case is limited.

The above solution is what an observer would see in the absolute frame. To transform it into the rotating frame every point $\boldsymbol{X}$ (including $\boldsymbol{B}$) has to be rotated according to the formula

$$\boldsymbol{X}(t) = \boldsymbol{A} + \exp(-\mathbf{W}t) \cdot [\boldsymbol{X}(t=0) - \boldsymbol{A}], \tag{A24}$$





which follows immediately from Eq. (A3). Alternatively, the formulation (A7) could be used. Equation (A24) is the coordinate-independent formulation of the rotation. It has to be measured in the coordinate system of choice for the application in practice. As shown in Eq. (A8) the shape of the sound wave relative to $\boldsymbol{B}(t)$ is unaffected by the rotation.

Finally, we will give a short outline for the implementation of this test case into a model. First of all, the gravitational acceleration $g$ has to be set to zero, or at least to a relatively small value. If rotation is switched on ($\boldsymbol{\Omega} \neq 0$), it is essential to take into account the centrifugal acceleration $\boldsymbol{\Omega} \times [\boldsymbol{\Omega} \times (\boldsymbol{X} - \boldsymbol{A})]$ explicitly in the dynamical core, as shown in Eq. (A1). If desired, the radius and angular velocity of the Earth are rescaled, $a \to \eta_1 a$ and $\Omega \to \eta_2 \Omega$, and the topography is set to zero. If possible, the slip boundary condition should be applied at model bottom and top. All physics parameterizations should be switched off. After specifying the background state of the atmosphere (at rest in the absolute frame) by $T_0$ and $p_0$, and the sound wave parameters $\boldsymbol{B}$, $b_1$, $b_2$, $\delta v$ or $\delta T$, and $n$, the dynamic fields of the (dry) atmosphere are initialized according to

$$\boldsymbol{v}(\boldsymbol{X}) = -\eta_2 \boldsymbol{\Omega} \times (\boldsymbol{X} - \boldsymbol{A}) + v'(x) \frac{(\boldsymbol{X} - \boldsymbol{B})}{x}, \tag{A25}$$

$$\pi(\boldsymbol{X}) = \pi_0 + \pi'(x), \tag{A26}$$

$$\theta(\boldsymbol{X}) = \theta_0 + \theta'(x), \tag{A27}$$

$$\rho(\boldsymbol{X}) = \rho_0 + \rho'(x). \tag{A28}$$

In order to keep the non-linear dynamics of the sound wave as simulated by the dynamical core negligibly small, the non-dimensional wind amplitude $\delta v / c_s = (c_v/R)(\delta T/T_0)$ should be small enough. Apart from that, we adhered closely to the guidelines given by Baldauf et al. (2014) in our implementation of the test case into UA-ICON.

**Appendix B: Parcel-dynamical stability analysis**

For a thorough derivation of the parcel-dynamical stability analysis as we used it in section 3.1.2, we refer to Ertel et al. (1941) (compare also Shutts and Cullen, 1987). Here, we confine ourselves to a mathematically less rigorous sketch of the derivation in Ertel et al. (1941). We start with the momentum budget of a fluid parcel to which a displacement is applied, denoted by the operator $\delta$ (without specifying the cause of this displacement)

$$\rho \frac{\mathrm{d}^2 \boldsymbol{r}}{\mathrm{d}t^2} + 2\mathbf{W} \cdot \rho \frac{\mathrm{d}\boldsymbol{r}}{\mathrm{d}t} + \rho \boldsymbol{\nabla} \phi_g + \boldsymbol{\nabla} p = 0 \quad \Big| \, \delta \tag{B1}$$

$$\Rightarrow \quad \rho \frac{\mathrm{d}^2 \delta\boldsymbol{r}}{\mathrm{d}t^2} + 2\mathbf{W} \cdot \rho \frac{\mathrm{d}\delta\boldsymbol{r}}{\mathrm{d}t} + \rho \delta(\boldsymbol{\nabla} \phi_g) + \delta(\boldsymbol{\nabla} p) + \delta\rho \underbrace{\left( \frac{\mathrm{d}^2 \boldsymbol{r}}{\mathrm{d}t^2} + 2\mathbf{W} \cdot \frac{\mathrm{d}\boldsymbol{r}}{\mathrm{d}t} + \boldsymbol{\nabla} \phi_g \right)}_{= -\boldsymbol{\nabla} p/\rho} = 0$$

$$\Leftrightarrow \quad \rho \frac{\mathrm{d}^2 \delta\boldsymbol{r}}{\mathrm{d}t^2} + 2\mathbf{W} \cdot \rho \frac{\mathrm{d}\delta\boldsymbol{r}}{\mathrm{d}t} + \rho \delta(\boldsymbol{\nabla} \phi_g) + \delta(\boldsymbol{\nabla} p) - \frac{\delta\rho}{\rho} \boldsymbol{\nabla} p = 0, \tag{B2}$$

where $\mathbf{W} = \boldsymbol{\Omega} \times \mathbf{1}$ denotes the constant Coriolis tensor, which is the cross product of the angular velocity vector $\boldsymbol{\Omega}$ and the identity tensor $\mathbf{1}$ (see Eq. (A2)). It is antisymmetric, i.e. $\mathbf{W}^\mathsf{T} = -\mathbf{W}$, where $(\cdot)^\mathsf{T}$ denotes the transpose. In addition, we used that the displacement operator $\delta$ and the material derivative operator $\mathrm{d}/\mathrm{d}t$ commute, since the latter is parcel-fixed and should




be unaffected by the displacement. In the second step, we assumed that Eq. (B1) still holds for the parcel to the extent that we can insert $[(\text{B1}) - \boldsymbol{\nabla} p]/\rho$ into the term with the factor $\delta\rho$. This is a somewhat vague step, but we use it as a short cut to the same final result as obtained by Ertel et al. (1941) in a more elaborate line of argumentation. For the changes in the gradients of the geopotential and the pressure as observed by the displaced parcel, the ansatzes $\delta(\boldsymbol{\nabla}\chi) = \boldsymbol{\nabla}\boldsymbol{\nabla}\chi \cdot \delta\boldsymbol{r}$ are used, where $\chi = \phi_g, p$

and $\boldsymbol{\nabla}\boldsymbol{\nabla}$ denotes the Hessian, a tensor of second order. In addition, the displacement is assumed to be an adiabatic change of state, so that $(\delta\rho/\rho) = (1/\kappa)(\delta p/p)$, with $\kappa = c_p/c_v$. Finally, the pressure of the parcel shall adjust to its new environment, i.e. $\delta p = \boldsymbol{\nabla} p \cdot \delta\boldsymbol{r}$. This yields for Eq. (B2)

$$\left\{ \mathbf{1}\frac{\mathrm{d}^2}{\mathrm{d}t^2} + 2\mathbf{W}\frac{\mathrm{d}}{\mathrm{d}t} + \underbrace{\left[ \boldsymbol{\nabla}\boldsymbol{\nabla}\phi_g + \frac{1}{\rho}\left( \boldsymbol{\nabla}\boldsymbol{\nabla} p - \frac{1}{\kappa p}(\boldsymbol{\nabla} p)(\boldsymbol{\nabla} p) \right) \right]}_{=\mathbf{S}} \right\} \cdot \delta\boldsymbol{r} = 0, \tag{B3}$$

where we divided Eq. (B2) by $\rho$. Furthermore, $(\boldsymbol{\nabla} p)(\boldsymbol{\nabla} p)$ is the dyadic product of the pressure gradient $\boldsymbol{\nabla} p$ with itself, and $\mathbf{S}$

is called the stability tensor. It is a symmetric tensor, i.e. $\mathbf{S}^\mathsf{T} = \mathbf{S}$.

Eq. (B3) is of the form $\mathbf{A} \cdot \boldsymbol{b} = 0$, where $\mathbf{A}$ denotes a given tensor (operator) and $\boldsymbol{b}$ is a variable vector. In the following, we will use that solutions $\boldsymbol{b} \neq 0$ require the determinant of $\mathbf{A}$, denoted by $|\mathbf{A}|$, to vanish

$$\mathbf{A} \cdot \boldsymbol{b} = 0 \quad |\widetilde{\mathbf{A}}\cdot \quad \Rightarrow \quad \widetilde{\mathbf{A}} \cdot \mathbf{A} \cdot \boldsymbol{b} = |\mathbf{A}|\boldsymbol{b} = |\mathbf{A}|\mathbf{1} \cdot \boldsymbol{b} = 0$$

$$\Rightarrow \quad \widetilde{\mathbf{A}} \cdot \mathbf{A} = |\mathbf{A}|\mathbf{1} \quad \Big|(\cdot)_\bullet \quad \Rightarrow \quad (\widetilde{\mathbf{A}} \cdot \mathbf{A})_\bullet = \widetilde{\mathbf{A}} \cdot\cdot \mathbf{A} = |\mathbf{A}|\mathbf{1}_\bullet = |\mathbf{A}|3$$

$$\Rightarrow \quad |\mathbf{A}| = \frac{1}{3}\widetilde{\mathbf{A}} \cdot\cdot \mathbf{A}, \tag{B4}$$

where $\widetilde{\mathbf{A}}$ denotes the adjugate tensor of $\mathbf{A}$, $(\cdot)_\bullet$ is the scalar of a tensor[6], and $\cdot\cdot$ denotes the double scalar product of two tensors[7]. The adjugate is given by (compare Wilson, 1929)

$$\widetilde{\mathbf{A}} = \frac{1}{2}\left( \mathbf{A}_\times^\times \mathbf{A} \right)^\mathsf{T} = \frac{1}{2}\mathbf{A}^\mathsf{T}{}_\times^\times \mathbf{A}^\mathsf{T}, \tag{B5}$$

where $_\times^\times$ denotes the double cross product of two tensors (see footnote 7). Now, our goal is, to find the determinant of the

operator on the left-hand side of Eq. (B3), $|\{\cdot\}|$, in order to simplify the analysis of that equation. For this purpose we abbreviate

---

[6]Given a dyadic product $\boldsymbol{ab}$ of the two arbitrary vectors $\boldsymbol{a}$ and $\boldsymbol{b}$, its scalar is defined as (Wilson, 1929; Zdunkowski and Bott, 2003): $(\boldsymbol{ab})_\bullet = \boldsymbol{a} \cdot \boldsymbol{b} = \boldsymbol{b} \cdot \boldsymbol{a} = [(\boldsymbol{ab})^\mathsf{T}]_\bullet$. So the scalar of an arbitrary tensor $\mathbf{A}$, measured in some normal basis $\boldsymbol{e_i}$, $i = 1, 2, 3$: $\mathbf{A} = A_{ij}\boldsymbol{e_i}\boldsymbol{e_j}$, reads: $(\mathbf{A})_\bullet = A_{ij}(\boldsymbol{e_i}\boldsymbol{e_j})_\bullet = A_{ij}\boldsymbol{e_i} \cdot \boldsymbol{e_j} = A_{ii}$. If the coefficients of the measurement of the tensor in some coordinate system, $A_{ij}$, are arranged in an algebraic matrix structure, the value of the scalar of $\mathbf{A}$ is equal to the trace of that matrix. For the scalar of the identity tensor, we find: $(\mathbf{1})_\bullet = (\boldsymbol{e_i}\boldsymbol{e_i})_\bullet = 1 + 1 + 1 = 3$.

[7] Given two arbitrary dyadic products $\boldsymbol{ab}$ and $\boldsymbol{cd}$, typically two kinds of double products can be found in the literature (Wilson, 1929; Pichler, 1984; Sihvola, 1999; Lebedev et al., 2010; Zdunkowski and Bott, 2003; Altenbach, 2012): $\boldsymbol{ab} \overset{\bullet}{\circ} \boldsymbol{cd} = (\boldsymbol{b} \bullet \boldsymbol{c})(\boldsymbol{a} \circ \boldsymbol{d})$ and $\boldsymbol{ab}{}_\circ^\bullet \boldsymbol{cd} = (\boldsymbol{a} \bullet \boldsymbol{c})(\boldsymbol{b} \circ \boldsymbol{d})$, where $\bullet, \circ \in \{\cdot, \times\}$. Depending on the combination, the product is either a scalar (for $\bullet \to \cdot$ and $\circ \to \cdot$), a vector (for $\bullet \to \cdot$ and $\circ \to \times$, or vice versa), or a dyadic product (for $\bullet \to \times$ and $\circ \to \times$). Here, we make use of the double scalar product of the first kind: $\boldsymbol{ab} \cdot\cdot \boldsymbol{cd} = (\boldsymbol{b} \cdot \boldsymbol{c})(\boldsymbol{a} \cdot \boldsymbol{d}) = (\boldsymbol{a} \cdot \boldsymbol{d})(\boldsymbol{b} \cdot \boldsymbol{c}) = \boldsymbol{ba} \cdot\cdot \boldsymbol{dc} = (\boldsymbol{ab})^\mathsf{T} \cdot\cdot (\boldsymbol{cd})^\mathsf{T} = (\boldsymbol{d} \cdot \boldsymbol{a})(\boldsymbol{c} \cdot \boldsymbol{b}) = \boldsymbol{cd} \cdot\cdot \boldsymbol{ab}$. For the scalar of the product of two dyadic products, we find: $(\boldsymbol{ab} \cdot \boldsymbol{cd})_\bullet = [\boldsymbol{a}(\boldsymbol{b} \cdot \boldsymbol{c})\boldsymbol{d}]_\bullet = (\boldsymbol{b} \cdot \boldsymbol{c})(\boldsymbol{ad})_\bullet = (\boldsymbol{b} \cdot \boldsymbol{c})(\boldsymbol{a} \cdot \boldsymbol{d}) = \boldsymbol{ab} \cdot\cdot \boldsymbol{cd}$. In addition, we need the double cross product of the second kind: $\boldsymbol{ab}{}_\times^\times \boldsymbol{cd} = (\boldsymbol{a} \times \boldsymbol{c})(\boldsymbol{b} \times \boldsymbol{d}) = (-\boldsymbol{c} \times \boldsymbol{a})(-\boldsymbol{d} \times \boldsymbol{b}) = \boldsymbol{cd}{}_\times^\times \boldsymbol{ab}$. If we transpose the factors, we find: $(\boldsymbol{ab})^\mathsf{T}{}_\times^\times (\boldsymbol{cd})^\mathsf{T} = \boldsymbol{ba}{}_\times^\times \boldsymbol{dc} = (\boldsymbol{b} \times \boldsymbol{d})(\boldsymbol{a} \times \boldsymbol{c}) = (\boldsymbol{ab}{}_\times^\times \boldsymbol{cd})^\mathsf{T}$. The above applies to tensors as well, since they can be expanded in a sum of dyadic products, e.g., $\mathbf{A} = \boldsymbol{a_i}\boldsymbol{b_i}$.





the material derivative by $D = d/dt$. Since the material derivative in Eq. (B3) acts solely on the displacement $\delta\boldsymbol{r}$, but not on the coefficients in $\{\cdot\}$, we can treat $D$ as a scalar in the following derivation. We start with the adjugate of $\{\cdot\}$

$$
\begin{aligned}
\widetilde{\{\cdot\}} &= \frac{1}{2}\left(D^2\mathbf{1}^{\mathsf{T}} + 2D\mathbf{W}^{\mathsf{T}} + \mathbf{S}^{\mathsf{T}}\right) \overset{\times}{\times} \left(D^2\mathbf{1}^{\mathsf{T}} + 2D\mathbf{W}^{\mathsf{T}} + \mathbf{S}^{\mathsf{T}}\right) = \frac{1}{2}\left(D^2\mathbf{1} - 2D\mathbf{W} + \mathbf{S}\right) \overset{\times}{\times} \left(D^2\mathbf{1} - 2D\mathbf{W} + \mathbf{S}\right) \\
&= \frac{1}{2}D^4\mathbf{1}\overset{\times}{\times}\mathbf{1} - 2D^3\mathbf{1}\overset{\times}{\times}\mathbf{W} + D^2\mathbf{1}\overset{\times}{\times}\mathbf{S} + 2D^2\mathbf{W}\overset{\times}{\times}\mathbf{W} - 2D\mathbf{S}\overset{\times}{\times}\mathbf{W} + \frac{1}{2}\mathbf{S}\overset{\times}{\times}\mathbf{S},
\end{aligned}
\tag{B6}
$$

where we have used that $\mathbf{1}$ and $\mathbf{S}$ are symmetric tensors, and $\mathbf{W}$ is an antisymmetric tensor in the second step, and the commutativity of the double cross product in the third step (e.g., $\mathbf{S}\overset{\times}{\times}\mathbf{W} = \mathbf{W}\overset{\times}{\times}\mathbf{S}$). In the following, we use a number of identities, the proof of which is beyond the scope of this work. We have to refer to the literature cited in footnote 7, but typically they can be proven directly using, for instance, the expansion of a tensor in a sum of (three) dyadic products, the laws of associativity and commutativity of the respective products, the expansion $\boldsymbol{a} \times (\boldsymbol{b} \times \boldsymbol{c}) = \boldsymbol{b}\boldsymbol{a} \cdot \boldsymbol{c} - \boldsymbol{c}\boldsymbol{a} \cdot \boldsymbol{b}$, and $\boldsymbol{a} \cdot (\boldsymbol{b} \times \boldsymbol{c}) =$

$\boldsymbol{c} \cdot (\boldsymbol{a} \times \boldsymbol{b}) = \boldsymbol{b} \cdot (\boldsymbol{c} \times \boldsymbol{a})$. Since the determinant of the identity tensor is 1, it follows from Eqs. (B4) and (B5) that $(1/2)\mathbf{1}\overset{\times}{\times}\mathbf{1} = \mathbf{1}$. In addition, $\mathbf{1}\overset{\times}{\times}\mathbf{W} = \mathbf{W}_{\bullet}\mathbf{1} - \mathbf{W}^{\mathsf{T}} = \mathbf{W}$ holds, where we have used that $\mathbf{W}$ is antisymmetric and $\mathbf{W}_{\bullet} = (\mathbf{W}^{\mathsf{T}})_{\bullet} = -\mathbf{W}_{\bullet} = 0$. Likewise $\mathbf{1}\overset{\times}{\times}\mathbf{S} = \mathbf{S}_{\bullet}\mathbf{1} - \mathbf{S}^{\mathsf{T}} = \mathbf{S}_{\bullet}\mathbf{1} - \mathbf{S}$, using the symmetry of $\mathbf{S}$. Finally, $\mathbf{W}\overset{\times}{\times}\mathbf{W} = (\boldsymbol{\Omega} \times \mathbf{1})\overset{\times}{\times}(\boldsymbol{\Omega} \times \mathbf{1}) = 2\boldsymbol{\Omega}\boldsymbol{\Omega}$. If we insert them in Eq. (B6), we find

$$
\widetilde{\{\cdot\}} = D^4\mathbf{1} - 2D^3\mathbf{W} + D^2\left(\mathbf{S}_{\bullet}\mathbf{1} - \mathbf{S}\right) + 4D^2\boldsymbol{\Omega}\boldsymbol{\Omega} - 2D\mathbf{S}\overset{\times}{\times}\mathbf{W} + \frac{1}{2}\mathbf{S}\overset{\times}{\times}\mathbf{S}.
\tag{B7}
$$

Following Eq. (B4), the determinant reads then

$$
\begin{aligned}
|\{\cdot\}| &= \frac{1}{3}\left[D^4\mathbf{1} - 2D^3\mathbf{W} + D^2\left(\mathbf{S}_{\bullet}\mathbf{1} - \mathbf{S}\right) + 4D^2\boldsymbol{\Omega}\boldsymbol{\Omega} - 2D\mathbf{S}\overset{\times}{\times}\mathbf{W} + \frac{1}{2}\mathbf{S}\overset{\times}{\times}\mathbf{S}\right] \cdot\cdot \left(D^2\mathbf{1} + 2D\mathbf{W} + \mathbf{S}\right) \\
&= \frac{1}{3}\left[D^6\mathbf{1}\cdot\cdot\mathbf{1} - 2D^5\mathbf{W}\cdot\cdot\mathbf{1} + D^4\left(\mathbf{S}_{\bullet}\mathbf{1}\cdot\cdot\mathbf{1} - \mathbf{S}\cdot\cdot\mathbf{1}\right) + 4D^4\boldsymbol{\Omega}\boldsymbol{\Omega}\cdot\cdot\mathbf{1} - 2D^3\left(\mathbf{S}\overset{\times}{\times}\mathbf{W}\right)\cdot\cdot\mathbf{1} + \frac{1}{2}D^2\left(\mathbf{S}\overset{\times}{\times}\mathbf{S}\right)\cdot\cdot\mathbf{1} \right. \\
&\quad + 2D^5\mathbf{1}\cdot\cdot\mathbf{W} - 4D^4\mathbf{W}\cdot\cdot\mathbf{W} + 2D^3\left(\mathbf{S}_{\bullet}\mathbf{1}\cdot\cdot\mathbf{W} - \mathbf{S}\cdot\cdot\mathbf{W}\right) + 8D^3\boldsymbol{\Omega}\boldsymbol{\Omega}\cdot\cdot\mathbf{W} - 4D^2\left(\mathbf{S}\overset{\times}{\times}\mathbf{W}\right)\cdot\cdot\mathbf{W} + D\left(\mathbf{S}\overset{\times}{\times}\mathbf{S}\right)\cdot\cdot\mathbf{W} \\
&\quad \left. + D^4\mathbf{1}\cdot\cdot\mathbf{S} - 2D^3\mathbf{W}\cdot\cdot\mathbf{S} + D^2\left(\mathbf{S}_{\bullet}\mathbf{1}\cdot\cdot\mathbf{S} - \mathbf{S}\cdot\cdot\mathbf{S}\right) + 4D^2\boldsymbol{\Omega}\boldsymbol{\Omega}\cdot\cdot\mathbf{S} - 2D\left(\mathbf{S}\overset{\times}{\times}\mathbf{W}\right)\cdot\cdot\mathbf{S} + \frac{1}{2}\left(\mathbf{S}\overset{\times}{\times}\mathbf{S}\right)\cdot\cdot\mathbf{S}\right].
\end{aligned}
\tag{B8}
$$

Here, we can use the identity $\mathbf{A}\cdot\cdot\mathbf{1} = (\mathbf{A}\cdot\mathbf{1})_{\bullet} = \mathbf{A}_{\bullet}$, which yields $\mathbf{1}\cdot\cdot\mathbf{1} = 3$, $\mathbf{S}\cdot\cdot\mathbf{1} = \mathbf{S}_{\bullet}$, $\mathbf{W}\cdot\cdot\mathbf{1} = \mathbf{W}_{\bullet} = 0$, $\boldsymbol{\Omega}\boldsymbol{\Omega}\cdot\cdot\mathbf{1} = \boldsymbol{\Omega}\cdot\boldsymbol{\Omega} = \boldsymbol{\Omega}^2$, and $(\mathbf{S}\overset{\times}{\times}\mathbf{W})\cdot\cdot\mathbf{1} = 0$, since $\mathbf{S}\overset{\times}{\times}\mathbf{W}$ is an antisymmetric tensor according to $(\mathbf{S}\overset{\times}{\times}\mathbf{W})^{\mathsf{T}} = \mathbf{S}^{\mathsf{T}}\overset{\times}{\times}\mathbf{W}^{\mathsf{T}} = -\mathbf{S}\overset{\times}{\times}\mathbf{W}$. In addition, $\mathbf{W}\cdot\cdot\mathbf{W} = -2\boldsymbol{\Omega}^2$, $\mathbf{S}\cdot\cdot\mathbf{W} = \mathbf{S}^{\mathsf{T}}\cdot\cdot\mathbf{W}^{\mathsf{T}} = -\mathbf{S}\cdot\cdot\mathbf{W} = 0$, and likewise $\boldsymbol{\Omega}\boldsymbol{\Omega}\cdot\cdot\mathbf{W} = 0$, $(\mathbf{S}\overset{\times}{\times}\mathbf{S})\cdot\cdot\mathbf{W} = 0$ (since $\mathbf{S}\overset{\times}{\times}\mathbf{S}$ is symmetric), $(\mathbf{S}\overset{\times}{\times}\mathbf{W})\cdot\cdot\mathbf{S} = 0$. Furthermore, $(\mathbf{S}\overset{\times}{\times}\mathbf{S})\cdot\cdot\mathbf{1} = (\mathbf{S}\overset{\times}{\times}\mathbf{S})_{\bullet} = \mathbf{S}_{\bullet}^2 - \mathbf{S}\cdot\cdot\mathbf{S}$, and $(\mathbf{S}\overset{\times}{\times}\mathbf{W})\cdot\cdot\mathbf{W} = -2\boldsymbol{\Omega}\boldsymbol{\Omega}\cdot\cdot\mathbf{S}$, and finally $(1/6)(\mathbf{S}\overset{\times}{\times}\mathbf{S})\cdot\cdot\mathbf{S} = |\mathbf{S}|$. Altogether, this yields for Eq. (B8)

$$
|\{\cdot\}| = D^6 + D^4\left(\mathbf{S}_{\bullet} + 4\boldsymbol{\Omega}^2\right) + D^2\left[\frac{1}{2}\left(\mathbf{S}_{\bullet}^2 - \mathbf{S}\cdot\cdot\mathbf{S}\right) + 4\boldsymbol{\Omega}\boldsymbol{\Omega}\cdot\cdot\mathbf{S}\right] + |\mathbf{S}|
\tag{B9}
$$

and we find for the transformation $\widetilde{\{\cdot\}} \cdot$ (B3)

$$
\left\{\frac{d^6}{dt^6} + \underbrace{\left(\mathbf{S}_{\bullet} + 4\boldsymbol{\Omega}^2\right)}_{=a}\frac{d^4}{dt^4} + \underbrace{\left[\frac{1}{2}\left(\mathbf{S}_{\bullet}^2 - \mathbf{S}\cdot\cdot\mathbf{S}\right) + 4\boldsymbol{\Omega}\boldsymbol{\Omega}\cdot\cdot\mathbf{S}\right]}_{=b}\frac{d^2}{dt^2} + \underbrace{|\mathbf{S}|}_{=c}\right\}\delta\boldsymbol{r} = 0.
\tag{B10}
$$





Finally, we summarize some aspects of the computation of the coefficients $a$, $b$ and $c$ in geographical spherical coordinates. Measuring $\mathbf{S}$ in the zonal, meridional and vertical unit vectors $\boldsymbol{e_\lambda}$, $\boldsymbol{e_\varphi}$ and $\boldsymbol{e_r}$, and arranging the coefficients of this measurement in a matrix, denoted by $(\mathbf{S})$, reads

$$(\mathbf{S}) = \begin{pmatrix} S_{\lambda\lambda} & S_{\lambda\varphi} & S_{\lambda r} \\ S_{\varphi\lambda} & S_{\varphi\varphi} & S_{\varphi r} \\ S_{r\lambda} & S_{r\varphi} & S_{rr} \end{pmatrix} = \begin{pmatrix} S_{\lambda\lambda} & S_{\lambda\varphi} & S_{\lambda r} \\ S_{\lambda\varphi} & S_{\varphi\varphi} & S_{\varphi r} \\ S_{\lambda r} & S_{\varphi r} & S_{rr} \end{pmatrix},$$ (B11)

$$S_{\lambda\lambda} = g\frac{a^2}{r^3} + \frac{1}{\rho}\left[\frac{1}{(r\cos\varphi)^2}\frac{\partial^2 p}{\partial\lambda^2} - \frac{\tan\varphi}{r^2}\frac{\partial p}{\partial\varphi} + \frac{1}{r}\frac{\partial p}{\partial r} - \frac{1}{\kappa p}\left(\frac{1}{r\cos\varphi}\frac{\partial p}{\partial\lambda}\right)^2\right],$$

$$S_{\lambda\varphi} = \frac{1}{\rho}\left[\frac{1}{r^2\cos\varphi}\frac{\partial^2 p}{\partial\lambda\partial\varphi} + \frac{\tan\varphi}{r^2\cos\varphi}\frac{\partial p}{\partial\lambda} - \frac{1}{\kappa p}\left(\frac{1}{r^2\cos\varphi}\frac{\partial p}{\partial\lambda}\frac{\partial p}{\partial\varphi}\right)\right],$$

$$S_{\lambda r} = \frac{1}{\rho}\left[\frac{1}{r\cos\varphi}\frac{\partial^2 p}{\partial\lambda\partial r} - \frac{1}{r^2\cos\varphi}\frac{\partial p}{\partial\lambda} - \frac{1}{\kappa p}\left(\frac{1}{r\cos\varphi}\frac{\partial p}{\partial\lambda}\frac{\partial p}{\partial r}\right)\right],$$

$$S_{\varphi\varphi} = g\frac{a^2}{r^3} + \frac{1}{\rho}\left[\frac{1}{r^2}\frac{\partial^2 p}{\partial\varphi^2} + \frac{1}{r}\frac{\partial p}{\partial r} - \frac{1}{\kappa p}\left(\frac{1}{r}\frac{\partial p}{\partial\varphi}\right)^2\right],$$

$$S_{\varphi r} = \frac{1}{\rho}\left[\frac{1}{r}\frac{\partial^2 p}{\partial\varphi\partial r} - \frac{1}{r^2}\frac{\partial p}{\partial\varphi} - \frac{1}{\kappa p}\left(\frac{1}{r}\frac{\partial p}{\partial\varphi}\frac{\partial p}{\partial r}\right)\right],$$

$$S_{rr} = -2g\frac{a^2}{r^3} + \frac{1}{\rho}\left[\frac{\partial^2 p}{\partial r^2} - \frac{1}{\kappa p}\left(\frac{\partial p}{\partial r}\right)^2\right],$$

where we used the symmetry of $\mathbf{S}$. In addition, $(2\boldsymbol{\Omega}) = (2\Omega_\lambda, 2\Omega_\varphi, 2\Omega_r) = (0, 2\Omega\cos\varphi, 2\Omega\sin\varphi) = (0, f_\varphi, f_r)$, where $\Omega = |\boldsymbol{\Omega}|$. Now, the coefficients can be computed according to

$$a = S_{\lambda\lambda} + S_{\varphi\varphi} + S_{rr} + 4\Omega^2$$ (B12)

$$b = S_{\lambda\lambda}S_{\varphi\varphi} - S_{\lambda\varphi}^2 + S_{\varphi\varphi}S_{rr} - S_{\varphi r}^2 + S_{\lambda\lambda}S_{rr} - S_{\lambda r}^2 + f_\varphi^2 S_{\varphi\varphi} + f_r^2 S_{rr} + 2f_\varphi f_r S_{\varphi r}.$$ (B13)

15 The coefficient $c$ is the determinant of $(\mathbf{S})$ and can be computed from a Laplace expansion or some other rule for the computation of the determinant of a $3 \times 3$ matrix (e.g. Bronstein et al., 2001).

According to the considerations of Thuburn and White (2013) an application of the above ansatz to the shallow-atmosphere equations might be less straight forward as it may seem. Although the effects may be small, the shallow-atmosphere approximation changes, for instance, the metric of the space into a non-Euclidian one. So the concept of a parcel displacement may

20 be reassessed to some extent. Here, we pass over such considerations and take a simple approach by applying the shallow-atmosphere approximations, summarized for instance in White et al. (2005), directly to the components of $(\{\cdot\})$ in Eq. (B3). However, we emphasize that other approaches are quite possible. To start with, $\boldsymbol{\nabla}\phi_g = \text{const.}$ holds, so that $\boldsymbol{\nabla}\boldsymbol{\nabla}\phi_g = 0$. For the approximation of $(\boldsymbol{\nabla}\boldsymbol{\nabla}p)$ and $((\boldsymbol{\nabla}p)(\boldsymbol{\nabla}p))$, we set $r = a + z$, replace the inverse radius $1/r$ by the inverse mean radius of



the Earth $1/a$, and $\partial/\partial r$ by $\partial/\partial z$. The metric terms in $(\nabla\nabla p)$ are neglected, apart from the terms involving $\tan\varphi$. We find

$$
(\mathbf{S}) = \begin{pmatrix} S_{\lambda\lambda} & S_{\lambda\varphi} & S_{\lambda z} \\ S_{\lambda\varphi} & S_{\varphi\varphi} & S_{\varphi z} \\ S_{\lambda z} & S_{\varphi z} & S_{zz} \end{pmatrix}, \tag{B14}
$$

$$
S_{\lambda\lambda} = \frac{1}{\rho}\left[ \frac{1}{(a\cos\varphi)^2}\frac{\partial^2 p}{\partial\lambda^2} - \frac{\tan\varphi}{a^2}\frac{\partial p}{\partial\varphi} - \frac{1}{\kappa p}\left(\frac{1}{a\cos\varphi}\frac{\partial p}{\partial\lambda}\right)^2 \right],
$$

$$
S_{\lambda\varphi} = \frac{1}{\rho}\left[ \frac{1}{a^2\cos\varphi}\frac{\partial^2 p}{\partial\lambda\partial\varphi} + \frac{\tan\varphi}{a^2\cos\varphi}\frac{\partial p}{\partial\lambda} - \frac{1}{\kappa p}\left(\frac{1}{a^2\cos\varphi}\frac{\partial p}{\partial\lambda}\frac{\partial p}{\partial\varphi}\right) \right],
$$

$$
S_{\lambda z} = \frac{1}{\rho}\left[ \frac{1}{a\cos\varphi}\frac{\partial^2 p}{\partial\lambda\partial z} - \frac{1}{\kappa p}\left(\frac{1}{a\cos\varphi}\frac{\partial p}{\partial\lambda}\frac{\partial p}{\partial z}\right) \right],
$$

$$
S_{\varphi\varphi} = \frac{1}{\rho}\left[ \frac{1}{a^2}\frac{\partial^2 p}{\partial\varphi^2} - \frac{1}{\kappa p}\left(\frac{1}{a}\frac{\partial p}{\partial\varphi}\right)^2 \right],
$$

$$
S_{\varphi z} = \frac{1}{\rho}\left[ \frac{1}{a}\frac{\partial^2 p}{\partial\varphi\partial z} - \frac{1}{\kappa p}\left(\frac{1}{a}\frac{\partial p}{\partial\varphi}\frac{\partial p}{\partial z}\right) \right],
$$

$$
S_{zz} = \frac{1}{\rho}\left[ \frac{\partial^2 p}{\partial z^2} - \frac{1}{\kappa p}\left(\frac{\partial p}{\partial z}\right)^2 \right].
$$

Conbined with $(2\boldsymbol{\Omega}) = (0,0,2\Omega\sin\varphi) = (0,0,f_z)$, the coefficients can be computed according to

$$
a = S_{\lambda\lambda} + S_{\phi\phi} + S_{zz} + f_z^2 \tag{B15}
$$

$$
b = S_{\lambda\lambda}S_{\phi\phi} - S_{\lambda\phi}^2 + S_{\phi\phi}S_{zz} - S_{\phi z}^2 + S_{\lambda\lambda}S_{zz} - S_{\lambda z}^2 + f_z^2 S_{zz}. \tag{B16}
$$

In section 3.1.2, we apply the ansatz to a zonally symmetric atmospheric state, so we can neglect all terms in $(\mathbf{S})$ that contain zonal derivatives $\partial/\partial\lambda$.

*Author contributions.* The upper-atmosphere extension of the ICON model was a close collaboration between the authors. The focus of G. Zhou and H. Schmidt was on the adaption, implementation and evaluation of the upper-atmosphere physics parameterizations. The focus of S. Borchert, M. Baldauf, G. Zängl and D. Reinert was on the implementation and evaluation of the deep-atmosphere modifications to the dynamical core.

*Competing interests.* The authors declare that they have no conflict of interest.

*Acknowledgements.* The authors are grateful to the entire ICON development team (MPI-M, DWD, and Karlsruhe Institute of Technology). We greatly appreciate many helpful discussions with Dr. Elisa Manzini (MPI-M) on aspects of modeling the middle atmosphere. Special thanks go to Drs. Florian Prill (DWD), Marco A. Giorgetta (MPI-M), and Sebastian Rast (MPI-M) for many helpful and illuminating





discussions on the dynamical core, the physics packages, and the code infrastructure of the ICON model. S.B., G.Zä., M.B., G.Zh. and H.S. thank the German Research Foundation (DFG) for partial support through the research unit Multiscale Dynamics of Gravity Waves (MS-GWaves) and through grants ZA 268/10-1 and SCHM 2158/5-1.





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
