# Peer review of "The upper-atmosphere extension of the ICON general circulation model"

_Geoscientific Model Development, 2018_

## Short Comment (SC1) · 9 Jan 2019

Dear authors,

in my role as Executive editor of GMD, I would like to bring to your attention our Editorial version 1.1:

http://www.geosci-model-dev.net/8/3487/2015/gmd-8-3487-2015.html

This highlights some requirements of papers published in GMD, which is also available on the GMD website in the 'Manuscript Types' section:

http://www.geoscientific-model-development.net/submission/manuscript_types.html

In particular, please note that for your paper, the following requirement has not been

met in the Discussions paper:

- "The main paper must give the model name and version number (or other unique identifier) in the title."

Please add a version number for ICON in the title upon your revised submission to GMD.

Yours,

Astrid Kerkweg

―――――――――――――――――

---

## Referee Comment (RC1) · Anonymous Referee #1 · 28 Jan 2019

**General comments**

The manuscript describes modifications of the dynamical core and physical parameterizations in the ICON general circulation model, which allows to extend the model domain to the lower thermosphere up to 150 km. In the first part the authors describe the changes in the dynamical core and the results of two idealized tests with great details. The climatological tests aimed at the evaluation of the overall model performance and the influence of two major modifications are described in the second part with much less details. The manuscript is in the scope of GMD and will be useful for climate community, because the vertical coupling of the atmosphere from the ground to the middle thermosphere is widely discussed in the recent publications. The number of appropriate models for this kind of studies is very limited and the appearance

of new model is very welcome. However, the structure and some disproportion in the manuscript make it rather difficult to read. The first part is interesting mainly to the developers of dynamical cores because the used experimental set up is very idealized and cannot be easily applied to the modeling of the real atmospheric processes. The second part is more interesting for the climate community, but in the present for it is too sketchy and does not convince readers that model is ready for operational use. The authors are constantly mentioning substantial biases in the zonal wind and temperature distributions and even state that they know how to improve the model in the future. It suggests that the model is not mature enough to be recommended as a tool for community. Of course, this aspect will make the manuscript less important and readable. If it is ok with authors I will not strongly insist on any major changes. However, I would advise to keep in the manuscript only the first part and the analysis of the climatological runs ICON, ICON-UA and ICON(UA), which makes possible to show the influence of introduced changes in dynamical core and physical parameterizations. Maybe it is even better to concentrate on dynamical core (with proper changes of the title). The model evaluation should be better postponed because more careful model tuning and large set of considered variables are necessary. This should be done before the publication of the manuscript, otherwise this important contribution will not be fully appreciated.

Major issues

1. The manuscript is too long and rather difficult to read. I can propose some changes (see in general comments), but I do not strongly insist on their implementation.

2. Introduction: Some review of the existing models is a must. The new model development should be considered in the context of the existing tools.

3. The atmospheric state is not just temperature and zonal wind. There are much more parameters to evaluate for the complete understanding how the model represents atmospheric processes. The most important processes were defined during several
model intercomparison campaigns. In the case when the upper atmosphere is involved this list can be extended by additional parameters (e.g., tidal waves).

Minor issues:

1. page 1, line 10: what is satisfactory and good agreement? I would not characterize major bias in the stratospheric zonal wind as a good agreement.

2. page 4, line 6: electronically? Electrically sounds better.

3. page 5, line 1: H is not constant. Is it considered in the described example?

4. page 5, first paragraph: I do not completely understand the message.

5. page 13, first paragraph: Why not to use HAMMONIA output for the initialization?

6. page 15, table 1: Any reason not to use HAMMONIA approach (Solomon and Qian, 2005).

7. page 16, line 1: I am not sure I understand the situation with GWD. Is it off in the standard ICON, but on in ICON-UA? Then the comparison between them is difficult because there are differences between models such as GWD and presence of sponge layers.

8. page 16, line 6: It is not correct. Lyman-alpha and SRB can contribute down to 60 or even 50 km.

9. page 14, line 16: why 0.23?

10. page 29, line 21-22: Unidentified bug? What is the reason to assume that this limitation is not important?

11. page 30, line 1: Would it be possible to discuss the role of damping when you compare different model versions. It can be rather dramatic.

12. page 30, line 27: Which minimum is discussed? It is not visible.
13. page 31, line 1: It would be interesting evaluate the influence of dynamical and physical processes separately.

14. page 34, line 24: Clarify good agreement. In some cases, the agreement is not so good.

15. page 35, lines 1-2: It would be interesting comment on the influence of dynamical and physical processes separately.

---

## Referee Comment (RC2) · Anonymous Referee #2 · 7 Feb 2019

Review of GMD-2018-289
**Title:** The upper-atmosphere extension of the ICON general circulation model
**Authors:** Sebastian Borchert, Guidi Zhou, Michael Baldauf , Hauke Schmidt, Günther Zängl, and Daniel Reinert

**1  General Comments**

The authors present an extension of the ICON model to the upper atmosphere. This consists of both a change to the dynamics to simulate the deep atmosphere equations and augmentation to the physical parameterizations to represent important processes to the upper atmosphere that are missing in the standard model configuration. These changes are tested on two dry dynamical core tests and a climatological test. The paper is very detailed in the discussion and analysis of the dynamical core simulations which support the conclusions of the authors, the discussion of the climatological test is less conclusive and maybe more investigation and comparison could be performed here.

The contents of this paper is certainly of interest to the wider community and suitable to GMD, my main comment however is that I found the paper to be somewhat long, which in of itself is not a problem, but it is quite wide in its scope in both providing the modifications to the dynamical core, a detailed analysis of the dynamical cores tests, as well as broader descriptions of the extension of the parameterizations and the more complex climatological test. I think this paper could therefore benefit from re-factoring somewhat and tightening up.

I think this paper could be split into two papers, the first could detail the changes to the dynamical core to simulate deep atmosphere dynamics along with the results from Section 3.1 and the appendices of which I think there is enough material in this paper to recommend publication of as it stands. The second paper would detail the specification of ICON-UA, the extension of the parameterizations (Section 2.2) along with test case in Section 3.2. The changes to model initialization (Section 2.1.3) could be relevant to both areas and it is up to hte authors where they think this would best sit.

**2  Specific Comments**

In addition to the above I have a number of more specific, minor, comments for the authors:

1. Page 8, line 2. The equation in this line contains terms $\partial/\partial t$ referring to differentiation with respect to the tangential direction, whilst equations (8) & (9) use $\partial/\partial t$ for temporal differentiation. Could the authors change either of the symbols here to avoid confusion?

2. Section 2.1.3 I think that this change to initialization is problem many forecasting centers will have moving to the deep atmosphere equations with high lids and I am glad to see the authors address this, however I found the interaction of the different profiles $(T_B, T_F, T_\infty, T_{120km}, T_{IFS})$ a little hard to follow and think this could be helped by a simple diagram showing the regions in which each profile is used and the blending regions where they overlap.

3. Table 1. It would be useful if the authors could add the height at which these different processes (which is described in the text) are used into table 1.

4. Page 20, line27: "$\Delta t = \eta_1 \cdot 13.2s$ fits both the maximum magnitude..." It's unclear to me what is being said here. Do you mean that this timestep fits any CFL restriction of the model in both configurations? or something else?

5. Figure 4. The caption states that the isolines are the analytical solution but these appear very noisy in panels (a) and (b) (particularly the zero contour) which I would not expect from the solution given in the appendices. Is this correct?

6. Page 23. Line 14 How and why has the vertical grid been changed from the Ullrich et.al. paper? (Given the results from figure 7 it appears unlikely that this will have changed the answers but if possible could the formula (or reason) used here be given?

7. Figure 7 In order to condense the paper I think this figure could be removed and it simply stated that the test appears converged with regards to vertical resolution.

8. Page 26, Line 5: Replace "ragard" with "regard"

9. Page 28 Line 25: Replace "mentionde" with "mentioned"

10. Page 29 line 22: I think the authors are correct and this limitation will not affect results greatly but it would be better if they could quantify this belief somewhat instead of just assuming

---

## Author Response (AR1)

Dear Dr. Añel,
Dear Referees,
Dear Commenter,

Thank you very much for handling our submission. A clean version of the revised manuscript will follow. Please find a version of the manuscript, in which all changes are tracked at the end of this document.
The major changes and improvements to our manuscript include:

- The title of the first version of our manuscript was missing a version specifier, as remarked by the commenter. We added this information to the title of the revised manuscript. Now, the title reads (with the addition in bold): "The upper-atmosphere extension of the ICON general circulation model **(version: ua-icon-1.0)**"

- Following a comment of referee 1, we significantly extended the review of existing general circulation models that extend into the upper atmosphere in section 1 ("Introduction") of the revised manuscript.

- Both referees remarked that the manuscript is too long. In order to reduce its length, we drastically abbreviated our arguments not to modify the dissipative terms of the governing equations for deep-atmosphere dynamics in section 2.1.1 ("Model equations"); we moved the technical details of the model initialization from section 2.1.3 ("Model initialization") to the appendix A, as suggested by referee 2; we drastically abbreviated our speculation on possible reasons for a numerical instability that occurred during one simulation of an idealized test case in section 3.1.2 ("Jablonowski-Williamson baroclinic instability test case") and removed a corresponding appendix.

- During the revision process, we found a flaw in the gas data that are required for the parameterization of the upper-atmosphere-specific radiative processes. This flaw has been removed and we repeated the climatological test case simulations presented in section 3.2, which make use of the upper-atmosphere physics (the two simulation variants named "UA-ICON" and "ICON(UA)"). The new simulation results presented in section 3.2.2 ("Comparison of simulated and observed climatologies") of the revised manuscript indicate, that the use of the corrected gas data results in an improved performance of UA-ICON (especially with regard to the mean temperature).

- Following a suggestion of referee 1, we included the results of a further climatological test case simulation, named "UAphys-ICON" in our revised manuscript. It is identical to the test case "UA-ICON", with its model top at 150 km and the upper-atmosphere physics switched on, except for the deep-atmosphere dynamics being switched off (i.e. the shallow-atmosphere dynamics are applied). Through the comparion "UAphys-ICON" $\leftrightarrow$ "UA-ICON", we tried to quantify the effects of the deep-atmosphere modification of the dynamical core of ICON. The new results are presented in section 3.2.2 ("Comparison of simulated and observed climatologies") of the revised manuscript.

- In the climatological test case simulations, whose results we presented in the first version of our manuscript, we were not able to apply the vertical variation of the gravitational acceleration as part of the deep-atmosphere dynamics, due to a numerical instability that resulted from too strong heating rates from the standard physics package. During the revision process, we were able to identify the problem in the code and removed it. The above-mentioned new simulations include the vertical variation of the gravitational acceleration, i.e. the complete deep-atmosphere modification of the dynamical core is applied.

We hope that we could address the concerns of the referees and the commenter adequately. With kind ragards, also on behalf of all co-authors,

Sebastian Borchert

**Response to the Short Comment by Dr. Kerkweg**

Dear Dr. Kerkweg,

Thank you very much for making us aware of this requirement. We changed the titel of our manuscript from "The upper-atmosphere extension of the ICON general circulation mode" to "The upper-atmosphere extension of the ICON general circulation model (version: ua-icon-1.0)" in our revised manuscript.

**Response to the Comments by Referee 1**

Dear Referee,

Thank you very much for your review. We appreciate the friendly and constructive comments. They have been very helpful in further improving the manuscript. Please find below our point-by-point response. We refer to the version of the manuscript reviewed by the Referee as "reviewed manuscript", and to the revised version as "revised manuscript". In addition, we provide a version of the manuscript, in which all changes are tracked at the end of this document. Text passages, which have been modified or deleted, are marked red and crossed out. New text is marked blue. We refer to this tracked version as "tracked manuscript" in the response below.

**Major points**

- *The manuscript is too long and rather difficult to read. I can propose some changes (see in general comments), but I do not strongly insist on their implementation.*

  We see the point of the Referee and we made several changes to the manuscript, in order to tighten the text and improve the readability. This was mainly achieved by deleting and abbreviating passages from sections 2.1 and 3.1 on the model dynamics and its evaluation, and moving some technical details to a new appendix. We hope that these measures improve the balance between the description of the upper-atmosphere physics and the deep-atmosphere dynamics. Nevertheless, we would very much prefer to keep the general structure of our manuscript with the description and evaluation of both the dynamics and the physics. We are aware that the idealized test cases in section 3.1 might appear rather technical. However, we think they are necessary to assess the performance of the dynamical core of the model. With regard to the evaluation of the upper-atmosphere physics we agree that it is certainly incomplete. However, completeness would very much depend on the scientific goals of specific future studies, and in order to keep the manuscript at an acceptable length we would prefer to stick to the evaluation of basic zonal mean fields. We have, however added a further sensitivity experiment to better understand the respective roles of upper-atmosphere modifications in model dynamics and physics. Below, we list the modifications we made to the manuscript, in order to reduce its length.

  P(age) 5 L(ine) 28 to L 31 of the reviewed manuscript:
  The reference to the entropy and potential temperature equations were made because of the remarks on the source $Q$ on P 8 L 8 to P 9 L 29 of the reviewed manuscript. These remarks have been abbreviated significantly (see subsequent modification) and the reference to the entropy and potential temperature equations is no longer necessary, so we removed it (see P 6 L 24 to L 27 of the tracked manuscript).

  P 8 L 8 to P 9 L 29 of the reviewed manuscript:
  We think this part on the dissipative and diabatic processes in Eqs. (1) and (3) has become a too exhaustive description of what we have actually not modified for the deep atmosphere in the dynamical core. For this reason we significantly tightened this part (see P 8 L 18 of the revised manuscript, and P 9 L 20 to P 11 L 17 of the tracked manuscript).

  Section 2.1.3, P 12 L 24 to P 14 L 26 of the reviewed manuscript:
  We assume that the details of the model initialization might appear rather technical. For this reason, we moved a large part of this section into a new appendix "Appendix A: Model initialization" (see P 11 L 13 and P 31 L 27 to P 34 L 6 of the revised manuscript, and P 14 L 19 to P 16 L 22 and P 44 L 20 to P 46 L 30 of the tracked manuscript).

  Figure 7, P 25 of the reviewed manuscript:
  We removed this figure on the convergence of the solution to the Jablonowski-Williamson test case with regard to the vertical grid resolution. We think, to mention in the text that the solution is largely converged with respect to the vertical resolution should be enough (see P 21 L 15 of the revised manuscript, and P 27 L 10 and P 29 of the tracked manuscript).

  P 25 L 5 (middle) to P 29 L 12 of the reviewed manuscript:
  In that passage we tried to list possible reasons for a numerical instability that occurred during testing the horizontal grid R3B7 for the Jablonowski-Williamson test case (compare Fig. 6, P 24 of the reviewed manuscript). One of the difficulties in finding the reason for this instability lies in the fact that if we switch off certain parts of our deep-atmosphere implementations in the dynamical core, in order to exclude that they contribute to the instabilty, the prescribed analytical solution for the atmospheric background state of

this test case would no longer fit to the modified dynamical core. As a consequence the model atmosphere undergoes a more or less strong adjustment process right away from the beginning of a simulation. This renders the usefulness of such simulations highly questionable. Determining new analytical solutions for the atmospheric background state for the deep-atmosphere equations with some parts switched off would be too cumbersome. To conclude, the possible reasons for the instability listed in the reviewed manuscript were speculative and still are. As such, we think they are not essential for the manuscript and we decided to remove that passage and the appendix B, P 39 to P 43 belonging to it. A note on the numerical instability and the fact that we were not yet able to clearly identify its reason remains on P 21 L 27 to P 22 L 11 of the revised manuscript (see P 28 L 9 to P 33 L 11 of the tracked manuscript).

- *Introduction: Some review of the existing models is a must. The new model development should be considered in the context of the existing tools.*

  We agree that our review of the existing models was too short. We extended our review in the introduction (see section 1, on P 4 L 8 to P 5 L 3 of the revised manuscript, and P 4 L 11 and P 5 L 6 of the tracked manuscript). The motivation for yet another upper atmosphere extension of a GCM is to benefit from specific features of the ICON also in an "entire atmosphere" context, for instance the option for very high model resolutions both globally through the efficient scaling of ICON on massively parallel computers or regionally through the nesting approach in combination with the non-hydrostatic dynamical core, and the initialization opportunities of an operational weather forecast model (see P 1 L 21 to P 2 L8 of the revised manuscript).

- *The atmospheric state is not just temperature and zonal wind. There are much more parameters to evaluate for the complete understanding how the model represents atmospheric processes. The most important processes were defined during several model intercomparison campaigns. In the case when the upper atmosphere is involved this list can be extended by additional parameters (e.g., tidal waves).*

  Yes, a thorough model evaluation would include more quality measures than the comparison of the zonally and climatologically averaged temperature and zonal wind component with satellite retrievals. Nevertheless, we regard the comparison in section 3.2.2 of the reviewed manuscript as a first important element of evaluation to build upon in the future. We decided for the temperature and the zonal wind component as the target fields of our comparison, since this seems to be a well-established practice in the community. We have, however added a further sensitivity experiment to better understand the respective roles of upper-atmosphere modifications in model dynamics and physics (see P 24 L 4, P 27 L 10 to P 29 L 11, P 30 L 30, and P 31 L 10 of the revised manuscript, and our comment on minor point 13 below).

**Minor points**

1. *P 1, L 10 of the reviewed manuscript:*
   *what is satisfactory and good agreement? I would not characterize major bias in the stratospheric zonal wind as a good agreement.*

   In the meantime we found a flaw in the data of the gases, which are required for the parameterization of the radiative processes listed in Table 1, P 15 of the reviewed manuscript. This flaw is removed, and we repeated the simulations of the two configuration variants that make use of the upper-atmosphere physics package (denoted UA-ICON and ICON(UA), see section 3.2.2, P 30 to P 33 of the reviewed manuscript). With the corrected gas data the results for the zonally averaged temperature and zonal wind component are better, when compared to the satellite retrievals, than the results presented in the reviewed manuscript (compare the reviewed and revised versions of Fig. 9 on P 35 and 36, and Fig. 10/11 on P 38 and 39 of the tracked manuscript). Apart from that, we replaced the respective formulation in the Abstract by the more neutral formulation: "A series of idealized test cases and climatological simulations is performed, in order to evaluate the upper-atmosphere extension of ICON." (see P 1 L 9 of the revised manuscript). In addition, we used somewhat more cautious formulations in section 4 (Conclusions) of the manuscript (see P 43 L 23 to L 33 of the tracked manuscript).

2. *P 4, L 6 of the reviewed manuscript:*
   *electronically? Electrically sounds better.*

   Thank you. We changed the formulation accordingly (see P 4 L 2 of the revised manuscript).

3. *P 5, L 1 of the reviewed manuscript:*
   *H is not constant. Is it considered in the described example?*

No, in this example we assume a constant H, to simplify matters. However, we removed this example from the manuscript, also because of the subsequent point 4: if this example appears rather confusing it defeats the purpose and should be removed (see P 5 L 22 of the tracked manuscript). However, we come back to this argument on P 27 L 10 to P 29 L 11 of the revised manuscript, where we discuss the difference between the UA-ICON simulation and a newly performed simulation, denoted UAphys-ICON, without deep-atmosphere dynamics (see our responses on points 13 and 15 below). There, it might fit better.

4. *P 5, first paragraph of the reviewed manuscript:*
   *I do not completely understand the message.*

   We removed the example (see the response to the preceding point 3).

5. *P 13, first paragraph of the reviewed manuscript:*
   *Why not to use HAMMONIA output for the initialization?*

   We decided for an initialization with analyses from the IFS model, as described in section 2.1.3 of the reviewed manuscript and appendix A of the revised manuscript, because it offers a high-frequency output covering a relatively long period in the past. This provides a high flexibility with regard to the start date of a simulation. If we would initialize with HAMMONIA output, we would have to perform a HAMMONIA simulation ahead of each UA-ICON simulation, in order to get the data for the desired start date. This would increase the computational costs and the total time needed for a simulation.

6. *P 15, table 1 of the reviewed manuscript:*
   *Any reason not to use HAMMONIA approach (Solomon and Qian, 2005).*

   HAMMONIA has been used in different configurations. Earlier configurations (as described by Schmidt et al., 2006) have used the approach based on Richardson et al. (1994), while in connection to the inclusion of ion chemistry and a more complex representation of the ion drag, the EUV heating was updated to the approach of Solomon and Qian. To allow for a cleaner comparison with earlier HAMMONIA simulations during the development phase of UA-ICON we stuck to the approach by Richardson et al. (1994). However, an update to Solomon and Qian (2005) is planned for future releases.

7. *P 16, L 1 of the reviewed manuscript:*
   *I am not sure I understand the situation with GWD. Is it off in the standard ICON, but on in ICON-UA? Then the comparison between them is difficult because there are differences between models such as GWD and presence of sponge layers.*

   We agree that we have not highlighted enough the modification of the Hines parameterization and the presence of the sponge layer in the two simulation configurations named "ICON" and "ICON(UA)", which have a model top at an altitude of 80 km. The standard ICON simulations use the Hines parameterization, but do not use its optional modification of the turbulent mixing coefficients. In contrast, the ICON(UA) (and UA-ICON) simulations make use of the optional modification of the turbulent mixing coefficients by the Hines scheme. To be more clear, we changed the formulations on P 13 L 6 of the revised manuscript (see P 18 L 6 of the tracked manuscript).

8. *P 16, L 6 of the reviewed manuscript:*
   *It is not correct. Lyman-alpha and SRB can contribute down to 60 or even 50 km.*

   Thank you. In the new climatological simulations for section 3.2.2, mentioned above under point 1, we changed the start height at least for the Schumann-Runge bands and continuum from 60 to 50 km. The start heights of all processes are now listed in the third column of table 1 on P 12 of the revised manuscript.

9. *P 17, L 16 of the reviewed manuscript:*
   *why 0.23?*

   We apologize for a sloppy formulation. However, double checking this issue made us aware of a bug in the code, which unfortunately we only spotted after rerunning all the simulations for this paper. As we don't think this will have a major influence on the model results, we suggest to stick with the existing simulations. We have added a better description of the approach to account for the energy that is a) going into breaking chemical bonds, and b) lost due to airglow, and a footnote to mention the bug (see P 14 L 21 and P 15 of the revised manuscript).

10. *P 29, L 21-22 of the reviewed manuscript:*
    *Unidentified bug? What is the reason to assume that this limitation is not important?*

    In the meantime, we could identify the reason, why the simulations become unstable, if the vertical variation of the gravitational acceleration (see Eq. (11), P 9 in the revised manuscript) is taken into account. It is related to too large heating rates from the parameterization of vertical turbulent mixing in the ECHAM

physics package of ICON. This results from certain computations within the parameterization that become inconsistent with each other, if the gravitational acceleration is no longer constant. We were able to fix this problem, and the new simulations for section 3.2.2, mentioned under point 1 above, include the vertical variation of the gravitational acceleration. The related sentences have been removed from the manuscript (see P 33 L 15 and P 43 L 15 of the tracked manuscript).

11. *P 30, L 1 of the reviewed manuscript:*
    *Would it be possible to discuss the role of damping when you compare different model versions. It can be rather dramatic.*

    On P 24 L 7 (and P 27 L 6) of the revised manuscript we tried to emphasize more strongly the impact of the sponge layer on the comparison of the three configurations "UA-ICON", "ICON" and "ICON(UA)".

12. *P 30, L 27 of the reviewed manuscript:*
    *Which minimum is discussed? It is not visible.*

    We refer to the minima visible in the SABER temperature climatologies, Fig. 7 top left and bottom right on P 25 of the revised manuscript. More precisely, the altitude range enclosed by the 200 K-isoline near 100 km, at about 80° North in Fig. 7 top right, and at about 80° South in Fig. 7 bottom right. We clarified this in the text on P 24 L 22 of the revised manuscript.

13. *P 31, L 1 of the reviewed manuscript:*
    *It would be interesting evaluate the influence of dynamical and physical processes separately.*

    Thank you for this suggestion. For the revised manuscript, we performed a further simulation variant, denoted UAphys-ICON, which is identical to the UA-ICON simulations, except for the deep-atmosphere dynamics being switched off. Through the comparison of the results from UAphys-ICON with the results from UA-ICON, we try to quantify the difference between shallow-atmosphere and deep-atmosphere dynamics. The description of this new simulation variant can be found on P 24 L 3 of the revised manuscript, and the discussion of the results on P 27 L 10 to P 29 L 11 of the revised manuscript. Corresponding additions to section 4 (Conclusions) are on P 30 L 30 and P 31 L 10 of the revised manuscript.

14. *P 34, L 24 of the reviewed manuscript:*
    *Clarify good agreement. In some cases, the agreement is not so good.*

    We tried to formulate our conclusions more cautiously (see P 31 L 1 to L 7 of the revised manuscript, and P 43 L 23 to L 33 of the tracked manuscript).

15. *P 35, L 1-2 of the reviewed manuscript:*
    *It would be interesting comment on the influence of dynamical and physical processes separately.*

    See our response to point 13 above.

<h1 style="text-align:center">Response to the Comments by Referee 2</h1>

Dear Referee,

Thank you very much for your review. We appreciate the friendly and constructive comments. They have been very helpful in further improving the manuscript. Please find below our point-by-point response. We refer to the version of the manuscript reviewed by the Referee as "reviewed manuscript", and to the revised version as "revised manuscript". In addition, we provide a version of the manuscript, in which all changes are tracked at the end of this document. Text passages, which have been modified or deleted, are marked red and crossed out. New text is marked blue. We refer to this tracked version as "tracked manuscript" in the response below.

**Major points**

- *The contents of this paper is certainly of interest to the wider community and suitable to GMD, my main comment however is that I found the paper to be somewhat long, which in of itself is not a problem, but it is quite wide in its scope in both providing the modifications to the dynamical core, a detailed analysis of the dynamical cores tests, as well as broader descriptions of the extension of the parameterizations and the more complex climatological test. I think this paper could therefore benefit from re-factoring somewhat and tightening up.*

We see the point of the Referee and we made several changes to the manuscript, in order to tighten the text and improve its readability. This was mainly achieved by deleting and abbreviating passages from sections 2.1 and 3.1 on the model dynamics and its evaluation, and moving some technical details to a new appendix. We hope that these measures improve the balance between the description of the upper-atmosphere physics and the deep-atmosphere dynamics. The corresponding modifications to the manuscript are as follows:

P(age) 5 L(ine) 28 to L 31 of the reviewed manuscript:
The reference to the entropy and potential temperature equations were made because of the remarks on the source $Q$ on P 8 L 8 to P 9 L 29 of the reviewed manuscript. These remarks have been tightened significantly (see subsequent modification) and the reference to the entropy and potential temperature equations is no longer necessary, so we removed it (see P 6 L 24 to L 27 of the tracked manuscript).

P 8 L 8 to P 9 L 29 of the reviewed manuscript:
We think this part on the dissipative and diabatic processes in Eqs. (1) and (3) has become a too exhaustive description of what we have actually not modified for the deep atmosphere in the dynamical core. For this reason we significantly tightened this part (see P 8 L 18 of the revised manuscript, and P 9 L 20 to P 11 L 17 of the tracked manuscript).

Section 2.1.3, P 12 L 24 to P 14 L 26 reviewed manuscript:
We assume that the details of the model initialization might appear rather technical. For this reason, we moved a large part of this section into a new appendix "Appendix A: Model initialization" (see P 11 L 13 and P 31 L 27 to P 34 L 6 of the revised manuscript, and P 14 L 19 to P 16 L 22 and P 44 L 20 to P 46 L 30 of the tracked manuscript).

Figure 7, P 25 of the reviewed manuscript:
We removed this figure on the convergence of the solution to the Jablonowski-Williamson test case with regard to the vertical grid resolution. To mention in the text that the solution is largely converged with respect to the vertical resolution should be enough, as suggested by the Referee (see P 21 L 15 of the revised manuscript, and P 27 L 10 and P 29 of the tracked manuscript).

P 25 L 5 (middle) to P 29 L 12 of the reviewed manuscript:
In that passage we tried to list possible reasons for a numerical instability that occurred during testing the horizontal grid R3B7 for the Jablonowski-Williamson test case (compare Fig. 6, P 24 of the reviewed manuscript). One of the difficulties in finding the reason for this instability lies in the fact that if we switch off certain parts of our deep-atmosphere implementations in the dynamical core, in order to exclude that they contribute to the instabilty, the prescribed analytical solution for the atmospheric background state of this test case would no longer fit to the modified dynamical core. As a consequence the model atmosphere undergoes a more or less strong adjustment process right away from the beginning of a simulation. This renders the usefulness of such simulations highly questionable. Determining new analytical solutions for the atmospheric background state for the deep-atmosphere equations with some parts switched off would be too cumbersome. To conclude, the possible reasons for the instability listed in the reviewed manuscript

were speculative and still are. As such, we think they are not essential for the manuscript and we decided to remove that passage and the appendix B, P 39 to P 43 belonging to it. A note on the numerical instability and the fact that we were not yet able to clearly identify its reason remains on P 21 L 27 to P 22 L 11 of the revised manuscript (see P 28 L 9 to P 33 L 11 of the tracked manuscript).

- *I think this paper could be split into two papers, the first could detail the changes to the dynamical core to simulate deep atmosphere dynamics along with the results from Section 3.1 and the appendices of which I think there is enough material in this paper to recommend publication of as it stands. The second paper would detail the specification of ICON-UA, the extension of the parameterizations (Section 2.2) along with test case in Section 3.2. The changes to model initialization (Section 2.1.3) could be relevant to both areas and it is up to hte authors where they think this would best sit.*

We see that covering the deep-atmosphere dynamics and the upper-atmosphere physics in one single paper leads to a rather long manuscript. Nevertheless, we would very much prefer not to split the paper into two papers, since we regard both the upper-atmosphere physics package and the deep-atmosphere modification of the dynamical core as integral parts of the upper-atmosphere extension of the ICON model. Both contribute in their respective field to the goal of a realistic simulation of the upper atmosphere. We hope that the tightening up of the manuscript mentioned above improves the readability of the manuscript to the point that the Referee would agree with keeping the upper-atmosphere physics and the deep-atmosphere dynamics as two parts of one paper.

**Minor points**

1. *P 8, L 2 of the reviewed manuscript:*
   *The equation in this line contains terms $\partial/\partial t$ referring to differentiation with respect to the tangential direction, whilst equations (8) & (9) use $\partial/\partial t$ for temporal differentiation. Could the authors change either of the symbols here to avoid confusion?*

   Thank you for pointing us to this flaw. We corrected it (see Eqs. (9) & (10) on P 8 of the revised manuscript). In addition, we replaced the symbols t and n for the edge-tangential and -normal directions by the symbols $\mathfrak{t}$ and $\mathfrak{n}$, respectively (see, e.g., Eqs. (9) & (10) on P 8 of the revised manuscript). We hope that by this means they are easier to distinguish from the symbols $t$ for the time and $n$ for numbers (e.g., in Eq. (18) on P 17 and Eq. (20) on P 21 of the revised manuscript).

2. *Section 2.1.3 of the reviewed manuscript:*
   *I think that this change to initialization is problem many forecasting centers will have moving to the deep atmosphere equations with high lids and I am glad to see the authors address this, however I found the interaction of the different profiles ($T_B$, $T_F$, $T_\infty$, $T_{120\,km}$, $T_{IFS}$) a little hard to follow and think this could be helped by a simple diagram showing the regions in which each profile is used and the blending regions where they overlap.*

   We agree with the Referee and decided to move the rather technical details of the model initialization from section 2.1.3 on P 12, L 24 to P 14, L 26 of the reviewed manuscript to a new appendix "Appendix A: Model initialization" on P 31 to P 34 of the revised manuscript. In addition, we followed your suggestion and added Figure A1 on P 32 of the revised manuscript. It depicts the vertical temperature profile employed to compute the climatological part of the blending along with some symbols used in the text. Furthermore, we removed the dependency on the longitude $\lambda$ and latitude $\varphi$ of the temperature profile $T_B$ from Bates (1959), since in the current formulation there is no such dependency (Eq. (23), P 13 of the reviewed manuscript, Eq. (A2), P32 of the revised manuscript). The symbols $\mathfrak{T}_{IFS}$ in Eq. (24), P 13 of the reviewed manuscript and $\mathfrak{p}_{IFS}$ on P 14 L 9 of the reviewed manuscript are also removed. Their introduction is not necessary.

3. *Table 1, P 15 of the reviewed manuscript:*
   *It would be useful if the authors could add the height at which these different processes (which is described in the text) are used into table 1.*

   Thank you for this suggestion. We added a third column to table 1 containing the heights above which a computation of tendencies from the respective processes starts (see P 12 of the revised manuscript).

4. *P 20, L 27 of the reviewed manuscript:*
   *"$\Delta t = \eta_1 \cdot 13.2\,s$ fits both the maximum magnitude ..." It's unclear to me what is being said here. Do you mean that this timestep fits any CFL restriction of the model in both configurations? or something else?*

   Yes, this time step is used for both configurations. We tried to clarify this by replacing the original formulation by "We use a time step of $\Delta t = \eta_1 \cdot 13.2\,s$ for both configurations. It should satisfy the CFL criterion in

both configurations. The maximum propagation velocity in the first configuration is $|v|_{\max} = c_s = 317\,\mathrm{m\,s^{-1}}$, whereas in the second configuration it is $|v|_{\max} = |u_F|_{\max} + c_s = 134\,\mathrm{m\,s^{-1}} + 317\,\mathrm{m\,s^{-1}} = 451\,\mathrm{m\,s^{-1}}$." (see P 18, L 3 of the revised manuscript).

5. *Figure 4, P 22 of the reviewed manuscript:*
   *The caption states that the isolines are the analytical solution but these appear very noisy in panels (a) and (b) (particularly the zero contour) which I would not expect from the solution given in the appendices. Is this correct?*

   The reason for the noisy contour lines is that we computed the analytical solution as a run time diagnostic output on the same grid as the numerical solution. This was easier to realize than writing an extra postprocessing script for this purpose. In order to improve the contour lines in the revised manuscript, we plotted the analytical solution from the grid R2B7L360, mentioned in the caption of Fig. 5 on P 23 of the reviewed manuscript, over the numerical solution from the grid R2B6L180, already shown in Fig. 4. This yields smoother contours. In addition, we removed the zero contour, since it appears still a bit noisy on the grid R2B7L360 and is not of particular interest. A corresponding note has been added to the caption of Fig. 4 on P 19 of the revised manuscript.

6. *P 23, L 14 of the reviewed manuscript:*
   *How and why has the vertical grid been changed from the Ullrich et.al. paper? (Given the results from figure 7 it appears unlikely that this will have changed the answers but if possible could the formula (or reason) used here be given?*

   We skipped over this point, because the convergence tests with regard to the vertical grid resolution made us believe that the differences between using the vertical grid given in Ullrich et al. (2014) and using the vertical grid of ICON are negligible. On P 21 L 7 of the revised manuscript we extended the information on the vertical grid, by stating the formula for the grid layer interface heights used in Ullrich et al. (2014) (Eq. (20)) and the corresponding formula for the vertical grid of ICON (Eq. (21)). In addition, we added a note that our convergence tests make us relatively confident that the numerical solution would not look much different, if we would have used the vertical grid of Ullrich et al. (2014) (see P 21, L 15 of the revised manuscript).

7. *Figure 7, P 25 of the reviewed manuscript:*
   *In order to condense the paper I think this figure could be removed and it simply stated that the test appears converged with regards to vertical resolution.*

   Thank you for this suggestion. We followed it and removed Fig. 7 of the reviewed manuscript. A note on the convergence test with regard to the vertical grid resolution remains in the text (see P 21, L 21 of the revised manuscript).

8. *P 26, L 5 of the reviewed manuscript:*
   *Replace "ragard" with "regard"*

   Thank you. The text passage, which contains this error has been removed (see our response to the first major point above).

9. *P 28, L 25 of the reviewed manuscript:*
   *Replace "mentionde" with "mentioned"*

   Thank you. The text passage, which contains this error has been removed (see our response to the first major point above).

10. *P 29, L 22 of the reviewed manuscript:*
    *I think the authors are correct and this limitation will not affect results greatly but it would be better if they could quantify this belief somewhat instead of just assuming*

    In the meantime we could identify the reason, why the simulations become unstable, if the vertical variation of the gravitational acceleration (see Eq. (11), P 9 in the revised manuscript) is taken into account. It is related to too large heating rates from the parameterization of vertical turbulent mixing in the ECHAM physics package of ICON. This results from certain computations within the parameterization that become inconsistent with each other, if the gravitational acceleration is no longer constant. We were able to fix this problem. The climatological simulation with UA-ICON, which we repeated for section 3.2 of the revised manuscript*, include now the vertical variation of the gravitational acceleration. The related sentences have been removed from the manuscript (see P 33 L 15 and P 43 L 15 of the tracked manuscript).

    * In the meantime we found a flaw in the data of the gases, which are required for the parameterization of the radiative processes listed in Table 1, P 15 of the reviewed manuscript. This flaw is removed, and we

repeated the simulations of the two configuration variants that make use of the upper-atmosphere physics package (denoted UA-ICON and ICON(UA), see section 3.2.2, P 30 to P 33 of the reviewed manuscript). With the corrected gas data the results for the zonally averaged temperature and zonal wind component are better, when compared to the satellite retrievals, than the results presented in the reviewed manuscript (compare the reviewed and revised versions of Fig. 9 on P 35 and 36, and Fig. 10/11 on P 38 and 39 of the tracked manuscript).

[revised manuscript text omitted]
_{tt}}(\boldsymbol{r})$ and $\boldsymbol{e_{nn}}(\boldsymbol{r})$, in order to simplify matters.

**(a)**

[Figure]

**(b)** Revised version

[Figure]

**Figure 1.** Schematic illustration of a global horizontal triangular grid as used by ICON ( left), and of a grid cell ( right). The black dashed lines illustrate a grid  of R2B0-type (twofold root division, zero bisections).  C-grid staggering is used for the prognostic variables, i.e. all scalars like density $\rho$, potential temperature $\theta$, and Exner pressure $\pi$ are defined in the cell center (red dot), tangential and normal wind components $v_{\underline{\mathrm{t}}\mathrm{t}}$, and $v_{\underline{\mathrm{n}}\mathrm{n}}$ on the side faces (corresponding to the three primal edges of a triangle on the horizontal grid), and vertical wind component $w$ on the bottom and top surfaces of the cell (assuming $v_{\underline{\mathrm{t}}\mathrm{t}}, v_{\underline{\mathrm{n}}\mathrm{n}}, w > 0$ for the cell drawing).

we would find $e_{\underline{n}n}$ being parallel to $e'_{\varphi}$ (the radial unit vectors of both systems are parallel anyway). The governing equations for the velocity components $v = u e_{\lambda} + v e_{\varphi} + w e_r$ (or here $v = u' e'_{\lambda} + v' e'_{\varphi} + w e_r$ ) can be found in many textbooks (e.g., Gill, 1982; Zdunkowski and Bott, 2003; Holton, 2004; Vallis, 2006). Evaluating them at the equator ($\varphi^{(\prime)} = 0$) yields almost immediately the components of the governing equations for $v = v_{\underline{t}t} e_{\underline{t}t} + v_{\underline{n}n} e_{\underline{n}n} + w e_r$ in the local edge coordinate systems,

5   if we identify $v_{\underline{t}t}$ with $u'$ and $v_{\underline{n}n}$ with $v'$. The equation for the vertical velocity component $w$ in the local coordinate system defined in the center of the vertical cell interfaces can be found in a similar way. Together they read

$$\frac{\partial v_{\underline{n}n}}{\partial \underline{t}t} + \left\{\frac{a}{r}\right\} \frac{\partial K_h}{\partial \underline{n}n} + (\zeta + f_r) v_{\underline{t}t} + w \left(\frac{\partial v_{\underline{n}n}}{\partial r} + \underline{\frac{v_{\underline{n}n}}{r}} - \underline{f_{\underline{t}t}}\right) + \underline{\Omega^2 r \sin(\varphi)\cos(\varphi) e_{\varphi} \cdot e_{\underline{n}n}} = -c_p \theta \left\{\frac{a}{r}\right\} \frac{\partial \pi}{\partial \underline{n}n} + \frac{F_{\underline{n}n}}{\rho}, \qquad (9)$$

$$\frac{\partial w}{\partial \underline{t}t} + v_{\underline{n}n}\left(\left\{\frac{a}{r}\right\}\frac{\partial w}{\partial \underline{n}n} - \underline{\frac{v_{\underline{n}n}}{r}} + \underline{f_{\underline{t}t}}\right) + v_{\underline{t}t}\left(\left\{\frac{a}{r}\right\}\frac{\partial w}{\partial \underline{t}t} - \underline{\frac{v_{\underline{t}t}}{r}} - \underline{f_{\underline{n}n}}\right) + w\frac{\partial w}{\partial r} - \underline{\Omega^2 r \cos^2(\varphi)} = -c_p\left[(\theta_0 + \theta')\frac{\partial \pi'}{\partial r} + \theta'\frac{d\pi_0}{dr}\right] + \frac{F_r}{\rho},$$

$$(10)$$

where $K_h = (v_{\underline{t}t}^2 + v_{\underline{n}n}^2)/2$ is the horizontal mass-specific kinetic energy, and $f_r = 2\Omega\sin(\varphi)$, $f_{\underline{t}t,\underline{n}n} = 2\Omega\cos(\varphi)e_{\varphi}\cdot e_{\underline{t}t,\underline{n}n}$

10  are the Coriolis parameters. The values of the projections $e_{\varphi}\cdot e_{\underline{t}t,\underline{n}n}$ are already provided by the standard shallow-atmosphere configuration of ICON, so they pose no additional problem. Here and in the following we will formulate the deep-atmosphere equations as a modification of the shallow-atmosphere equations. This simplifies the comparison and corresponds actually to the way we have implemented the deep-atmosphere dynamics in ICON. For instance, we made use of the expansion of the gradient in the local coordinate systems $\nabla = e_{\underline{t}t}\{a/r\}\partial/\partial\underline{t}t + e_{\underline{n}n}\{a/r\}\partial/\partial\underline{n}n + e_r\partial/\partial r$, if $\nabla = e_{\underline{t}t}\partial/\partial\underline{t}t + e_{\underline{n}n}\partial/\partial\underline{n}n +$

[revised manuscript text omitted]
 $B$, therefore depending only on $|x - B| = |(x - A) - (B - A)| = \{[(x - A) - (B - A)] \cdot [(x - A) - (B - A)]\}^{1/2}$, we would find the solution unaltered in the rotating frame relative to $\exp(-Wt) \cdot (B - A)$, since e.g.

$$[\exp(-Wt) \cdot (X - A)] \cdot [\exp(-Wt) \cdot (B - A)] = [(X - A) \cdot \exp(-Wt)^\intercal] \cdot [\exp(-Wt) \cdot (B - A)]$$

$$= (X - A) \cdot [\exp(Wt) \cdot \exp(-Wt)] \cdot (B - A)$$

$$= (X - A) \cdot (B - A). \tag{B8}$$

Here, $(\cdot)^\intercal$ denotes the transpose, and we have used the identity $\exp(-Wt)^\intercal = \exp(Wt)$ which can be derived from Eq. (B6) and $\widehat{\Omega}\widehat{\Omega}^\intercal = \widehat{\Omega}\widehat{\Omega}$, $\mathbf{1}^\intercal = \mathbf{1}$ and $\widehat{W}^\intercal = -\widehat{W}$. In addition, $\exp(Wt) \cdot \exp(-Wt) = \
[revised manuscript text omitted]

$$= \frac{1}{2}D^4\mathbf{1}\underset{\times}{\times}\mathbf{1} - 2D^3\mathbf{1}\underset{\times}{\times}\mathbf{W} + D^2\mathbf{1}\underset{\times}{\times}\mathbf{S} + 2D^2\mathbf{W}\underset{\times}{\times}\mathbf{W} - 2D\mathbf{S}\underset{\times}{\times}\mathbf{W} + \frac{1}{2}\mathbf{S}\underset{\times}{\times}\mathbf{S}, \tag{C6}$$

where we have used that $\mathbf{1}$ and $\mathbf{S}$ are symmetric tensors, and $\mathbf{W}$ is an antisymmetric tensor in the second step, and the commutativity of the double cross product in the third step (e.g., $\mathbf{S}\underset{\times}{\times}\mathbf{W} = \mathbf{W}\underset{\times}{\times}\mathbf{S}$). In the following, we use a number of identities, the proof of which is beyond the scope of this work. We have to refer to the literature cited in footnote 7, but typically they can be proven directly using, for instance, the expansion of a tensor in a sum of (three) dyadic products, the laws of associativity and commutativity of the respective products, the expansion $\mathbf{a} \times (\mathbf{b} \times \mathbf{c}) = \mathbf{b}\mathbf{a} \cdot \mathbf{c} - \mathbf{c}\mathbf{a} \cdot \mathbf{b}$, and $\mathbf{a} \cdot (\mathbf{b} \times \mathbf{c}) = \mathbf{c} \cdot (\mathbf{a} \times \mathbf{b}) = \mathbf{b} \cdot (\mathbf{c}$
Since the determinant of the identity tensor is 1, it follows from Eqs. (C4) and (C5) that $(1/2)\mathbf{1}\underset{\times}{\times}\mathbf{1} = \mathbf{1}$. In addition, $\mathbf{1}\underset{\times}{\times}\mathbf{W} = \mathbf{W}_\bullet\mathbf{1} - \mathbf{W}^\mathsf{T} =$
holds, where we have used that $\mathbf{W}$ is antisymmetric and $\mathbf{W}_\bullet = (\mathbf{W}^\mathsf{T})_\bullet = -\mathbf{W}_\bullet = 0$. Likewise $\mathbf{1}\underset{\times}{\times}\mathbf{S} = \mathbf{S}_\bullet\mathbf{1} - \mathbf{S}^\mathsf{T} = \mathbf{S}_\bullet\mathbf{1} - \mathbf{S}$, using the symmetry of $\mathbf{S}$. Finally, $\mathbf{W}\underset{\times}{\times}\mathbf{W} = (\boldsymbol{\Omega} \times \mathbf{1})\underset{\times}{\times}(\boldsymbol{\Omega} \times \mathbf{1}) = 2\boldsymbol{\Omega}\boldsymbol{\Omega}$. If we insert them in Eq. (C6), we find

$$\widetilde{\{\cdot\}}= D^4\mathbf{1} - 2D^3\mathbf{W} + D^2\left(\mathbf{S}_\bullet\mathbf{1} - \mathbf{S}\right) + 4D^2\boldsymbol{\Omega}\boldsymbol{\Omega} - 2D\mathbf{S}\underset{\times}{\times}\mathbf{W} + \frac{1}{2}\mathbf{S}\underset{\times}{\times}\mathbf{S}. \tag{C7}$$

Following Eq. (C4), the determinant reads then

[revised manuscript text omitted]

---

## Author Response (AR2)

Dear Dr. Añel,
Dear Referees,

Thank you very much for handling our manuscript, and for your review. It was a great help to us. Please, find below our response to the comments. A version of the manuscript, in which all changes are tracked, is attached to this file.

**Response to the comment by Referee 2**

- *I would suggest changing the references to the Unified Model on line 5, page 3 to Davies et.al. 2005 "A new dynamical core for the Met Office's global and regional modelling of the atmosphere". Q. J. R. Meteorol. Soc. 131:1759-1782 and Wood et.al. QJRMS 2014 as these are the more relevant model description papers than those currently cited.*

  Thank you. We changed the references accordingly (see page 3, line 5 of the tracked manuscript).

**Response to the comments by Referee 3**

- *1. Upper boundary condition: According to the paper, the vertical wind at the upper boundary is set to 0 in UA-ICON. 0 vertical wind boundary is used in many GCMs, and I think this is reasonable for hydrostatic models with pressure coordinate, since the isentropic surface can be approximately regarded as a material surface. I am not sure what is the effect of the zero vertical wind boundary in a nonhydrostatic GCM with geometric height coordinate. Has this been examined in UA-ICON by comparing with the hydrostatic vertical of the model (UAphys-ICON)?*

  The rigid upper boundary is an important element to guarantee global mass conservation in ICON. It is an integral part of the dynamical core and it is not straightforward to replace it by another boundary condition. So we can only speculate on the effects of setting $w = 0$ at the model top. There might be column-integrated effects, such as an imprint on the surface pressure, when synoptic scale updrafts (e.g., due to large-scale convergent horizontal flows) are blocked by the rigid lid. The higher the model top the smaller such integral effects likely become. More local effects become probably significant the closer the model top, e.g., due to wave reflection (which is damped by the sponge layer).

  In our attempt to explain the temperature difference between UA-ICON and UAphys-ICON, we hypothesized that the difference becomes smaller, if we compare the temperatures within a coordinate system that uses the mass of air in a grid cell column between the model top and some height $z$ as the vertical coordinate. To simplify the comparison with the geometric height as vertical coordinate, we tried to translate the mass coordinate into a height-like scale. For this purpose we employed the hydrostatic balance. However, it is well possible that this translation becomes worse the closer the model top. (Just to make sure that there was no misunderstanding: all our simulations (including UAphys-ICON) are non-hydrostatic.) Apart from that, one of the main effects of the model top is that the optical thickness in zenith direction becomes zero there, because the model neglects the air above it. This might be one of the reasons why our approach with the mass coordinate does not work above about 130 km, as can be seen in Fig. 11. Our formulation of this paragraph was probably too vague. We tried to improve this. In addition, we included the results from

the simulations of three further winters into the analysis (now six winters in total are anlyzed: 2013/14, 2014/15, 2015/16 and newly 2016/17, 2017/18 and 2018/19), in order to improve the statistics. However, there is practically no difference visible in the plots shown in Fig. 11, so three winters yield already good mean values. Furthermore, we inspected the heating rates from the short-wave radiative processes in the altitude range of significant temperature differences between UAphy-ICON and UA-ICON at the beginning of the simulations, when the model atmosphere is still in its hydrostatically balanced initial state. They are significantly stronger in UAphys-ICON than in UA-ICON, as might be expected, if the optical thickness of the overlying air is different. We mentioned that in the text, too. See page 28, line 9 to page 29 line 28 and Fig. 11 on page 30 of the tracked manuscript.

- *2. Page 8, line 17: Does this sentence mean v_t is derived from v_n and w from continuity equation? I don't understand how v_t can be calculated from v_n alone.*

  We apologize for being sloppy in our description of how $v_t$ is diagnosed. The continuity equation (see eq. (2) on page 5 of the tracked manuscript) is not used to diagnose $v_t$. Instead a radial basis function ansatz is used to interpolate the unknown tangential component $v_t \boldsymbol{e_t}$ at the center of an edge from the known $v_n \boldsymbol{e_n}$ at the edges within some specified vicinity. We tried to clarify this in the text. In addition, we added the reference to the Ph.D. thesis of H. Wan (2009), where the details of the radial basis function interpolation of the horizontal wind are described in the appendix D. See page 8, line 17.

- *3. Page 13, lines 14-17: Is 105-125nm (including Lynman-alpha line) considered? It is missing from this description.*

  No, we don't consider this wavelength range explicitly in our heating rate parameterizations. In this sense, the text in the manuscript is correct in referring only to the Strobel ($> 125$ nm) and Richards ($< 105$ nm) approaches used in the model. Although obviously inaccurate, this omission is not untypical for models covering this altitude range (see e.g. the Extended CMAM (Fomichev et al., 2002), and we think it is not inappropriate for our purposes. According to Brasseur and Solomon (2005, their Figure 4.34) at most altitudes, the contribution of Ly-alpha to the photodissociation of O2 (and hence to heating) is at least an order of magnitude smaller than that of either SRB or SRC. Only in a small altitude region near 75 km, the contribution of Lyman-alpha is only by a factor of about 3 smaller than that of SRB. However, even here we implicitly include about half of the energy deposition of Lyman-alpha through the consideration of chemical heating rates calculated with the HAMMONIA model which does include Lyman-alpha.

**Further changes**

- Page 6, line 1 of the tracked manuscript: add forgotten definition of $\theta'$, introduced in Eq. (6).

- Page 28, line 1, 2 and 6: remove typos, improve formulations.

- Page 29, line 32: replace "upper atmosphere" by "thermosphere", to be more precise.

- Page 30, line 1: remove typo.

- Page 32, line 16, 20 and 21: remove typos.

- Page 32, line 16: the information on our last access to the webpage with the license information is probably not necessary there.

- We updated the Supplementary Material to our manuscript, which contains the data shown in section 3 of our manuscript and the scripts for their production, postprocessing and plotting.

We hope that we could address the concerns of the referees adequately. With kind ragards, also on behalf of all co-authors,

Sebastian Borchert

[revised manuscript text omitted]

where $T_{120\text{km}} = T_\text{F}(z_g = 120\,\text{km})$, and $T_\infty$ is approximately the temperature for the limit $z \to \infty$. This limit corresponds to a 10 geopotential height of $z_g = a$, which follows from Eq. (A1) by multiplying the right-hand side of the first equation with $1 = (1/z)/(1/z)$ and applying the limit. The value of $T_\infty$ could be set, for instance, to the mean exospheric temperature of about $1035\,\text{K}$ (in practice we use a value of $400\,\text{K}$, since higher values could challenge the numerical stability in the initial phase of the simulation). For a steady transition between $T_\text{F}$ and $T_\text{B}$, the scale height is set to $H_\text{B} = (T_\infty - T_{120\text{km}})/(\text{d}T_\text{F}/\text{d}z_g)_{z_g=120\,\text{km}}$. In addition, the temperature blending requires extrapolating the temperature data from below $\mathfrak{z}_g$ to above, which is done by a

simple linear extrapolation

$$T_{\mathrm{IFS}}(\lambda,\varphi,z_g) = T_{\mathrm{IFS}}(\lambda,\varphi,z_g = \mathfrak{z}_g) + \gamma(\lambda,\varphi)\,(z_g - \mathfrak{z}_g)\,, \tag{A3}$$

with $\gamma = \mathrm{d}T_{\mathrm{IFS}}/\mathrm{d}z_g|_{z_g=\mathfrak{z}_g}$. To obtain a statically stable stratification, $\gamma$ is limited by the dry adiabatic lapse rate: $\gamma \geq -\Gamma_d = -g/c_p$ (e.g., Holton, 2004). The blending reads

5  $$T(\lambda,\varphi,z_g) = T_{\mathrm{IFS}}(\lambda,\varphi,z_g)\alpha(z_g) + T_{\mathrm{clim}}(z_g)\,[1-\alpha(z_g)]\,, \tag{A4}$$

$$\alpha(z_g) = \begin{cases} 1, & \text{for} \quad z_g < \mathfrak{z}_g \\ \frac{1}{2}\left[1 + \cos\left(\frac{z_g - \mathfrak{z}_g}{H_{\mathrm{blend}}}\pi\right)\right], & \text{for} \quad \mathfrak{z}_g \leq z_g \leq \mathfrak{z}_g + H_{\mathrm{blend}} \\ 0, & \text{for} \quad \mathfrak{z}_g + H_{\mathrm{blend}} < z_g \end{cases} \cdot \tag{A5}$$

where $T_{\mathrm{clim}}$ is $T_{\mathrm{F}}$ or $T_{\mathrm{B}}$, respectively, and $H_{\mathrm{
[revised manuscript text omitted]